# Specific presynaptic functions require distinct *Drosophila* Ca$_v$2 splice isoforms

Christopher Bell[1†], Lukas Kilo[2†], Daniel Gottschalk[1], Jashar Arian[1], Lea Deneke[1], Hanna Kern[1], Christof Rickert[1], Oliver Kobler[3], Julia Strauß[1], Martin Heine[1], Carsten Duch[1], Stefanie Ryglewski[1]*

[1]Johannes Gutenberg University Mainz, Institute of Developmental Biology and Neurobiology, Biocenter 1, Mainz, Germany; [2]RWTH Aachen University, Lehrstuhl für Entwicklungsbiologie, Aachen, Germany; [3]Leibniz Institute for Neurobiology Magdeburg, Combinatorial NeuroImaging Core Facility, Magdeburg, Germany

## eLife Assessment

Cav2 voltage-gated calcium channels play key roles in regulating synaptic strength and plasticity. In contrast to mammals, invertebrates like *Drosophila* encode a single Cav2 channel, raising questions on how diversity in Cav2 is achieved from a single gene. Here, the authors present **solid** evidence that two alternatively spliced Cac isoforms enable **important** changes in Cav2 expression, localization, and function in synaptic transmission and plasticity at the *Drosophila* neuromuscular junction. How the isoforms affect synaptic calcium channel levels remains less clear. This study provides insights into the roles of voltage-gated calcium channel splice isoforms in synaptic transmission.

*For correspondence:
ryglewsk@uni-mainz.de

†These authors contributed equally to this work

Competing interest: The authors declare that no competing interests exist.

**Abstract** At many vertebrate synapses, presynaptic functions are tuned by expression of different Ca$_v$2 channels. Most invertebrate genomes contain only one *Ca$_v$2* gene. The *Drosophila* Ca$_v$2 homolog, cacophony (cac), induces synaptic vesicle release at presynaptic active zones (AZs). We hypothesize that *Drosophila* cac functional diversity is enhanced by two mutually exclusive exon pairs that are not conserved in vertebrates, one in the voltage sensor and one in the loop binding Ca$_\beta$ and G$_{\beta\gamma}$ subunits. We find that alternative splicing in the voltage sensor affects channel activation voltage. Only the isoform with the higher activation voltage localizes to AZs at the glutamatergic *Drosophila* larval neuromuscular junction and is imperative for normal synapse function. By contrast, alternative splicing at the other alternative exon pair tunes multiple aspects of presynaptic function. While expression of one exon yields normal transmission, expression of the other reduces channel number in the AZ and thus release probability. This also abolishes presynaptic homeostatic plasticity. Moreover, reduced channel number affects short-term plasticity, which is rescued by increasing the external calcium concentration to match release probability to control. In sum, in *Drosophila* alternative splicing provides a mechanism to regulate different aspects of presynaptic functions with only one *Ca$_v$2* gene.

## Introduction

Information transfer at fast chemical synapses requires action potential triggered calcium influx through voltage-gated calcium channels (VGCCs) into the presynaptic terminal, which in turn initiates synaptic vesicle (SV) release through a complex cascade of biophysical and biochemical reactions (*Südhof, 2013*; *Südhof, 2014*; *Dittman and Ryan, 2019*). The millisecond temporal precision of SV release upon the arrival of an action potential in the axon terminal requires clustering of the VGCCs in a spatially restricted presynaptic specialization, named the presynaptic active zone (AZ; *Eggermann*

*et al., 2011*; *Südhof, 2012*; *Van Vactor and Sigrist, 2017*; *Dolphin and Lee, 2020*; *Emperador-Melero and Kaeser, 2020*). At the AZ, protein–protein interactions ensure a nanoscale coupling of VGCCs to SVs in the readily releasable pool (RRP) (*Kittel et al., 2006*), an arrangement that supports efficient and fast neurotransmission (*Eggermann et al., 2011*).

Despite these common organizational principles, fast chemical synapses are highly heterogeneous to accommodate the diverse computational requirements of different types of neurons and brain circuits (*Moser et al., 2023*; *Zhang et al., 2022*). One means of synapse diversification is to organize VGCCs heterogeneously at the AZ to modify either their nanoscale spatial relation to the SVs ready for release (*Dittman and Ryan, 2019*; *Rebola et al., 2019*), or the profiles of the calcium dynamics that shape release probability ($P_r$; *Zhang et al., 2022*). The calcium dynamics in nano- or microdomains are affected by VGCC number, clustering, and properties, and are thus strongly dependent on calcium channel subtype.

The vertebrate genome contains 10 genes encoding the $\alpha_1$-subunit of VGCCs that fall into three families ($Ca_v1$, $Ca_v2$, and $Ca_v3$; *Dolphin, 2009*). With the exception of $Ca_v1$ triggering release at retinal ribbon and auditory brain stem synapses (*Moser et al., 2020*), $Ca_v2$ usually mediates evoked release at other mammalian synapses. The three subtypes of $Ca_v2$ exhibit different biophysical properties (*Sheng et al., 2012*), which may explain their different contribution to SV release at a given synapse (*Li et al., 2007*). $Ca_v2.1$ and $Ca_v2.2$ trigger SV release at most synapses and $Ca_v2.3$ likely make only a small contribution (*Dietrich et al., 2003*). However, all three $Ca_v2$ subtypes may co-localize to the same synapse and contribute to SV release (*Takahashi and Momiyama, 1993*; *Wheeler et al., 1994*; *Wu et al., 1998*), and different types of mammalian synapses can utilize either only one subtype or combinations of $Ca_v2$ subtypes (reviewed in *Zhang et al., 2022*).

The *Drosophila* genome contains only three genes encoding $\alpha_1$-subunits of VGCCs, each one homologous to one vertebrate $Ca_v$ family (*Littleton and Ganetzky, 2000*). Therefore, the joint functions of mammalian $Ca_v2.1$, $Ca_v2.2$, and $Ca_v2.3$ are covered by only one *Drosophila* gene, namely *Dmca1A* (also named *cacophony* or *nightblindA*, *Smith et al., 1996*; *Smith et al., 1998*). Cacophony (cac) is, in fact, essential for fast synaptic transmission in *Drosophila* (*Kawasaki et al., 2002*) and loss of cac cannot be compensated for by the *Drosophila* counterparts of $Ca_v1$ or $Ca_v3$ (*Krick et al., 2021*). This suggests that *Drosophila* lacks the combinatorial logic of employing different blends of $Ca_v2.1$, $Ca_v2.2$, and $Ca_v2.3$ for fine-tuning of $P_r$ as present in mammals. However, the *Drosophila cac* locus contains two mutually exclusive alternative splice sites that are not present in the vertebrate *Ca_v2* gene family, one in the voltage sensor in the fourth transmembrane domain of the first homologous repeat (IS4) and one in the intracellular linker between the first and the second homologous repeats (I–II). Alternative splicing at these mutually exclusive sites may provide mechanisms to fine-tune action potential triggered calcium dynamics at presynaptic AZs by adjusting the numbers, nanoscale localization, and properties of presynaptic cacophony VGCCs. We test this hypothesis by employing the CRISPR–Cas9 technology to remove mutually exclusive exons at either the *IS4* or the *I–II* site and test the resulting functional consequences at the *Drosophila* neuromuscular junction (NMJ), a well-established model for presynaptic function of a fast glutamatergic synapse (*Atwood and Karunanithi, 2002*; *Harris and Littleton, 2015*).

We find that alternative splicing at the *IS4* site affects cac voltage activation and only one of both mutually exclusive splice events allows for presynaptic AZ localization and thus evoked synaptic transmission at a fast glutamatergic synapse. By contrast, alternative splicing at the *I–II* site does not affect presynaptic AZ localization, but it affects $P_r$, short-term plasticity, and presynaptic homeostatic plasticity.

## Results

*Cacophony* (*cac*) is located on the X-chromosome and contains multiple alternative splice sites but only two that are mutually exclusive (*Figure 1A*, see also enlarged schematic of the cac regions housing these two mutually exclusive exon pairs (green boxes); cac schematic in *Figure 1B*). The first one is located in the fourth transmembrane domain of the first homologous repeat (IS4), and thus affects the voltage sensor. Both alternative *IS4* exons are 99 bp long and encode 33 amino acids (AAs). The polypeptide encoded by *IS4A* (first alternative exon of the *IS4* locus) differs from that of *IS4B* in 13 AAs and contains one additional positively charged AA. The second mutually exclusive exon pair encodes part of the intracellular linker between homologous repeats I and II

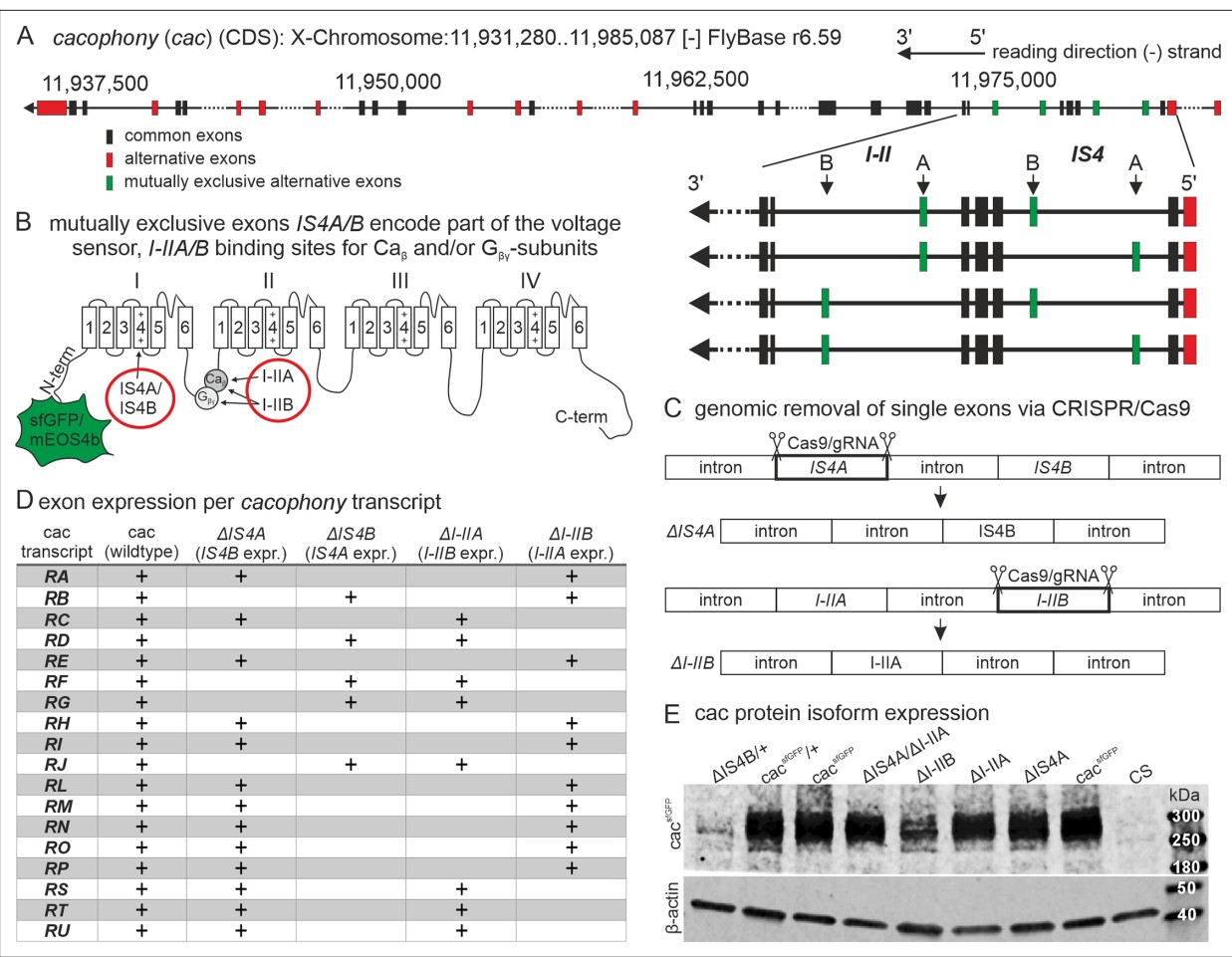

**Figure 1.** Cacophony alternative splicing gives rise to different protein isoforms. (**A**) *Cacophony* is located on the non-coding strand of the X-chromosome between positions 11,931,280 and 11,985,087 (Flybase version r6.59). Reading direction is indicated on the top right (arrow). Horizontal black lines indicate introns, larger introns are broken by dots. Black boxes indicate exons shared by all *cacophony* splice variants, red boxes indicate alternative exons, and green boxes represent exons that are mutually exclusively spliced. Enlarged area on the right emphasizes mutually exclusive exon pairs *IS4A/B* and *I–IIA/B*. (**B**) Schematic of cacophony with N-terminal sfGFP or mEOS4b tag and two exon pairs that are spliced mutually exclusively. *IS4A* and *IS4B* exons encode isoforms of the fourth transmembrane domain (S4) and thus part of the voltage sensor of the first homologous repeat (I) of the calcium channel, while *I–IIA* and *I–IIB* give rise to two versions of the intracellular linker between homologous repeats I and II. I–IIA contains a non-well conserved binding site for voltage-gated calcium channel β-subunits (Ca$_\beta$) whereas I–IIB gives rise to a conserved Ca$_\beta$-binding site as well as a binding site for G-protein βγ-subunits (G$_{\beta\gamma}$). (**C**) Genomic removal of one of the mutually exclusively spliced exons of one or two exon pairs by CRISPR/Cas9-mediated double strand breaks in the germ line results in exon-out mutants by imprecise excision of the cut exons and cell-intrinsic DNA repair. (**D**) *Cacophony* gives rise to 18 annotated transcripts (RA-RU, left). Multiple variants express the same mutually exclusive exons but differ with respect to expression of other alternatively spliced but not mutually exclusive exons. Removal of the mutually exclusive exons *IS4A/IS4B* and/or *I–IIA/I–IIB* allows expression of fewer *cacophony* splice variants. Transcript variants that are possible upon exon excision are marked by +. (**E**) Western blots reveal expression of GFP-tagged cacophony protein (cac$^{sfGFP}$) for all excision variants at the expected band size of ~240 kDa (cacophony) plus ~30 kDa (sfGFP, top), while no band can be detected in the Canton S wildtype (CS, top, right) that does not express cac$^{sfGFP}$. Ten adult brains were used of hemizygous males for all genotypes except for *ΔIS4B* which is homo-/hemizygous lethal. For *ΔIS4B* 20 brains of heterozygous females and a heterozygous cac$^{sfGFP}$ control were used (F1 females from cross with Canton S wildtype flies(+)). β-Actin was used as loading control (bottom). *ΔIS4B* shows weak expression (top, left), while all other exon-out variants express strongly, although *ΔI–IIB* shows somewhat weaker expression (top, middle).

The online version of this article includes the following source data for figure 1:

**Source data 1.** Original 8-bit gray scale image of western blot data of α-actin.

**Source data 2.** Original 8-bit gray scale image of western blot data of α-GFP (representing cacophony$^{sfGFP}$ protein).

**Source data 3.** Original 8-bit gray scale image of western blot data of α-actin with inverted colors.

**Source data 4.** Original 8-bit gray scale image of western blot data of α-GFP (representing cacophony$^{sfGFP}$ protein) with inverted colors.

**Source data 5.** Original 8-bit gray scale image of western blot data of α-actin with inverted colors and linear brightness adjustment.

*Figure 1 continued*

**Source data 6.** Original 8-bit gray scale image of western blot data of α-GFP (representing cacophony[sfGFP] protein) with inverted colors and linear brightness adjustment.

(*I–II*; *Figure 1B*), and is thus predicted to affect binding sites for calcium channel β-subunits (Ca$_\beta$) and G-protein βγ subunits (G$_{\beta\gamma}$). Both alternative *I–II* exons are 117 bp long and encode 39 AAs. The polypeptide encoded by *I–IIA* (first alternative exon of the *I–II* locus) differs from that of *I–IIB* in 23 AAs. Sequence analysis strongly suggests that I–IIB contains both, a Ca$_\beta$- as well as a G$_{\beta\gamma}$-binding site (AID: α-interacting domain) because the binding motif QXXER is present. In mouse Ca$_v$2.1 and Ca$_v$2.2 channels the sequence is QQIER, while in *Drosophila* cacophony I–IIB it is QQLER. In the alternative I–II linker I–IIA, this motif is not present, strongly suggesting that G$_{\beta\gamma}$ subunits cannot interact at the AID. However, as already suggested by *Smith et al., 1998*, also based on sequence analysis, Ca$_\beta$ should still be able to bind, although possibly with a lower affinity. We employ CRISPR–Cas9 to excise one of both alternative exons either at the *IS4* or the *I–II* loci (*Figure 1C*). Genomic removal of *IS4A* results in fly strains that contain only the *IS4B* exon and are named *ΔIS4A* (*Figure 1C*). Accordingly, genomic removal of the *I–IIA* exon results in flies with *I–IIB* only that are named *ΔI–IIA* (*Figure 1C*). To analyze the functions of mutually exclusive exon variants, we create exon-out fly strains with removal of one exon at a time of each of the four exons (*IS4A, IS4B, I–IIA, I–IIB*). Excision of the respective exon is always confirmed by sequencing (see methods). Fly strains with first chromosomes that carry exon excisions are cantonized by 10 generations of backcrossing into Canton S wildtype flies. Exon-out strains are produced from either wildtype (Canton S) or white mutant flies (*w$^{1118}$*, see methods) as well as from fly strains with previously introduced N-terminal fluorophore tagging of the *cac* gene (*Gratz et al., 2019*). Neither on-locus super folder GFP (sfGFP) tagging, nor on-locus tagging of the endogenous cacophony channel with mEOS4b (*Ghelani et al., 2023*) impairs cac localization at presynaptic AZs or synaptic function at the larval *Drosophila* NMJ. In this study, sfGFP-tagged (*Figure 1B*) exon-variants serve analysis of channel localization and mEOS4b-tagged (*Figure 1B*) ones serve to count channel number in AZs, and untagged fly strains serve to control for possible effects of the fluorescent tags.

Of the 18 annotated *cacophony* transcripts in *Drosophila* (Flybase, r6.59), 13 remain upon removal of *IS4A*, 5 upon removal of *IS4B*, 8 upon removal of *I–IIA*, and 10 upon removal of *I–IIB* (*Figure 1D*). For each mutually exclusive exon, we have created multiple exon-out fly strains. Each exon removal induces reproducible phenotypes (see methods). Removal of *IS4B* is embryonic lethal as confirmed in all CRISPR–Cas9 mediated excisions (*n* = 12), both before and after crossing out into a wildtype background. Removal of each of the three other mutually exclusive exons results in viable fly strains, *Drosophila* larvae without any obvious deficits, before and after crossing out into a wildtype background, but with distinct behavioral phenotypes in adult flies (the latter are not further addressed in this study). Western blots with GFP-tagged cac channels test whether the channel protein is expressed in the CNS of all exon-out fly strains (*Figure 1E*). *ΔIS4B* flies are homozygous lethal and are thus used heterozygously over untagged wildtype channels that contain all exons for western blot analysis. Brain homogenate from sfGFP-tagged controls and from all exon-out fly strains with sfGFP-tagged cac channels yield a band at roughly the expected size of 250–270 kDa (*Drosophila* cac isoforms range from 212 to 242 kDa and sfGFP is 27 kDa), whereas no band is detected in control brains without tagged cac (*Figure 1E*, right lane). Protein level in *ΔIS4B* flies (first band from left, *Figure 1E*) is lower as compared to heterozygous controls with all isoforms (second band from left, *Figure 1E*). Similarly, protein level in *ΔI–IIB* flies (fifth band from left, *Figure 1E*) is lower as compared to homozygous controls with all isoforms (third and eighth band from left, *Figure 1E*), thus indicating that cac isoforms containing IS4B and I–IIB are normally abundantly expressed. By contrast, homozygous removal of *IS4A* or of *I–IIA* does not cause obvious differences in expression levels as compared to homozygous sfGFP-tagged controls (*Figure 1E*). Taken together, CRISPR–Cas9 is successfully employed to produce all possible distinct exon excisions at the mutually exclusive splice sites of *Drosophila cacophony*. This now allows for analysis of *cac* exon-specific functions.

We next analyze the functional consequences of the removal of each of the mutually exclusive exons at the *IS4* and at the *I–II* loci for cac channel localization, channel number, and channel function at motoneuron presynaptic terminals of the larval *Drosophila* NMJ.

# IS4B localizes to presynaptic AZs and is required for evoked synaptic transmission

*Drosophila* cac channels interact with the scaffold protein bruchpilot (brp) to localize to presynaptic AZs (*Kittel et al., 2006*; *Ghelani et al., 2023*). Immunohistochemical triple label at the larval *Drosophila* NMJ tests whether either one of the mutually exclusive exons at the *IS4* locus is required for correct presynaptic cac localization. On the level of CLSM, we define correct cac localization in presynaptic AZs at the larval NMJ as cac puncta that overlap with brp puncta but no cac label anywhere else in the presynaptic bouton. Motoneuron axon terminals on larval muscles 6 and 7 (M6/7) are labeled with HRP (*Figure 2A–C*, right column, blue), cac$^{sfGFP}$ by α-GFP immunolabel (*Figure 2A–C*, left column, green), and AZs by α-brp immunohistochemistry (*Figure 2A–C*, second column, magenta). For a better visualization of label in AZs, *Figure 2Ai–Ci* shows selective enlargements of single boutons with all labels. In controls, cac$^{sfGFP}$ channels with full isoform diversity strictly co-localize with brp in presynaptic AZs (*Figure 2A*, cac and brp overlay, third column) as previously reported (*Gratz et al., 2019*). The same is the case for cac channels that contain IS4B but lack IS4A (*ΔIS4A, Figure 2B*). By contrast, cac channels that contain IS4A but lack IS4B (*ΔIS4B*) do not localize to presynaptic AZs (*Figure 2C*). On the level of CLS microscopy we define strict AZ localization as cac puncta clearly overlapping with brp puncta but no cac label anywhere else in the synaptic bouton. Importantly, upon excision of *IS4B*, cac channels are not only absent from AZs, but immunolabel for tagged IS4A channels cannot be detected above background anywhere in the axon terminals (*Figure 2Ci*, see also *Figure 3A*, bottom). Since *ΔIS4B* is homozygous lethal, heterozygous animals are used for triple labels of tagged IS4A channels, the AZ marker brp, and motoneuron terminals on M6/7. Axon terminal shape and brp label in AZs are qualitatively normal, but the label for cac$^{IS4A}$ (=*ΔIS4B*) channels is absent (*Figure 2Ci*). Accordingly, the Pearson's correlation coefficient of IS4A and brp is ~0.2 (*Figure 2D*), which is considered negligible (*Mukaka, 2012*), whereas cac and IS4B (=*ΔIS4A*) channels show a significant Pearson's correlation coefficient as previously reported for controls (*Krick et al., 2021*). In addition, quantification of brp labeling intensity did not show any difference between genotypes, not even in the absence of cac upon excision of IS4B (data not shown). These data indicate that the *IS4B* exon is required for targeting/localizing cac channels within the AZ.

Previous studies showed that presynaptic cac channels are required for normal evoked synaptic transmission at the *Drosophila* NMJ (*Kawasaki et al., 2002*). Reduced cac label in presynaptic AZs goes along with reduced synaptic transmission (*Kittel et al., 2006*), whereas increased cac channel numbers during presynaptic homeostatic plasticity are effective to increase mean quantal content (*Ghelani et al., 2023*; *Medeiros et al., 2024*). If cac presynaptic AZ localization requires *IS4B* but not the *IS4A* exon, removal of *IS4B* but not *IS4A* should impair evoked synaptic transmission. Two electrode voltage clamp (TEVC) recordings from larval muscle 6 in abdominal segment 3 indicate that excitatory postsynaptic currents (EPSCs) as evoked by single presynaptic action potentials are similar in wildtype (Canton S, *Figure 2E, F*, blue) and in animals with sfGFP-tagged cac channels (*Figure 2E, F*, green). Upon removal of *IS4A*, EPSC amplitude is slightly but not significantly increased (*Figure 2E, F*, *ΔIS4A*, red). In transheterozygous animals (*ΔIS4B/ΔIS4A*) with removal of *IS4A* on one chromosome and *IS4B* on the other, EPSC amplitude is significantly reduced (*Figure 2E, F*, black). Spontaneous SV release as characterized by the amplitude (*Figure 2G*) and the frequency (*Figure 2H*) of miniature excitatory postsynaptic currents (mEPSCs) is not affected in *ΔIS4B/ΔIS4A* transheterozygotes. Together, these data indicate that *IS4A* is neither required for AZ localization of channels, nor for evoked synaptic transmission, whereas *IS4B* is required for normal AZ localization and synaptic transmission.

A limitation of these experiments is that homozygous removal of *IS4B* is embryonic lethal, so that presynaptic terminals that are devoid of cac$^{IS4B}$ channels can only be studied in mosaic animals that are heterozygous for *IS4B* excision at most synapses but hemizygous *ΔIS4B* at few synapses of interest. Employing the FlpStop method (see methods, *Fisher et al., 2017*), mosaic animals with motoneurons to muscle 12 (M12) that are *ΔIS4B/cac$^{null}$* can be produced in otherwise heterozygous animals (*ΔIS4B*; *Figure 3A*, bottom row). In control animals, cac$^{sfGFP}$ channels localize to brp positive AZs in motoneuron axon terminals on M12 (*Figure 3A*, top row and selective enlargement). By contrast, cac$^{IS4A}$ do not localize to brp positive AZs in motoneuron axon terminals on M12 (*Figure 3A*, bottom row and selective enlargement). Therefore, *IS4B* is indeed required for cac AZ localization in excitatory glutamatergic axon terminals at the *Drosophila* NMJ. Consequently, evoked synaptic transmission is

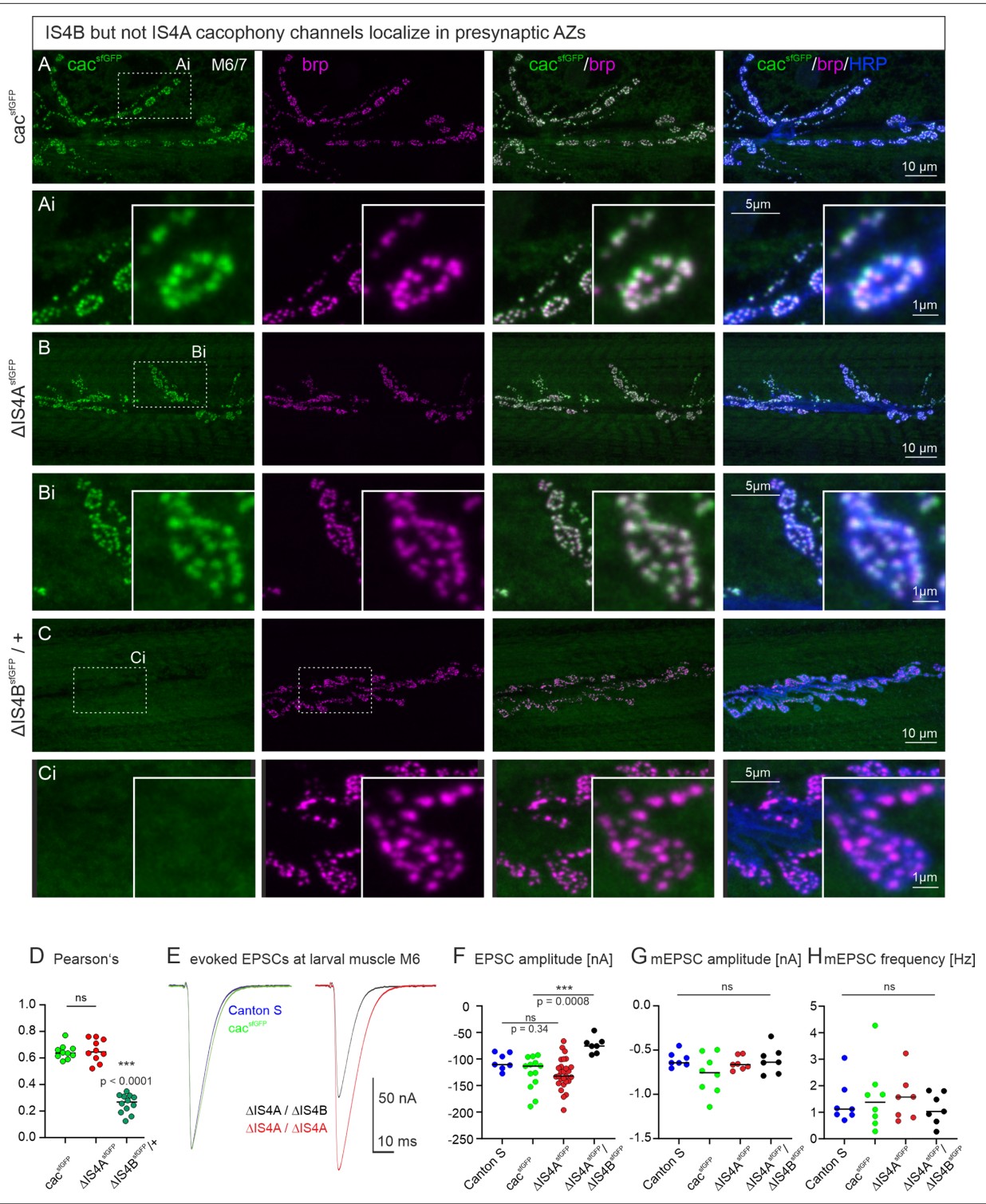

**Figure 2.** The IS4B exon is required for cacophony localization to AZs and for evoked synaptic transmission. (A-C) Representative confocal projection views of triple labels for GFP tagged cac channels (green), the AZ marker brp (magenta), and HRP to label axonal membrane (blue) as well as enlargements of the area marked by dotted rectangles (Ai-Ci). Label was done in control animals with all cac exons (cacsfGFP, top row, A, Ai), in animals with selective excision of either the alternative exon IS4A (ΔIS4AsfGFP, middle rows, B, Bi), or the alternative exon IS4B (ΔIS4BsfGFP, bottom row, C, Ci). Excision of IS4B is embryonic lethal, so that localization analysis was conducted in heterozygous animals (ΔIS4BsfGFP/+). The gross morphology of the neuromuscular junctions (muscle fibers, bouton numbers and sizes, AZ numbers) was similar in all three genotypes. GFP tagged cac channels localize to AZs (A, Ai) as previously reported (*Gratz et al., 2019*; *Krick et al., 2021*). Excision of the IS4A exon does neither impact cac AZ localization nor labeling

*Figure 2 continued on next page*

*Figure 2 continued*

intensity (**B, Bi**). By contrast, upon excision of the IS4B exon, no cac label is detected (**C, Ci**). (**D**) Pearson's correlation analysis of cac and brp in cacsfGFP (green), ΔIS4A (red), and heterozygous ΔIS4B (dark green) animals reveals correlation coefficients of ~0.6 for cacsfGFP and ΔIS4A but not for ΔIS4B (Pearson's correlation coefficient ~0.26, which is significantly different from cacsfGFP control (p<0.001, ANOVA with Dunnett's post hoc test, comparison only against control) and considered negligible. This is in line with absent label in ΔIS4B animals (**C, Ci**). (**E, F**) Representative traces of evoked synaptic transmission as recorded in TEVC from muscle fiber 6 upon extracellular stimulation of the motor nerve from a wildtype control animal (CS, blue), an animal cacsfGFP (green), and animals with cacsfGFP and either IS4A exon excision (ΔISA4, red) or transheterozygous female animals with IS4B excision over IS4A excision (ΔISAB/ΔISA4, black). (**E**) Excitatory postsynaptic currents (EPSCs) are similarly shaped between CS control (blue) and animals expressing cacsfGFP (green), and (**F**) EPSC amplitudes are not statistically different (p=0.34, two sided Tukey's multiple comparison test). In animals with homozygous IS4A exon excision (red), EPSC amplitude is slightly but not significantly increased (p=0.18, two sided Tukey's multiple comparison test). In transheterozygous animals with IS4A excision on one chromosome and IS4B excision on the other one, EPSC amplitude is significantly decreased (***p < 0.001), two sided Tukey's multiple comparison test). (**G**) Quantal size (mEPSC amplitude) and spontaneous release frequency (**H**) show no significant difference between genotypes.

nearly abolished in motoneuron terminals that lack IS4B (*Figure 3B*). Please note that even upon *IS4B* excision, and thus without any cac channels in AZs, the intensity of brp puncta remained unchanged as compared to control (p = 0.81; Mann–Whitney *U*-test). We conclude that *IS4B* is required for evoked synaptic transmission from chemical presynaptic terminals, whereas *IS4A* has no essential function for action potential induced neurotransmitter release from presynaptic terminals.

## The IS4B exon is required for sustained HVA cac-mediated calcium current

Technical constraints prohibit a voltage clamp characterization of IS4B containing cac channels at the larval motoneuron presynaptic terminal, and the somatodendritic calcium current in larval motoneurons is mediated by the *Drosophila* $Ca_v1$ homolog *DmCa1D* (*Worrell and Levine, 2008*; *Kadas et al., 2017*). However, *Drosophila* cac currents have previously been shown in somatodendritic voltage clamp recordings from pupal and adult motoneuron somata (*Ryglewski et al., 2012*; *Ryglewski et al., 2014a*; *Ryglewski et al., 2014b*). Although these recordings are not representative for cac currents at larval presynaptic terminals, they show that *Drosophila* cac channels can in principle contribute to transient and sustained as well as high- (HVA) and low- (LVA) voltage activated currents. Please note that in these neurons most of the somatodendritic LVA and HVA calcium current is mediated by cac plus a small portion of LVA is mediated by Dmca-α1T ($Ca_v3$ homolog) but not by Dmca1D ($Ca_v2$ homolog) (*Ryglewski et al., 2012*). Given that mutually exclusive splicing at the *IS4* site affects the voltage sensor (*Figure 1B*) and that only the *IS4B* exon is required for evoked synaptic transmission, we next tested whether *IS4B* makes a significant contribution to a specific subtype of cac mediated calcium current. The comparison of voltage clamp recordings from adult flight motoneuron somata in animals with full cac isoform diversity and mosaic animals with *IS4B* excision in only these motoneurons (see methods) shows that *IS4B* is required for sustained HVA cac current. Following electrical inactivation of LVA currents with prepulses to −50 mV (*Figure 3C*, top left), HVA with an activation voltage of ~−30 mV shows a sustained component that is reliably recorded with full cac isoform diversity (*Figure 3C*, top left current traces and top IV diagram) but the sustained HVA component is absent in ΔIS4B motoneurons (*Figure 3C*, top right current traces and top IV diagram). By contrast, total cac current (bottom traces) that contains LVA and HVA components is not abolished but only reduced by the excision of *IS4B* (*Figure 3C*, bottom). Somatodendritic calcium current as recorded in neurons without the IS4B exon prove that cac[IS4A] channel isoforms are expressed and functional (*Figure 3C*) although cac[IS4A] isoforms are not present in presynaptic terminals at the NMJ (*Figures 2C and 3A*). The small HVA current that is left upon excision of *IS4B* cannot be mediated by Dmca-α1T as this mediates LVA current (*Ryglewski et al., 2012*). Therefore, the *IS4B* exon is vital for evoked synaptic transmission and promotes sustained HVA calcium current, whereas the *IS4A* exon is not sufficient for evoked synaptic transmission but mediates somatodendritic cacophony current in adult flight motoneurons.

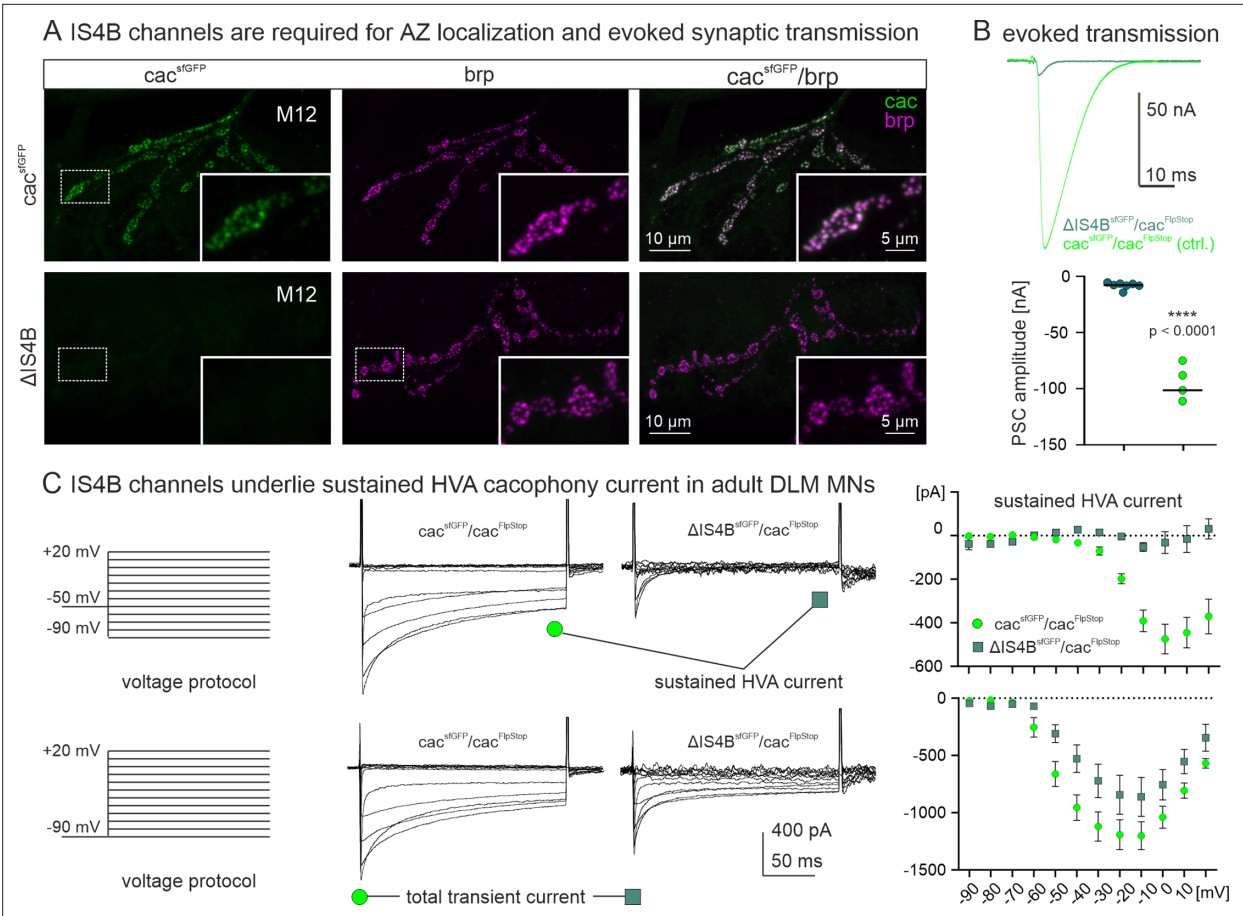

**Figure 3.** The IS4B exon is required for cacophony localization to AZs and for evoked synaptic transmission. (**A, B**) Since animals homozygous for IS4B exon excision are lethal, we created mosaic animals that were heterozygous for cac in most neurons but hemizygous for either cacsfGFP or ΔIS4B in motoneurons innervating muscle M12 (see methods, cacFlpStop). In controls with all cac exons (cacsfGFP, green, top row), cacsfGFP colocalizes with brp (magenta) in presynaptic AZs on M12 (A, enlargements of area indicated by dotted white rectangle in bottom right corner of each image) and evoked synaptic transmission induces EPSCs of about 100 nA amplitude (**B**). By contrast, upon deletion of IS4B (A, bottom row, see also enlargement in bottom right corner) in motoneurons to M12 no cac label is found throughout the motor terminals that are marked by brp (magenta) and evoked synaptic transmission is reduced by more than 90%, Student's T-test, ****p < 0.0001 (**B**). (**C**) Cacophony mediates HVA as well as LVA calcium currents in adult flight motoneurons. All currents are recorded from the somata of adult flight motoneurons in mosaic animals with only one copy of the cac locus in flight motoneurons (see methods). HVA currents (upper traces) are measured by starting from a holding potential of -50 mV (LVA inactivation) followed by step command voltages from -90 mV to +20 mV in 10 mV increments (left, top row), while total cac current (lower traces) is elicited by step commands in 10 mV increments from a holding potential of -90 mV allowing activation of LVA currents (left, bottom row). In GFP-tagged controls (cacsfGFP / cacFlpStop) this reveals transient and sustained HVA current components (middle, top traces). However, following excision of the IS4B exon (ΔISABsfGFP / cacFlpStop), the sustained HVA current is absent. By contrast, upon excision of IS4B, the total cac current that also contains cac LVA currents is only partially decreased (middle, bottom traces). Current-voltage (IV) relation of sustained HVA and for total cac current for controls with all cac exons (cacsfGFP, green circles, n = 7) and following excision of IS4B (ΔIS4BsfGFP, dark green squares, n = 4).

The online version of this article includes the following figure supplement(s) for figure 3:

**Figure supplement 1.** Cacophony[IS4A] channels are sparsely expressed in the larval brain and ventral nerve cord.

## Alternative splicing at the I–II site does not affect AZ localization but channel number and release probability

Immunohistochemical triple label at the larval *Drosophila* NMJ tests whether either one of the mutually exclusive exons at the *I–II* locus is required for correct presynaptic cac localization. Motoneuron axon terminals on larval muscles 6 and 7 (M6/7) are labeled with HRP (*Figure 4A–C*, right column, blue), sfGFP-tagged cac channels by α-GFP immunolabel (*Figure 4A–C*, left column, green), and AZs by α-brp immunocytochemistry (*Figure 4A–C*, second column, magenta). For a better visualization of brp and cac puncta, selective enlargements are shown for each genotype in *Figure 4Ai–Ci*. In

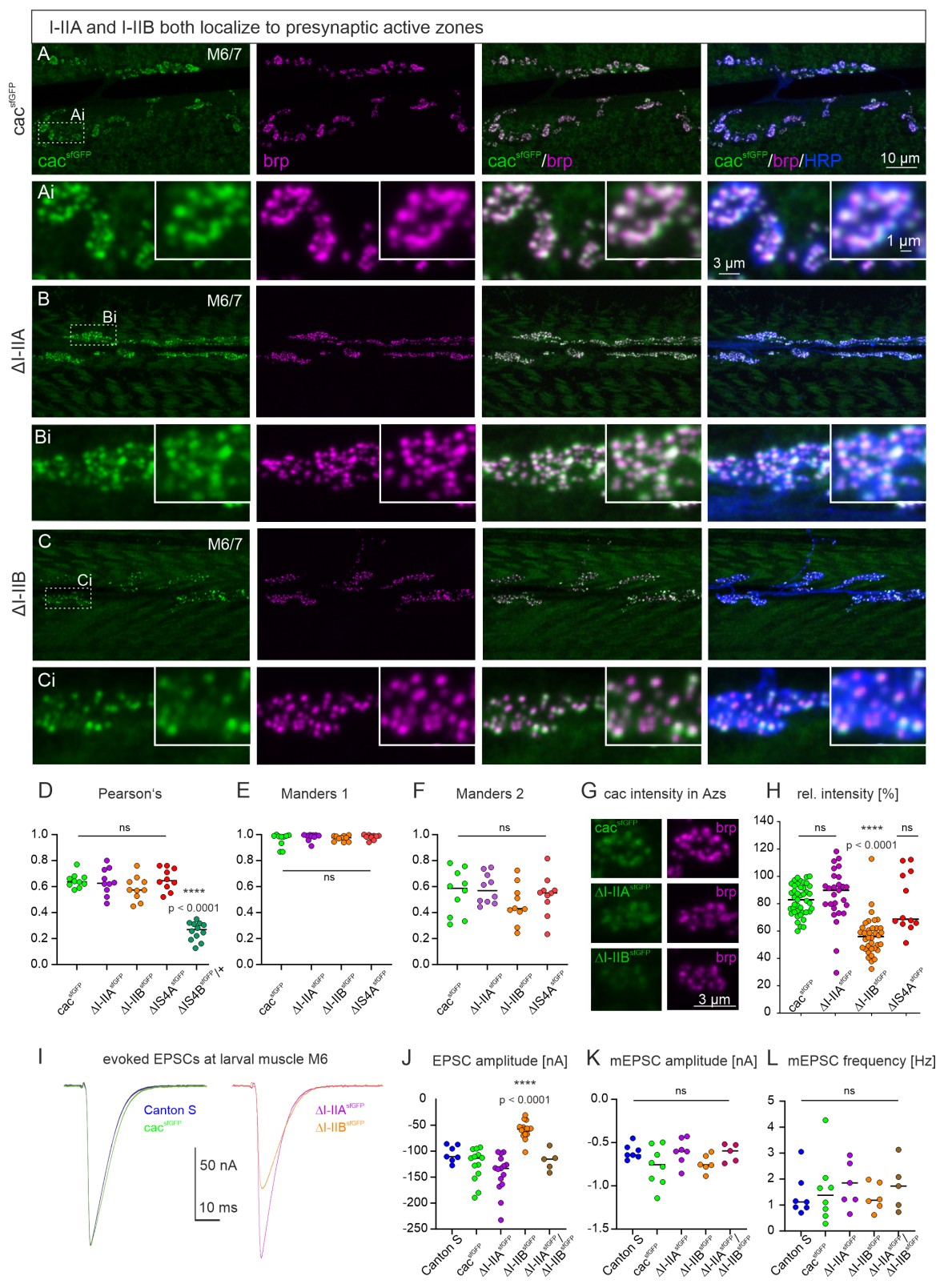

**Figure 4.** Excisions at the I–II exon do not affect active zone (AZ) cacophony localization but can alter cac^sfGFP label intensity in AZs and excitatory postsynaptic current (EPSC) amplitude. (**A-C**) Representative confocal projection views of triple labels for GFP tagged cac channels (green), the AZ marker brp (magenta), and HRP to label axonal membrane (blue) with enlargements that are indicated by white dotted rectangles in A-C (**Ai-Ci**). Label in control animals with all cac exons (cacsfGFP, top two rows, **A, Ai**), with selective excision of either the alternative exon I-IIA (ΔI-IIAsfGFP, middle two

*Figure 4 continued on next page*

*Figure 4 continued*

rows, **B, Bi**), or the alternative exon I-IIB (ΔI-IIBsfGFP, bottom two rows, **C, Ci**). The gross morphology of the neuromuscular junctions (muscle fibers, bouton numbers and sizes, AZ numbers) was similar in all three genotypes (not shown). Excision of the I-IIA exon does neither impact cac AZ localization nor labeling intensity (**B, Bi, H**). Excision of I-IIB does not impact cac AZ localization but labeling intensity seems lower (**C, Ci, H**). (**D-F**) Quantification of cac co-localization with the AZ marker brp yields a similar Pearson's colocalization coefficient (**D**) as well as similar Manders 1 (**E**) and Manders 2 (**F**) coefficients for controls (green) and both exon-out variants of the I-II locus (ΔI-IIA purple, ΔI-IIB orange) as well as for ΔIS4A (red) but not for ΔIS4B (dark green, ANOVA with Dunnett's post hoc test, $p < 0.0001$). (**G**) ΔI-IIBsfGFP shows fainter immunofluorescence signals in the AZ as compared to control (cacsfGFP) and ΔI-IIAsfGFP. (**H**) Quantification confirms a significant reduction in ΔI-IIBsfGFP labeling intensity (Kruskal Wallis ANOVA with Dunn's post hoc test, ****$p < 0.0001$) and no differences between ΔI-IIAsfGFP and control ($p > 0.99$). (**I**) Evoked synaptic transmission as recorded in TEVC from muscle fiber 6 upon extracellular stimulation of the motor nerve. Excitatory postsynaptic currents (EPSCs) are of similar shape and amplitude for CS control (blue) and animals with GFP-tagged cac channels (cacsfGFP, green, $p=0.34$, two sided Tukey's multiple comparison test). Excision of I-IIA (ΔI-IIAsfGFP, purple) has no effect on evoked release amplitude ($p=0.52$, two sided Tukey's multiple comparison test), but excision of I-IIB (ΔI-IIBsfGFP, orange) reduces evoked release significantly ($p < 0.0001$). (**J**) Quantification of EPSC amplitude reveals a highly significant reduction in ΔI-IIBsfGFP (orange) as compared cacsfGFP controls (green, ****$p < 0.0001$), but neither animals with excision of the I-IIA exon (purple), nor transheterozygous animals with excision of I-IIA on one and I-IIB on the chromosome (brown) show differences to control ($p=0.97$, two sided Tukey's multiple comparison test). (**K**) Quantal size (mEPSC amplitude) and spontaneous release frequency (**L**) show no significant difference among genotypes.

The online version of this article includes the following figure supplement(s) for figure 4:

**Figure supplement 1.** GluRIIA intensity across the neuromuscular junction (NMJ) is not altered upon excision of I–IIA or I–IIB.

controls, sfGFP-tagged cac channels with full isoform diversity co-localize with brp in presynaptic AZs (*Figure 4A, Ai*) as also shown above (*Figure 2A, Ai*). The same is the case for both mutually exclusive variants at the *I–II* locus. Cac channels that contain I–IIB but lack I–IIA (Δ*I–IIA*, *Figure 4B, Bi*), and vice versa, cac channels that contain I–IIA but lack I–IIB (Δ*I–IIB*, *Figure 4C, Ci*) show AZ localization (see overlays of brp and cac^sfGFP in *Figure 4B, Bi, C, Ci*, third column). For all cac isoforms that are targeted to presynaptic AZs, quantification reveals similar Pearson's correlation coefficients of ~0.65 (*Figure 4D*), which corresponds to previous reports (*Krick et al., 2021*). Moreover, the Manders 1 and 2 co-localization coefficients are similar for control, cac^IS4B, and both I–II locus isoforms cac^I–IIB and cac^I–IIA (*Figure 4E, F*). In sum, alternative splicing at the *I–II* site does not affect cac expression in presynaptic AZs.

However, removal of the *I–IIB* exon reduces the intensity of cac immunolabel in AZs as compared to control or Δ*I–IIA* highly significantly by ~50% (*Figure 4G, H*). These data indicate that *I–IIB* excision may result in fewer cac channels per presynaptic AZ. We employ TEVC recordings from the postsynaptic cell (M6) to test for the resulting consequences on synaptic transmission (*Figure 4I–L*). The amplitude of the postsynaptic current (EPSC) as evoked by an action potential in the presynaptic motor axon is similar in Canton S with untagged cac and animals expressing cac^sfGFP channels (*Figure 4I, J*, see also above, *Figure 2E, F*). Removal of the *I–IIA* exon (Δ*I–IIA*, *Figure 4I*, purple trace) does not result in significant differences of EPSC amplitude as compared to tagged or untagged controls (*Figure 4I, J*). By contrast, removal of the *I–IIB* exon (Δ*I–IIB*, *Figure 4I*, orange trace) results in a reduction in EPSC amplitude by ~50% (*Figure 4J*), which matches the reduced channel immunofluorescence signal in AZs (*Figure 4H*) upon removal of the *I–IIB* exon. By contrast, neither removal of the *I–IIA* exon nor removal of the *IS4A* exon reduces tagged cac channel intensity in AZs (*Figure 4H*), and neither of these manipulations affect EPSC amplitude (*Figures 2F and 4J*). The differences or similarities between genotypes in EPSC amplitude also hold when analyzing EPSC charge instead of amplitude (significantly smaller EPSC charge in Δ*I–IIB*, $p < 0.0001$, ANOVA with Dunnet post hoc comparison). In contrast to different cac puncta intensities in I–IIA versus I–IIB containing cac channels the intensity of brp puncta remains control like across the exon excision mutants (Kruskal–Wallis ANOVA, $p = 0.1$, data not shown).

It seems unlikely that presynaptic cac channel isoform type affects glutamate receptor types or numbers, because the amplitude of spontaneous mEPSCs (*Figure 4K*) and the labeling intensity of postsynaptic GluRIIA receptors are not significantly different between controls, I–IIA, and I–IIB junctions (see *Figure 4—figure supplement 1*, $p = 0.48$, ordinary one-way ANOVA, mean and SD intensity values are $61.0 \pm 6.9$ (control), $55.8 \pm 8.5$ (Δ*I–IIA*), and $61.1 \pm 17.3$ (Δ*I–IIB*)). However, we cannot exclude altered GluRIIB numbers and have not quantified GluR receptor field sizes. Similarly, the frequency of miniature postsynaptic currents (mEPSCs) remains unaltered (*Figure 4L*). Since mEPSC frequency has been related to RRP size at some synapses (*Pan et al., 2009*; *Ralowicz et al., 2024*), this indicates unaltered RRP size upon *I–IIB* excision, but we have not directly measured RRP size.

In sum, these data show that neither exon at the *I–II* locus is required for cac localization to AZs, but *I–IIB* is required for normal evoked synaptic transmission amplitude. A reduced amplitude of evoked synaptic transmission along with less intensive presynaptic cac immunolabel in Δ*I–IIB* animals (*Figure 4H, J*) is indicative for fewer calcium channels in AZs. Alternatively, the nanoscale localization of cac could be affected by alternative splicing.

We assess the latter by dual color STED microscopy of the AZ marker brp and cac^sfGFP in different exon excision mutants (*Figure 5A–E*). Collecting STED image stacks of synaptic boutons on muscle 6 (*Figure 5A*) reveals numerous AZs in various spatial orientations relative to the focal plane (*Figure 5A–C*). In a strict top view (*Figure 5C1*), the orientation of the synapse is planar so that the central cac cluster (magenta) is surrounded by four brp puncta (green) all in the same plane. If the AZ lies tilted relative to the focal plane, the same arrangement is viewed from different angles (side views, *Figures 5C2–6*). The localization of cac relative to brp in different exon-out variants (*Figure 5D, E*) is quantified by measuring the distance between the center of the cac cluster and the center of the nearest brp punctum in top and side views within the same optical sections (see methods). There is no difference in the cac to brp distance between different views/synapse orientation in relation to the focal plane. In all cac exon-out variants that are expressed in AZs (control, Δ*IS4A*, Δ*I–IIA*, and Δ*I–IIB* but not Δ*IS4B*) the median distance ranges between 103 and 109 nm and reveals no significant differences (Kruskal–Wallis test, p = 0.62) between control (median distance, 106.1 nm) and any of the exon-out variants shown (Δ*IS4A*, Δ*I–IIA*, and Δ*I–IIB*, *Figure 5E*). Alternative splicing at the *I–II* locus does therefore neither affect targeting of cac to AZs (*Figure 4A–F*), nor cac localization within the brp scaffold of the AZ (*Figure 5A–E*).

This leaves changes in cac properties or channel number in AZs as plausible causes for the reduction in evoked synaptic transmission upon removal of the *I–IIB* exon (*Figure 4I, J*). As previously reported, channel number in presynaptic AZs can be estimated by live sptPALM imaging of mEOS4b-tagged cac channels at the NMJ (cac^mEOS4b, *Ghelani et al., 2023*). To estimate channel number in AZs of axon terminals on larval muscle M6 in controls, Δ*I–IIA*, and Δ*I–IIB* (*Figure 5F*), the bleaching behavior of cac^mEOS4b signals in individual AZs is imaged during steady illumination. Discrete bleaching steps (*Figure 5G*, dotted lines) indicate bleaching events of individual cac^mEOS4b molecules, and thus the fluorescence intensity of a single cac^mEOS4b channel. Larger channel numbers produce integer multiple fluorescence intensity amplitudes. Dividing the full fluorescence amplitude that is measured at the illumination onset of all channels in the AZ by the fluorescence intensity from a single channel yields total channel number per AZ. Quantification from three animals per genotype with at least 30 AZs per animal confirms a previous study (*Ghelani et al., 2023*) showing that control animals with full cac channel isoform diversity express ~10 cac channels per AZ (*Figure 5F*, green, 10.8 ± 2 channels). Removing the alternative exon *I–IIA* does not affect channel number per AZ (*Figure 5F*, purple, 10.5 ± 2 channels), but excision of *I–IIB* reduces channel number to ~50% (*Figure 5F*, orange, 5.6 ± 1 channels). A ~50% reduction in channel number counts in AZs (*Figure 5F*) is in line with ~50% reduction in cac immunofluorescence in AZs (*Figure 4H*) and evoked synaptic transmission (*Figure 4I, J*) upon excision of *I–IIB*. In sum, these data indicate that the reduction of evoked synaptic transmission amplitude in Δ*I–IIB* is a consequence of reduced channel number.

## Functional consequences of I–II site alternative splicing during repetitive firing

In addition to reducing the numbers of SVs that are released upon one presynaptic action potential (quantal content), removal of the *I–IIB* exon has significant effects on synaptic transmission during repetitive stimulation and on synaptic plasticity. First, the paired pulse ratio (PPR) is affected. In 0.5 mM calcium, controls with sfGFP-tagged cac channels show slight paired pulse (PP) depression at interpulse interval (IPI) durations below 20 ms (*Figure 6A*). Similarly, upon removal of the *I–IIA* exon (Δ*I–IIA^sfGFP*) PP depression is observed for IPIs below 20 ms (*Figure 6B*). By contrast, upon removal of the *I–IIB* exon, some animals show PP depression and others PP facilitation, so that the average PPR is close to 1 for all IPIs, but the variance is large, in particular for short IPIs (*Figure 6C*). To test whether increased PPRs and increased PPR variance can be fully explained by reduced channel numbers upon excision of *I–IIB*, or whether additional factors were required to explain these findings, we increased the external calcium concentration to 1.8 mM so that the first pulse amplitude in Δ*I–IIB* animals matched that of control animals in 0.5 mM calcium. This fully rescued the effect of *I–IIB*

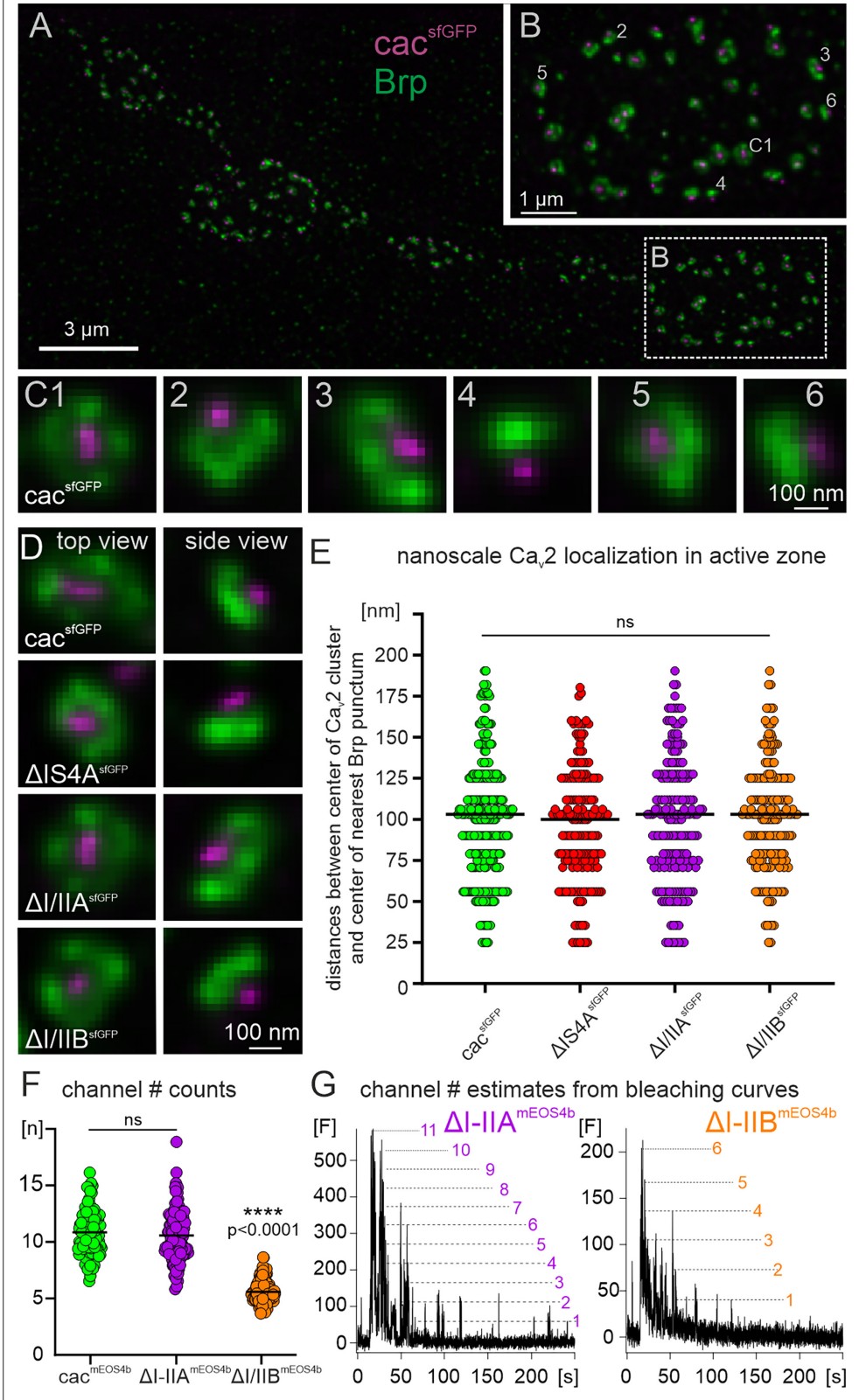

**Figure 5.** Dual color STED imaging reveals equal nanoscale channel localization in AZs of cacophony for all exon-out variants, and live sptPALM imaging reduced channel numbers in AZs for ΔI-IIB. (**A**) Representative intensity projection image of the AZ marker bruchpilot (labeled with anti-brp, green) and cac clusters (cacsfGFP labeled with anti-GFP, magenta) as imaged with dual color STED at motoneuron axon terminal boutons on larval muscle

*Figure 5 continued on next page*

*Figure 5 continued*

M6. The dotted white box demarks one bouton that is enlarged in (**B**). Each cac cluster (magenta) is in close spatial proximity to the AZ marker brp (green) and depending on AZ orientation, with cac and brp is viewed from different angles. Top views (=planar views, see C1 in B and in selective enlargement) show 4 brp puncta that symmetrically surround the central cac cluster. Viewing AZs at the edge of the bouton shows the cac cluster facing to the outside and the brp puncta in close proximity (see 2-6). (**C**) Selective enlargements of each AZ that is numbered in B. (**D**) Top views (left column) and side views (right column) of the cac/brp arrangement in AZs in controls with cacsfGFP (top row), with excision of exon IS4A (ΔISA4sfGFP, second row), with excision of exon I/IIA (ΔI/IIAsfGFP, third row), and with excision of exon I/IIB (ΔI/IIBsfGFP, bottom row). (**E**) Quantification of the distances between the center of each cac punctum to the nearest brp punctum in the same focal plane. (**F-G**) Live sptPALM imaging of mEOS4b tagged cac channels from AZs of MN terminals on muscle 6 in controls with full isoform diversity (cacmEOS4b, green) and following the removal of either I-IIA (ΔI-IIAmEOS4b, purple) or I-IIB (ΔI-IIBmEOS4b, orange). (**F**) Quantification of channel numbers from bleaching curves (**G**) reveals ~ 9-11 cacophony channels per AZ for tagged controls, which matches previous reports (*Ghelani et al., 2023*). Counts for ΔI-IIA reveal no significant differences (Kruskal-Wallis test with Dunn's posthoc comparison, p=0.94), but cac number in AZs is reduced by ~50 % in ΔI-IIB (****p < 0.0001). (**G**) Bleaching curves of single AZs were illuminated after ~10 s and then imaged under constant illumination for another 240 s. Discrete bleaching steps (dotted lines) indicate the bleaching of single cacmEOS4b channels. Comparing the amplitudes of single events and their integer multiples (dotted lines) to the maximum fluorescence at illumination start allows estimates of the total channel number per AZ.

excision. In fact, mean and variance of PPRs as recorded from *ΔI–IIB* in 1.8 mM external calcium were similar to the PPRs as observed in controls and in *ΔI–IIA* in 0.5 mM external calcium across all IPIs (*Figure 6D*). Considering the variance at 0.5 mM external calcium, for control and for excision of *I–IIA* the coefficient of variation is ~5–10% across IPIs whereas upon excision of *I–IIB* it reaches 15–20% for short IPIs (*Figure 6E*). However, increasing external calcium to 1.8 mM in recordings in *ΔI–IIB* animals increases the first EPSC amplitude to that observed in controls in 0.5 mM external calcium (*Figure 6D*) and eliminates altered PPRs as well as increased variance (*Figure 6D, E*).

We next tested whether the time course of synaptic depression during low-frequency repetitive stimulation at 1 Hz is affected upon removal of either *I–II* exon. In 0.5 mM external calcium, repetitive stimulation of the NMJ to muscle M6/7 at 1 Hz frequency for 1 min causes synaptic depression that reaches steady state at about 80% of the initial transmission amplitude with a time constant of about 5 s (*Krick et al., 2021*). The same is observed for cac^sfGFP controls (*Figure 6F1*) and upon removal of *I–IIA* (*Figure 6G1*). Removal of *I–IIB* does not affect the magnitude and time course of depression at 1 Hz stimulation (*Figure 6H*), but it increases the variability of the time constant until steady-state depression is reached (*Figure 6I*). Again, we triturated the external calcium concentration (1.8 mM) so that the first EPSC amplitude in the train in *ΔI–IIB* matched the first one in controls recorded in 0.5 mM external calcium (gray trace in *Figure 6H* is *ΔI–IIB* in 1.8 mM calcium and black trace in 0.5 mM calcium). This did neither affect the mean time constant of depression nor the larger variance observed in *ΔI–IIB* (*Figure 6I*), indicating that these parameters do not depend on calcium influx into AZs. At 10 Hz stimulation neither removal of *I–IIA* nor of *I–IIB* affects the time course or amplitude of synaptic depression (*Figure 6J–M*). Increasing the external calcium concentration (1.8 mM) so that the first EPSC amplitude in the train in *ΔI–IIB* matches the first one in controls recorded in 0.5 mM external calcium (gray trace in *Figure 6L* is *ΔI–IIB* in 1.8 mM calcium and black trace in 0.5 mM calcium) does not affect magnitude or time course of depression (*Figure 6M*). Similarly, increasing external calcium to 1.8 mM in controls does not affect the time constant of synaptic depression, in line with the findings at 1 Hz that these parameters do not depend on the amount of calcium influx into AZs.

Taken together removal of *I–IIA* does not alter channel number and does not affect PPRs or the time course of synaptic depression. By contrast, removal of *I–IIB* reduces channel number. This in turn is the cause for altered PPRs at short IPIs (*Figure 6C, E*) and can be rescued by increasing external calcium. Neither excision of *I–IIA*, nor of *I–IIB* has significant effects on the magnitude or time course of synaptic depression at low stimulation frequencies.

However, motoneuron firing frequencies as previously measured during tethered crawling (*Kadas et al., 2017*) reach ~120 Hz during bursts of about 200 ms duration. Given that the main function of excitatory synaptic transmission to larval muscles is locomotion, it seems important to test stimulation protocols that reflect behaviorally relevant motoneuron firing patterns. Applying motoneuron stimulation for 200 ms duration at either 60 Hz (*Figure 6N*) or 120 Hz frequency (*Figure 6O*) in 0.5 mM

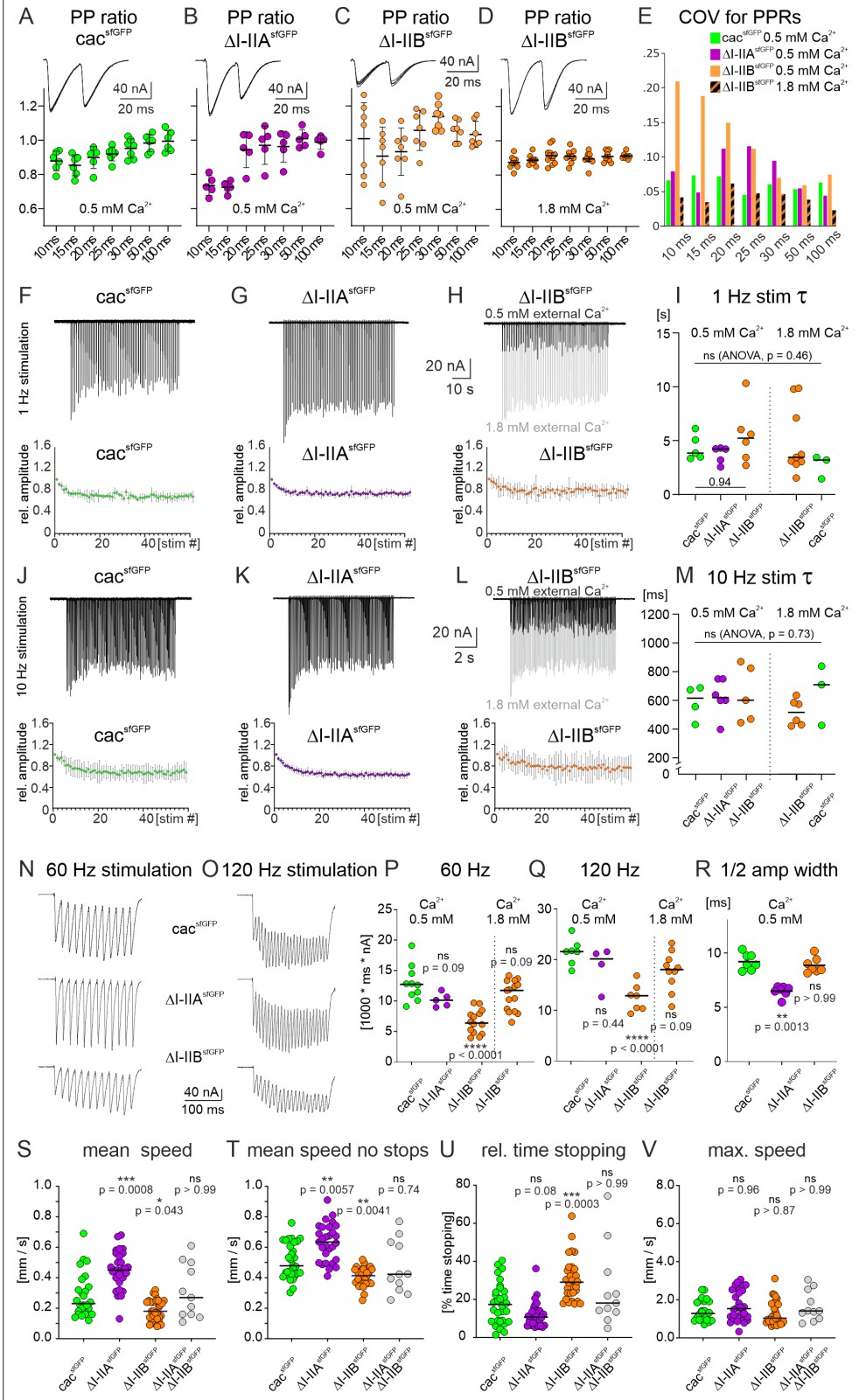

**Figure 6.** Alternative splicing in the I-II linker affects short term plasticity due to decreased calcium influx and motor behavior. (**A-D**) Paired pulse ratio (PPR, ratio of second EPSC amplitude divided by first EPSC amplitude) as measured in 0.5 mM external calcium (**A-C**) or in 1.8 mM external calcium (D) at different interpulse intervals (IPIs ranging from 10 ms to 100 ms) in control animals (cacsfGFP, A), in animals with removal of I-IIA (ΔI-IIAsfGFP, B), and

*Figure 6 continued on next page*

*Figure 6 continued*

in animals with removal of I-IIB (ΔI-IIBsfGFP) either in 0.5 mM calcium (**C**) or 1.8. mM calcium (**D**). The large variance in PPR upon excision of I-IIB (**C**) is rendered control-like if the first EPSC amplitude of the twin pulse is adjusted to 0.5 mM external calcium control level (**D**; comp. with A). This is also reflected in the coefficient of variation (COV) for PPRs (E, 0.5 mM calcium: cacsfGFP green, ΔI-IIA purple, ΔI-IIB orange; 1.8 mM calcium: ΔI-IIB orange/ black pattern). (**F-I**) Synaptic depression as measured in 0.5 mM external calcium in response to stimulus trains of 1 minute duration at 1 Hz frequency for animals with GFP-tagged cac (cacsfGFP, **F**), following removal of I-IIA (ΔI-IIAsfGFP, **G**), and with removal of I-IIB (ΔI-IIBsfGFP, **H**). The top traces show representative TEVC recordings from the postsynaptic muscle cell, and the diagrams mean values (n=5 for F and G, N=6 for H, error bars are SD). (**H**) The light gray trace shows ΔI-IIB in 1.8 mM external calcium, while the black trace shows ΔI-IIB in 0.5 mM calcium. For all 3 genotypes, depression reaches steady state at ~ 80 % of the original EPSC amplitude, but upon excision of I-IIB it is more variable (H). Depression time courses do not differ between genotypes but are more variable in ΔI-IIB, independent of external calcium concentration (**H, I**). (**J-M**) Synaptic depression in response to stimulus trains at 10 Hz frequency for animals with GFP-tagged cac (cacsfGFP, **J**), following removal of I-IIA (ΔI-IIAsfGFP, **K**), and with removal of I-IIB (ΔI-IIBsfGFP, **L**: black in 0.5 mM calcium, gray trace in 1.8 mM calcium). Again, depression is most variable between animals upon excision of I-IIB (**L, M**) but time courses do not differ between genotypes. However, time course variation decreases in 1.8 mM calcium in animals with excision of I-IIB (**M**). Motoneuron stimulation at 60 (**N**) or 120 Hz (**O**) frequency, both for durations of 200 ms in animals with GFP-tagged cac (cacsfGFP, top traces), following removal of I-IIA (ΔI-IIAsfGFP, middle traces), and with removal of I-IIB (ΔI-IIBsfGFP, bottom traces). To compare charge transfer across the NMJ during high frequency bursts the total EPSC area below baseline (prior to stimulation) was measured during each 200 ms burst and plotted for each genotype for 60 Hz stimulation in (**P**) and for 120 Hz stimulation in (Q). Decreased charge transfer in animals with excision of the I-IIB exon is rescued to control level if external calcium is increased to 1.8 mM so that the first EPSP matches control amplitude in 0.5 mM external calcium (**P, Q**, far right data points ΔI-IIB in 1.8 m calcium). (**R**) shows single evoked EPSC half amplitude width. (**S-V**) show different measurements during larval crawling for control animals with GFP-tagged cac (cacsfGFP), removal of I-IIA (ΔI-IIAsfGFP), removal of I-IIB (ΔI-IIBsfGFP), and in transheterozygous animals with removal of I-IIA on one and removal of I-IIB on the other chromosome (ΔI-IIAsfGFP/ΔI-IIBsfGFP). The measured parameters are mean speed during 10 minutes of crawling (**S**), mean speed without any stops (**T**), the relative time spent stopping (**U**) and the maximum speed reached (**V**). In all diagrams each dot demarks a measurement from a different animal and horizontal bars the medians. For statistics, non-parametric Kruskal Wallis ANOVA with planned Dunn's posthoc comparison to control was conducted. ****$p < 0.0001$, ***$p < 0.001$, **$p < 0.01$, *$p < 0.05$.

---

external calcium reveals summation of the EPSCs in control animals with GFP-tagged cac channels (*Figure 6N, O*, top traces) as well as in animals without the *I–IIB* exon (*ΔI-IIB*, *Figure 6N, O*, bottom traces). By contrast, upon excision of *I–IIA*, summation of EPSCs is absent at 60 Hz but present at 120 Hz stimulation frequency (*ΔI–IIA*, *Figure 6N, O*, middle traces). The reason why EPSP summation occurs only at very high motoneuron firing frequencies (120 Hz) in *ΔI–IIA*, but already at 60 Hz in controls and in *ΔI–IIB* animals is a significantly smaller EPSP half-width upon excision of the *I–IIA* exon (*Figure 6R*). This might be an indication for an effect of *I–IIA* exon excision on cac channel biophysical properties (see discussion). At both high-frequency stimulation protocols (60 and 120 Hz) charge transfer is significantly reduced upon excision of *I–IIB* (*Figure 6P, Q*). We measure charge transfer as the total area under the postsynaptic response traces during 200 ms long stimulation bursts at either 60 Hz (*Figure 6N, P*) or 120 Hz (*Figure 6O, Q*). To test whether this is caused by reduced calcium influx during an EPSC in *ΔI–IIB* (*Figure 6N, O*), we again increased the external calcium concentration to 1.8 mM so that the first EPSC of the train matches the first EPSC amplitude observed in controls in 0.5 mM external calcium (see *Figure 6N, O*, bottom gray overlay traces). This increased median total charge in *ΔI–IIB* to control values for both 60 and 120 Hz stimulation (*Figure 6P, Q*), thus indicating that the reduced charge transfer in *ΔI–IIB* at 0.5 mM external calcium can be explained by reduced presynaptic calcium channel number and concomitant reduced calcium influx.

Although crawling speed of *Drosophila* larvae is mainly adjusted by varying the duration between subsequent peristaltic waves of motoneuron bursting (peristaltic wave period duration, *Liu et al., 2023*), differences in cac-mediated neuromuscular synaptic transmission may also play an important role. In accordance with reduced charge transfer to the postsynaptic muscle cell upon excision of the *I–IIB* exon (*Figure 6N–R*), mean crawling speed is significantly reduced in *ΔI–IIB* animals (*Figure 6S*). The mean crawling speed (including stops) of cac$^{sfGFP}$ control larvae is roughly 0.2 mm per second and not significantly affected in transheterozygous *ΔI–IIA* over *ΔI–IIB* larvae that contain full cac isoform

diversity (*Figure 6T, U*). Removal of *I–IIB* (*ΔI–IIB*) causes a significant decrease in mean locomotion speed (*Figure 6S*, orange) whereas removal of *I–IIA* significantly increases speed (*ΔI–IIA*, *Figure 6S*, purple). Alterations in locomotion speed upon alternative cac exon removal could either be caused by changes in mean ground speed (mean speed excluding stops), or by changes in the duration of stopping, or by both. Mean ground speed in controls is about 0.5 mm per second (*Figure 6T*), which is slightly slower but within the range previously reported (0.65–0.8 mm per second, *Wang et al., 1997*; *Guo et al., 2016*). The decreased net locomotion speed in *ΔI–IIB* larvae is caused by significant reductions in the mean ground speed (*Figure 6T*) paired with significant increases in the duration of stopping (*Figure 6U*). By contrast, the net locomotion speed increase as observed in *ΔI–IIA* larvae is caused by significant increases in mean ground speed (*Figure 6T*) without significant changes in the duration of stopping (*Figure 6U*). However, the maximum locomotion speed observed does not differ significantly between controls and any of the test groups (Kruskal–Wallis test, p = 0.14; *Figure 6V*), although *ΔI–IIB* NMJs show significantly reduced charge transfer at 120 Hz motoneuron bursting (*Figure 6Q*). This may indicate a safety plateau of charge transfer at maximum speed.

## I–IIB is required for presynaptic homeostatic plasticity

Chemical synapses are subject to various forms of short-term (*Fioravante and Regehr, 2011*), Hebbian (*Nicoll and Schmitz, 2005*), and homeostatic plastic adjustments (*Turrigiano, 2008*). The *Drosophila* larval NMJ has become a prominent model to analyze the mechanisms underlying presynaptic homeostatic potentiation (PHP, *Frank et al., 2006*; *Davis and Müller, 2015*). PHP is a compensatory increase in the number of SVs that are released upon one presynaptic action potential (quantal content) in response to reduced postsynaptic receptor function. Consequently, EPSC amplitude is restored to its original setpoint despite reduced mEPSC amplitudes. Quantal content can be increased by a larger size of the RRP of SVs, or by elevated release probability ($P_r$). A recent study has shown that the induction of PHP at the *Drosophila* NMJ requires an increase in $P_r$ that is mediated by increasing the number of cac channels in the presynaptic AZ (from ~10 to ~12, *Ghelani et al., 2023*; see also *Gratz et al., 2019*). Our data show that exclusion of *IS4B* impairs cac localization to the AZ (*Figures 2C, Ci, and 3A*), whereas exclusion of *I–IIB* reduces cac number in the presynaptic AZ (*Figure 5F*), raising the question whether homeostatic plasticity is affected by *I–II* exon splicing. Acute pharmacological blockade of postsynaptic glutamate receptors with the bee wolf toxin, philanthotoxin (PhTx), is a well-established means to induce PHP within minutes at the *Drosophila* larval NMJ (*Frank et al., 2006*; *Davis and Müller, 2015*; *Gratz et al., 2019*; *Ghelani et al., 2023*). In control animals with GFP-tagged cac channels as well as upon excision of the *I–IIA* exon (normal channel numbers, *Figure 5F*), bath application of PhTx reliably reduces the amplitude of spontaneously occurring minis (mEPSCs; data are presented as % of baseline without PhTx within genotype: cac$^{sfGFP}$ control: *Figure 7A, E*, green; *ΔI–IIA*: *Figure 7B, E*, purple), but evoked EPSC amplitude is not reduced as compared to control (data are presented as % of baseline without PhTx; cac$^{sfGFP}$ control: *Figure 7A, D*, green; *ΔI–IIA*: *Figure 7B, D*, purple). Normal EPSC amplitudes at reduced mEPSC amplitudes are caused by a compensatory increase in mean quantal content (data are presented as % of baseline without PhTx; *Figure 7F*; cac$^{sfGFP}$ control: green; *ΔI–IIA*: purple). By contrast, PHP is not observed upon removal of the *I–IIB* exon (*Figure 7C–F*). As in controls and in *ΔI–IIA* animals, PhTx application reduces mEPSC amplitude by ~0.2 nA as expected (*Figure 7C, E*), but in *ΔI–IIB* no compensatory increase in mean quantal content is observed (*Figure 7F*), so that EPSC amplitude is not restored to its original setpoint (*Figure 7D*). Similarly, in *ΔI–IIB* animals, PHP is also not possible in response to a permanent reduction of glutamate receptor function in *GluRIIA* mutants, which has been named PHP maintenance and is genetically separable from PHP induction (*James et al., 2019*). In *GluRIIA* mutants, compensatory upregulation of mean quantal content is observed in animals with GFP-tagged cac and upon excision of *I–IIA*, but not following excision of the *I–IIB* exon (*Figure 7G*). Therefore, the lack of the *I–IIB* exon causes fewer cac calcium channels in the AZ and impairs both PHP initiation and maintenance.

Given that synapses that contain only cac isoforms without the *I–IIB* exon show a lower release probability, more variable PPRs and synaptic depression, and an inability for compensatory increases in mean quantal content, the question arises whether normal synapses contain the cac$^{I–IIA}$ channels at all. This can be tested in transheterozygous *ΔI–IIA$^{GFP}$/ΔI–IIB$^{mEOS4b}$* animals, which show normal EPSC amplitudes as well as mEPSC amplitudes and frequencies (*Figure 4J–L*). Co-labeling of AZs with anti-brp (*Figure 7H*, blue) in animals that contain cac channels without I–IIB but with GFP-tagged I–IIA

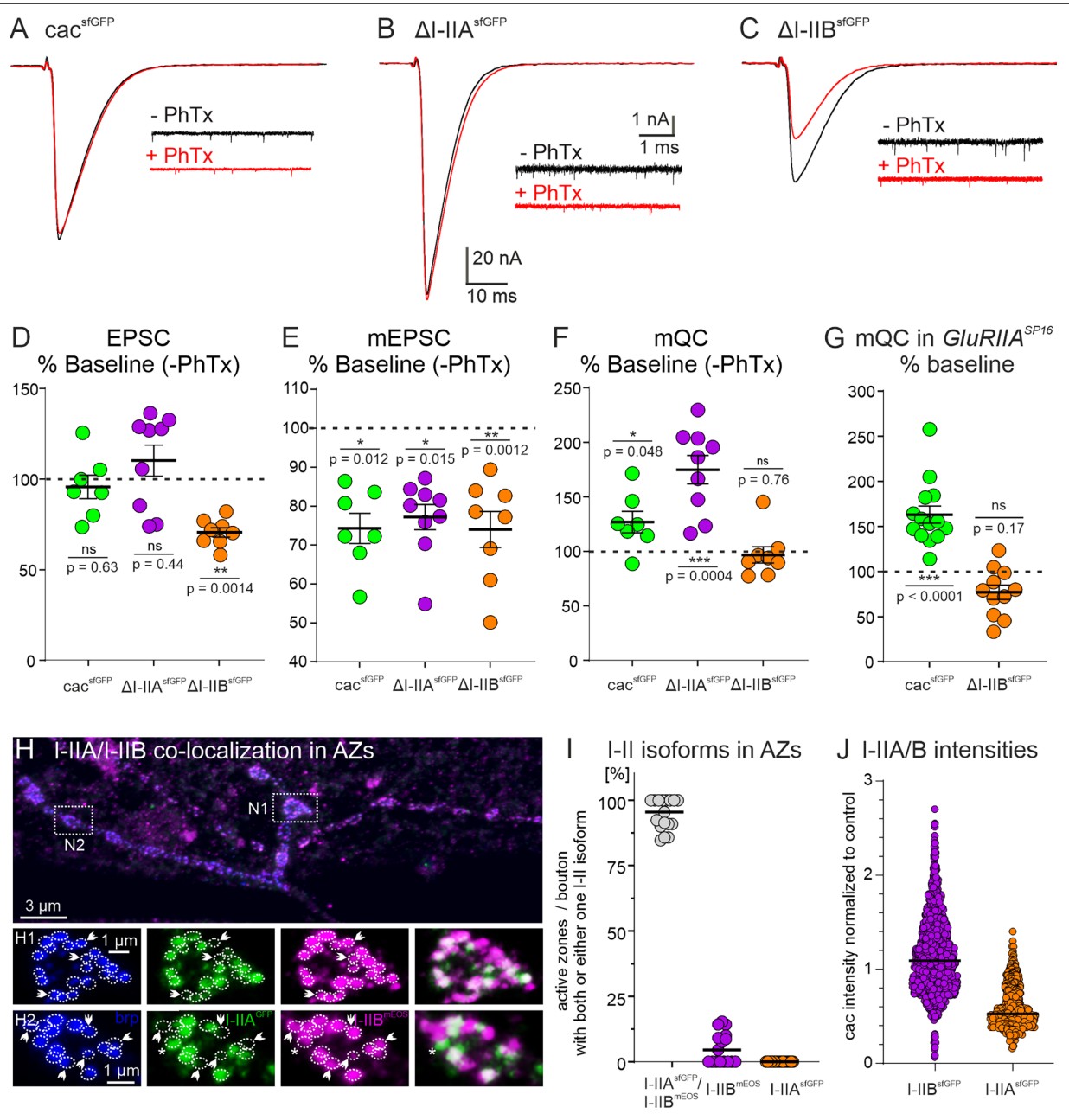

**Figure 7.** Removal of I-IIB impairs presynaptic homeostatic potentiation. (**A-C**) Acute presynaptic homeostatic potentiation (PHP) can be induced in control (cacsfGFP) and ΔI-IIA but not in ΔI-IIB animals by bath application of the glutamate IIA receptor (GluRIIA) blocker philanthotoxin (PhTx) (**A-C**; black traces: without PhTx, red traces after PhTx application). (**D-F**) Quantification of EPSC amplitude, mEPSC amplitude, and mean quantal content (mQC) is shown as % change in PhTx treated animals to untreated control within genotypes (% Baseline (-PhTx). Untreated controls are set to 100 % (dotted line in D-F). EPSC amplitudes in cacsfGFP and ΔI-IIA animals are not significantly affected (D, cacsfGFP, green p=0.63, ΔI-IIA, purple p=0.44), while the EPSC amplitude in ΔI-IIB animals is significantly reduced after PhTx treatment (D, orange, p=0.0014). However, reduction of mEPSC amplitude in all genotypes shows successful block of GluRIIA compared to the respective control (**E**, % change to untreated control within genotype; cacsfGFP, green p=0.012; ΔI-IIA, purple p=0.0015, ΔI-IIB, orange p=0.0012). Accordingly, mean quantal content (mQC) is increased in cacsfGFP and ΔI-IIA animals but not in ΔI-IIB animals (**F**, cacsfGFP, green p=0.048; ΔI-IIA, purple p=0.0004, ΔI-IIB p=0.76). (**G**) PHP maintenance is typically assessed in GluRIIA mutants. PHP maintenance is observed in cacsfGFP animals because mQC is increased in a GluRIIASP16 mutant background (GluRIIASP16; green, p<0.0001). By contrast, in ΔI-IIB animals the GluRIIA mutant background does not cause an increase in mQC (orange p=0.17). All pairwise comparisons in (**D-G**) use unpaired Student's T-test between untreated and treated condition within each genotype. (**H**) In animals that are transheterozygous for the removal of I-IIA and the removal of I-IIB, and carry GFP-tagged I-IIA and mEOS4b-tagged I-IIB cacophony channels (I-IIAsfGFP/I-IIBmEOS), triple immunolabel for the AZ marker brp, GFP, and mEOS4b show that most AZs (blue) but not all (white arrow heads in H1, H2 and asterisk in H2) contain both, GFP-tagged I-IIA (green) and mEOS4b-tagged I-IIB (magenta) channels. Magenta label in the overlay (H1, H2, right column, arrow heads) indicates

*Figure 7 continued on next page*

*Figure 7 continued*

AZs with only I-IIB channels, green label in the overlay (H2, right column, asterisk) indicates one AZ with only I-IIA channels. (**I**) Quantification shows that >95 % of all brp positive AZs contain I-IIA and I-IIB channels (gray dots), few AZs (~5 %) contain only I-IIB (=ΔI-IIA; purple), and almost no AZ contains only I-IIA (=ΔI-IIB; orange). (**J**) Relative fluorescence intensity of sfGFP-tagged I-IIB channels (=ΔI-IIA; purple) and sfGFP-tagged I-IIA channels (=ΔI-IIB; orange). For assessment of I-IIB channels, tagged channels were expressed transheterozygously over untagged I-IIA channels (ΔI-IIAsfGFP/ΔI-IIBno tag; purple) and vice versa (ΔI-IIBsfGFP/ΔI-IIAno tag; orange). Quantification of relative fluorescence intensity compared to transheterozygous control (cacsfGFP/cacno tag) reveals an expression ratio of I-IIB to I-IIA channels of 2:1. \*\*\*p < 0.001, \*\*p < 0.01, \*p < 0.05.

(*Figure 7H*, green) on one chromosome and cac channels without I–IIA but with mEOS4b-tagged I–IIB (*Figure 7H*, magenta) indicates that all AZs contain cac with I–IIB, some AZs contain only I–IIB channel (*Figure 7H1, H2*, white arrow heads), most contain both I–IIB and I–IIA, but very few AZs contain cac channels with only I–IIA (*Figure 7H2*, asterisk). Quantification reveals that ~95% of all AZs contain cac channels with both, the I–IIA and the I–IIB exons, less than 5% of the AZs contain only cac with the I–IIB exon and almost no AZ contains only cac with the I–IIA exon (*Figure 7I*). These data show that the vast majority of presynaptic AZs house a mixture of I–IIA and I–IIB cac. To test whether there is a consistent average ratio of I–IIA and I–IIB channels in presynaptic AZs we measured the cac puncta fluorescence intensities for heterozygous $cac^{sfGFP}/cac$, $cac^{I–IIA\ sfGFP}/cac^{I–IIB}$, and $cac^{I–IIB\ sfGFP}/cac^{I–IIA}$ animals. This way intensity is always measured from cac puncta with the same GFP tag. Normalizing all values to the intensities obtained in AZs from heterozygous $cac^{sfGFP}/cac$ controls reveals a consistent ratio 2:1 in the relative intensities of I–IIB and I–IIA across junctions and animals (*Figure 7J*). This is consistent with the counts in our spt-PALM analysis (see *Figure 5F*) and indicates on average roughly twice as many I–IIB as compared to I–IIA channels across AZs.

## Discussion

The *Drosophila* $Ca_v2$ homolog is named *cacophony* and contains two mutually exclusive splice sites that do not exist in vertebrate $Ca_v2$ VGCC genes. Our data show that alternative splicing at these sites substantially increases *Drosophila* cac functional heterogeneity. We report that the first mutually exclusive exon, that is located in the fourth transmembrane domain of the first homologous repeat (*IS4*), affects cac biophysical properties and is decisive as to whether the channels localize to presynaptic AZs and thus participate in fast synaptic transmission. By contrast, mutually exclusive splicing at the second site encoding the intracellular linker between the first and the second homologous repeat (I–II) does not affect cac presynaptic AZ localization, but instead, fine-tunes multiple different aspects of presynaptic function. In vertebrates, substantial functional synaptic heterogeneity can result from different combinations of $Ca_v2.1$, $Ca_v2.2$, and/or $Ca_v2.3$ in the presynaptic AZ (reviewed in *Zhang et al., 2022*). In *Drosophila*, the collective functions of mammalian $Ca_v2.1$, $Ca_v2.2$, and $Ca_v2.3$ must be portrayed by one $Ca_v2$ gene. Mutually exclusive splicing at the *IS4* and *I–II* sites might have been a different evolutionary strategy to create cac mediated functional synaptic heterogeneity. However, alternative splicing increases functional diversity also in mammalian $Ca_v2$ channels. Although the mutually exclusive splice site in the S4 segment of the first homologous repeat (IS4) is not present in vertebrate $Ca_v$ channels, alternative splicing in the extracellular linker region between S3 and S4 is at a position to potentially change voltage sensor properties (*Bezanilla, 2002*). Alternative splice sites in rat $Ca_v2.1$ exon 24 (homologous repeat III) and in exon 31 (homologous repeat IV) within the S3–S4 loop modulate channel pharmacology, such as differences in the sensitivity of $Ca_v2.1$ to Agatoxin. Alternative splicing is thus a potential cause for the different pharmacological profiles of P- and Q-channels (both $Ca_v2.1$; *Bourinet et al., 1999*). Moreover, the intracellular loop connecting homologous repeats I and II is encoded by three to five exons and provides strong interaction with $G_{βγ}$-subunits (*Herlitze et al., 1997*). In $Ca_v2.1$ channels, binding to $G_{βγ}$-subunits is potentially modulated by alternative splicing of exon 10 (*Bourinet et al., 1999*). Moreover, whole-cell currents of splice forms α1A-a (no Valine at position 421) and α1A-b (with Valine) represent alternative variants for the I–II intracellular loop in rat $Ca_v2.1$ and $Ca_v2.2$ channels. While α1A-a exhibits fast inactivation and more negative activation, α1A-b has delayed inactivation and a positive shift in the IV-curve (*Bourinet et al., 1999*). This is phenotypically similar to what we find for the mutually exclusive exons at the IS4 site, in which IS4B mediates HVA cacophony currents while IS4A channels activate at more negative potentials and show transient current (*Figure 3*; see also *Ryglewski et al., 2012*). Furthermore, altered $Ca_β$

interaction have been shown for splice isoforms in loop I–II (*Bourinet et al., 1999*), similar to what we suspect for the I–II site in cacophony. Finally, in mammalian VGCCs, the C-terminus presents a large splicing hub affecting channel function as well as coupling distance to other proteins. Taken together, $Ca_v2$ channel diversity is greatly enhanced by alternative splicing also in vertebrates, but the specific two mutually exclusive exon pairs investigated here are not present in vertebrate $Ca_v2$ genes. Below, we will discuss the consequences of *Drosophila cac* splicing at these sites for presynaptic function.

## Exon IS4B is required for sustained HVA current and presynaptic AZ localization

Mutually exclusive splicing in the fourth transmembrane domain of the first homologous repeat (*IS4*) yields either *IS4A* or *IS4B*. Our data show that the *IS4B* exon is required for presynaptic AZ localization of cac and for evoked synaptic transmission at a fast glutamatergic synapse, the *Drosophila* NMJ. Accordingly, excision of the *IS4B* exon is embryonic lethal. This is in accordance with the finding that the only cac knock-in construct that has ever been reported to rescue lethality of *Drosophila cac* null mutants contains the *IS4B* exon (*UAS-cac1*; *Kawasaki et al., 2004*). The other alternative exon at the *IS4* site, *IS4A*, is not sufficient to mediate presynaptic AZ localization, does not contribute to evoked synaptic transmission at fast synapses, but it gives rise to functional cacophony channels that localize to other neuron types and neuronal compartments. Therefore, the absence of cac$^{IS4A}$ isoforms from presynaptic AZs at the NMJ is not a consequence of general channel degradation upon excision of *IS4B*, but of exon-specific channel localization. In fact, cac$^{IS4A}$ isoforms are expressed sparsely but in stereotypical patterns in the larval brain and VNC (see *Figure 3—figure supplement 1*). Moreover, in the absence of IS4B isoforms, cac$^{IS4A}$ channels mediate somatodendritic calcium current in adult motoneurons. By contrast, cac$^{IS4A}$ isoforms are not detectable in presynaptic boutons at the larval NMJ. Accordingly, animals with excision of *IS4A* show normal neuromuscular transmission.

Our voltage clamp recordings from motoneuron somata show that splicing in the fourth transmembrane domain, where the voltage sensor is located, affects cac activation voltage. Removing the *IS4B* exon virtually abolishes fast activating, sustained cac mediated HVA current. We infer that *IS4B* containing cac mediate fast activating, sustained HVA current also in presynaptic AZs, although we cannot exclude the possibility that cac$^{IS4B}$ interacts with different accessory calcium channel subunits or with different other proteins to give rise to macroscopically different calcium currents, depending on whether the cac $\alpha_1$-subunit localizes to presynaptic AZs, or to other subcellular compartments. However, fast activating, sustained HVA current is in accordance with the demands of the *Drosophila* larval NMJ that transmits burst of ~200 ms duration with action potential frequencies of ~120 Hz during crawling (*Kadas et al., 2017*). Fast voltage-gated activation of sustained HVA calcium current with fast inactivation upon repolarization is useful at large intraburst firing frequencies without excessive cac inactivation. Our somatic voltage clamp recordings further demonstrate that *IS4B* is not only essential for evoked synaptic transmission, but cac with the *IS4B* exon can also give rise to somatodendritic HVA current. Therefore, the *IS4B* exon is essential for presynaptic function but it can also give rise to cac currents in other neuronal compartments and it is abundantly expressed throughout VNC neuropils. By contrast, the cac$^{IS4A}$ channels are not expressed in NMJ presynaptic terminals, but they show a reproducible sparse expression pattern in some VNC regions, thus likely mediating some distinctly different functions that are yet uncharacterized. An abundant function of IS4B channels in presynaptic AZs of fast chemical synapses is in line with strong protein detection in Westerns from brains, whereas the sparse expression of IS4A in distinct sub-regions of the CNS is in line with a faint band in western blots.

## I–II exon alternative splicing fine-tunes presynaptic function

In contrast to the *IS4* exon, mutually exclusive splicing in the intracellular loop between the first and the second homologous repeats (I–II) does not affect cac localization in AZs at the NMJ. In fact, >95% of all presynaptic AZs contain both, cac isoforms with I–IIA and cac isoforms with I–IIB. Single on-locus alternative exon removal at the *I–II* site allows the study of presynaptic cac channel function with only one of the mutually exclusive I–II exons. Removal of *I–IIA* (*ΔI–IIA*) leaves eight isoforms that contain *I–IIB*. This does not affect cac localization or channel number in presynaptic AZs as compared to control. However, half amplitude width of evoked EPSCs is significantly decreased whereas median amplitude of evoked EPSCs is increased from ~115 to ~130 nA, although the amplitude increase is statistically

just not significant (one sided Mann–Whitney *U*-test, p = 0.078). Together a slight amplitude increase and a decreased half width likely cancel out changes in EPSC charge. Similar channel numbers and localizations but altered EPSC amplitudes and shapes could potentially be caused by different cac properties in the absence of *I–IIA* as compared to control. Specifically, decreased EPSC half width could be caused by significantly faster channel inactivation kinetics, and slightly increased single channel conductance may increase EPSC amplitude. Alternatively, altered EPSC width and amplitude could also be caused by changes in presynaptic action potential shape or postsynaptic glutamate receptor properties or compositions. Although we found no difference in GluRIIA abundance between *ΔI–IIA* and control, we cannot exclude other changes in postsynaptic receptor fields. However, given that *I–IIA* excision primarily affects the relative abundance of I–IIA and I–IIB cac in the presynaptic AZ but not mean channel number, given that postsynaptic VGCCs in muscle are encoded by the *Drosophila* $Ca_v1$ and $Ca_v3$ homologs, and given that cac channels localize to axon terminal AZs, we consider altered properties of different cac isoforms a possible explanation for altered EPSC properties upon *I–IIA* excision. During repetitive firing, the median increase of EPSC amplitude by ~10% is potentially counteracted by the significant decrease in EPSC half amplitude width by ~25 %, so that neither PPRs, nor synaptic depression show significant differences between control and *ΔI–IIA*. Even for stimulus trains that mimic intraburst motoneuron activity as observed during restrained crawling in semi-intact preparations (*Kadas et al., 2017*), there is no difference in charge transfer from the motoneuron axon terminal to the postsynaptic muscle cell between *ΔI–IIA* and control. Surprisingly, crawling is significantly affected by the removal of *I–IIA*, in that the animals show a significantly increased mean crawling speed but no significant change in the number of stops. Given that the presynaptic function at the NMJ is not strongly altered upon *I–IIA* excision, and that *I–IIA* likely mediates also cac functions outside presynaptic AZs (see above) and in other neuron types than motoneurons, and that the muscle calcium current is mediated by $Ca_v1$ and $Ca_v3$, the effects of *I–IIA* excision of increasing crawling speed is unlikely caused by altered pre- or postsynaptic function at the NMJ. We judge it more likely that excision of *I–IIA* has multiple effects on sensory and premotor processing, but identification of these functions is beyond the scope of this study.

Removal of *I–IIB* (*ΔI–IIB*) leaves 10 cac isoforms that contain I–IIA. This does not affect presynaptic AZ localization of cac or the proximity to the AZ scaffold protein brp (*Kittel et al., 2006*), but it significantly reduces $P_r$ by ~50%. This bisection in $P_r$ is correlated with a 50% reduction in the number of cac channels in AZs from about 10 to about 5, as determined by tagged cac fluorescence intensity measurement in fixed specimen and by live sptPALM counting of $cac^{mEOS4b}$ (*Heck et al., 2019*; *Ghelani et al., 2023*). These data suggest that $P_r$ is nearly linearly related to the number of VGCCs in the presynaptic AZ, at least at 0.5 mM external calcium which we mainly used for this study. Please note that the relation between external calcium and $P_r$ at the *Drosophila* NMJ has been reported non-linear at external calcium concentrations above 1.5 mM, but linear between 0.5 and 1.5 mM (*Weyhersmüller et al., 2011*). Accordingly, at 1.5 mM external calcium a linear correlation between $P_r$ and tagged cac channel fluorescence intensity has recently been reported (*Medeiros et al., 2024*). This also corresponds to observations at the mammalian calyx of Held, where the number of AZ VGCCs correlates linearly with $P_r$, and moreover, influences whether subsequent release is depressed or facilitated (*Sheng et al., 2012*). Similarly, in synapses with fewer cac channels upon *I–IIB* excision, we find a significant decrease in $P_r$ along with a significant reduction in PPR. The effects on PPR can by fully attributed to reduced channel numbers in presynaptic AZs upon *I–IIB* excision, because they can be fully rescued by triturating the external calcium concentration so that the first pulse amplitude matches that of controls with normal $P_r$. Therefore, regulation of VGCC number in presynaptic AZs may be a conserved mechanism to tune $P_r$ and short-term plasticity from flies to mammals.

At the *Drosophila* NMJ a steep gradient of synaptic transmission amplitude exists along the motoneuron axons over their target muscle fibers, with the highest presynaptic $Ca^{2+}$ influx in most distal presynaptic sites along the axon. This has been interpreted as gradient control of $P_r$ along the axon of the same neuron (*Guerrero et al., 2005*), which is at least in part regulated by the balance of different release enhancing and suppressing proteins at proximal versus distal release sites, such as complexin (*Newman et al., 2022*). Another potential means to regulate $P_r$ at different release sites of the same neuron could be uneven ratios of I–IIA and I–IIB cac isoforms, which requires additional analysis. The prediction would be that the more distal the release site the more I–IIB channels are expressed. Given that these have a highly conserved $Ca_β$-binding motif, targeting to distal release sites might

be promoted in comparison to I–IIA channels with a less conserved $Ca_\beta$-binding motif (**Smith et al., 1998**).

The reduction in AZ cac number upon removal of *I–IIB* has three additional important functional consequences. First, reduced $P_r$ and charge transfer across the NMJ during crawling-like motoneuron bursting patterns are reflected in a significant decrease in mean crawling speed and a highly significant increase in the number of stops, whereas maximum crawling speed is not affected. Unaffected maximum speed at significantly reduced $P_r$ indicates that the charge transfer at high motoneuron intrabust firing frequencies is above the one needed for maximum muscle contraction. Moreover, mean speed is significantly but just slightly decreased upon removal of *I–IIB* and a highly significant reduction in $P_r$, whereas the number of stops is increased highly significantly by ~50%. It seems likely that large effects on the continuation of motor behavior but only mild effects on the speed are caused not only by effects on synaptic transmission at the NMJ but also by effects on other neuronal compartments or on other neurons in sensory or premotor circuitry. This interpretation would be in accordance with the finding that cac currents are also measured from the somatodendritic domain of pupal and adult *Drosophila* (this study) and that larval *Drosophila* motoneuron excitability as measured by I/F relationships are altered upon *cac*-RNAi (**Worrell and Levine, 2008**). Second, removal of *I–IIB* increases the variability of PPR and the time course of synaptic depression, but this is solely due to reduced channel numbers upon *I–II* excision, because the effect is rescued by increasing the external calcium concentration. It seems plausible that mechanism with probabilistic features, such as cac activation/inactivation/de-inactivation as well as $Ca_v2$ channel mobility in AZs (**Heck et al., 2019**; **Ghelani et al., 2023**) exert a higher impact with fewer channels. Therefore, increasing variability of presynaptic function might be another consequence of reduced AZ cac number upon *I–IIB* excision. And third, removal of *I–IIB* abolished the ability of both initiation and maintenance of PHP, which in turn, requires an upregulation of the number of cac and of brp molecules in the presynaptic AZ (**Ghelani et al., 2023**; **Gratz et al., 2019**). Similarly, PHP is also blocked in cac hypomorphic mutants which also reduce EPSC amplitude, likely due to reduced calcium influx (**Frank et al., 2006**), thus indicating that fast PHP induction might not be possible with reduced calcium conductance in AZs. Similarly, increased calcium influx into the presynaptic AZ after induction of PHP (**Müller and Davis, 2012**; **Davis and Müller, 2015**) fits with an increase in cac channel number. Increasing the number of cac$^{I–IIA}$ in AZs as a compensatory response to reduced postsynaptic receptor function seems not likely with reduced $Ca_\beta$-binding affinity, or in the face of fewer channels to start with, or for additional reasons, which will require additional studies.

In summary, our study suggests mutually exclusive splicing at two *cac* splice sites, that do not exist in mammals, as an alternative strategy to increase functional heterogeneity at the presynaptic AZ of fast chemical synapses, so that the *Drosophila* $Ca_v2$ homolog *cacophony* may portrait some of the functional heterogeneity that arises in mammals from the combinatorial usage of $Ca_v2.1$, $Ca_v2.2$, and $Ca_v2.3$. Splicing at the first *Drosophila* mutually exclusive site directs cac to different subcellular compartments and tunes cac biophysical properties. Splicing at the second mutually exclusive site does not direct cac to different subcellular compartments, but it fine tunes multiple aspects of presynaptic function by changing presynaptic AZ channel numbers or ratios between cac splice variants.

**Table 1.** gRNA sequences used for cas9 target site.
Vertical lines depict intended break points. Bold and underlined nucleotides indicated PAM (protospacer adjacent motif) sequences.

| Target exon | Pre-exon | Post-exon |
|---|---|---|
| IS4A | **cca**atatcacctatg\|tttaagtc | gttgctac\|taccaaaaacct**agg** |
| IS4B | **ccg**aaactatagagt\|gactgacc | gtggcaca\|tgattgtcgtgg**agg** |
| I–IIA | gctacgtg\|catgtgcataca**cgg** | **cca**agtaaactgcca\|taatcaac |
| I–IIB | **cca**gtaaatgatttc\|gactttct | **ccc**ccatatgttcat\|ccacatcc |

**Table 2.** Cycler settings for exon-out verification.

| Step | Temperature | | Duration |
|------|-------------|---|----------|
| 1 | 95°C | | 3:00 min |
| 2 | 95°C | | 0:30 min |
| 3 | 57°C | −1°C per cycle | 0:45 min |
| 4 | 68°C | | 1:30 min |
| 5 | Go to step 2 | 10× | |
| 6 | 95°C | | 0:30 min |
| 7 | 52°C | | 0:45 min |
| 8 | 68°C | | 1:30 min |
| 9 | Go to step 6 | 24× | |
| 10 | 68°C | | 5:00 min |
| 11 | 4°C | | ∞ |

# Methods

## Generation of CRISPR flies

Exon excision was performed via the CRISPR/Cas9 method (*Doudna and Charpentier, 2014*; *Sternberg and Doudna, 2015*). Cacophony is located on the X-chromosome in *Drosophila*. Cac$^{sfGFP}$ exon-out flies were generated by crossing female virgin flies expressing super folder GFP (sfGFP)-tagged cac (C cac$^{sfGFP}$, *Gratz et al., 2019*) channels along with the Cas9 enzyme under the control of the germ line active *nanos*-promoter (*nos-cas9*, Bloomington stock center #78781) to male flies expressing a *gRNA* transgene under the control of the germ line active *U6*-promoter. The *gRNA* sequences were designed such to specifically target sequences flanking the exon to be excised (see *Table 1*). CRISPR events take place as soon as both *nos-cas9* as well as *U6-gRNA* transgenes are present in the same fly, no matter whether these flies are male or female. For simplicity, for excision of exons *IS4A*, *I–IIA*, and *I–IIB*, male progeny were collected as such flies are hemizygous vital, whereas for excision of *IS4B*, females were collected because removal of *IS4B* is lethal. Flies were then back-crossed into suitable balancer strains to keep the X-chromosome with the putative cac exon excision and to be able to follow out-crossing of *nos-cas9* or *U6-gRNA* transgenes. Successful exon excision was confirmed with single fly genomic PCR with suitable primers and Taq DNA polymerase (New England Biolabs, #M0267S) (for cycler (Bio-Rad T100 thermo cycler) settings and primers, see *Tables 2 and 3*) and subsequent 1% agarose gel electrophoresis. Gene sequence around the excision was confirmed with next generation sequencing (StarSeq, University of Mainz Campus with the exon-out verification primers, see *Table 3*). Gels were run at 100 V for 45 min. Primers were obtained from Integrated DNA Technologies, Germany. To minimize contamination of exon-out fly stocks, successful excision mutants were cantonized for at least five generations to enhance the likelihood of outcrossing of undesired mutations due to CRISPR off-target events. To clean up the X-chromosome itself on which the desired CRISPR event took place, flies were subjected to recombination with Canton S flies. Lack of the desired exon was then re-confirmed by PCR again.

## Generation of gRNA transgenic flies

First, the eligibility for Cas9 cleavage was assessed, core properties are the protospacer adjacent motif NGG flanking the 20 bp *gRNA* (see *Table 1*) to facilitate site recognition, and a 5' G as the vector used for *gRNA* expression later used the *U6*-promoter. Moreover, CRISPR sites must not disrupt recognition sites for the splicing machinery. For this, online tools were used (CRISPR target finder: *Gratz et al., 2014*, http://targetfinder.flycrispr.neuro.brown.edu/index.php and Cas9 target finder: https://shigen.nig.ac.jp/fly/nigfly/cas9/cas9TargetFinder.jsp). Sites for double strand breaks were picked at a distance from the intron/exon borders to not affect splice acceptor sites. Furthermore, sites known

**Table 3.** Exon-out verification primers.

| Excised exon | Forward primer | Reverse primer | Expected band size with/without exon |
|--------------|----------------|----------------|--------------------------------------|
| IS4A | CACGCCGTGCAAGCATTATC | CCTAAATTCGTGGCTACCAC | 988/482 bp |
| IS4B | CTGTGTGTGATTCTCGCGAC | CCGTTCCGATTCGATCCAG | 1305/340 bp |
| I–IIA | GCCACTTCTACACCAGTTCAC | GAACTCTTTGTAGCCGGG | 951/571 bp |
| I–IIB | CTCATTGAAGGTTCCCGTC | GGAGAGGGATGCTAAGAATG | 954/446 bp |

to impact splice efficiency (*Blanchette et al., 2005*; *Brooks et al., 2011*) were avoided. In addition, sites for minimal loss of total genomic region were determined and the *gRNA* sequences with as few as possible predicted off-target cleavage sites were selected.

## Fly rearing

Flies were kept at 25°C in 25 mm diameter plastic vials with mite proof foam stoppers on a cornmeal, yeast, agar, glucose diet on a 12-hr light/dark regimen. Wandering L3 larvae were collected directly from food vials. Two-day-old adult flies were collected from their food vials and placed on ice in pre-chilled empty food vials for no more than 1 min prior to dissection (for patch clamp recordings).

## Flies

For electrophysiological recordings as well as for immunohistochemical labeling, sfGFP-tagged cac flies were used. First, validity of sfGFP-tagged cac channel flies was confirmed by comparing $w^+$ TI{TI} cac$^{sfGFP-N}$ flies with the wildtype strain Canton special (Canton S). After that, the control strain for all sfGFP-tagged exon-out flies was $w^+$ TI{TI}cac$^{sfGFP-N}$.

Exon-out flies were $w^+$ TI{TI}cac$^{sfGFP-N\ \Delta IS4A}$, $w^+$ TI{TI}cac$^{sfGFP-N\ \Delta IS4B}$/FM7c P{2x Tb-RFP}, $w^+$ TI{TI}cac$^{sfGFP-N\ \Delta I-IIA}$, and $w^+$ TI{TI}cac$^{sfGFP-N\ \Delta I-IIB}$. All CRISPR fly strains were originally white mutant. To reduce the likelihood of accumulation of off-target effects resulting from CRISPR events, we replaced the mutant white gene on the X-chromosome (on which *cacophony* is also located) by a Canton S-derived wildtype white gene from our Canton S lab stock that we also used as control (see above). In addition, after CRISPR events, flies were crossed to remove nos-Cas9 and gRNA transgenes on the second chromosome and replaced these chromosomes by ones without transgenes or other known mutations. This first removed transgenes that were needed for CRISPR induction and second, this also removed possible off-target events on the second chromosome. In the process, third and fourth chromosomes were replaced automatically as well. We back-crossed with Canton S strains for at least five generations. Thus, off-target chromosomal aberrations were minimized by recombination on the X-chromosome and by replacement of all other chromosomes by out-crossing.

For channel counting and for assessment of cac$^{I-IIA}$ and cac$^{I-IIB}$ expression in AZs (both see below), we used exon-out flies that carried mEOS4b endogenously directly after the start codon of *cacophony* (*Ghelani et al., 2023*). mEOS4b-tagged exon-out fly strains were: $w^*$ TI{TI}cac$^{mEOS4b-N}$, $w^*$ TI{TI} cac$^{mEOS4b-N\ \Delta I/IIA}$, $w^*$ TI{TI}cac$^{mEOS4b-N\ \Delta I/IIB}$. All mEOS4b-tagged fly strains were white mutant.

For GluRIIA immunohistochemistry, transgenic flies expressing GluRIIA$^{GFP}$ under a native promoter were used (*Rasse et al., 2005*).

## Western blots

For proof of protein expression after exon excision, western blots were performed with detection of the N-terminal sfGFP tag (due to lack of cacophony antibodies). As loading control, β-actin was used. Cacophony bands were expected above 250 kDa (cacophony plus GFP) and actin was expected at 42 kDa. For viable exon excision mutants (cac$^{sfGFP}$ as control, cac$^{sfGFP\ \Delta IS4A}$, cac$^{sfGFP\ \Delta I-IIA}$, and cac$^{sfGFP\ \Delta I-IIB}$), 10 adult male brains per lane were prepared, for lethal exon excisions (Canton S as control, cac$^{sfGFP}$/+, and cac$^{sfGFP\ \Delta IS4B}$/+), 20 adult female brains of heterozygous animals were used. Brains were dissected with forceps one by one and collected in 23 µl squishing buffer (composition below) in 0.5 ml low bind sample tubes on ice. When 10 or 20 brains, respectively, were collected, samples were squished with a sterile pestle using a motorized squishing device. After squishing, 3 µl sample buffer (composition below) were added to the homogenized samples, which were subsequently boiled at 95°C for 5 min. Samples were then stored at −30°C until use. For loading, samples were taken out of the freezer, boiled for 3 min at 95°C and directly loaded into a 10-well Mini-Protean TGX Precast Gel with a 4–15% polyacrylamide gel gradient with 50 µl volume per well (Bio-Rad, Cat# 456-1084). The gel was run in running buffer (composition below) for 2 hr 40 min at 80 V. As protein marker, 8 µl Spectra Multicolor High Range Ladder (Thermo Scientific, Cat# 26625) was used. Transfer was done in transfer buffer (composition below) on nitrocellulose membrane overnight at 35 V with a cool block and the setup sitting in an ice box. Next day, the nitrocellulose membrane was washed with TBS-Tween20 and then subjected to blocking with Intercept T20 Antibody Diluent (LI-COR, Cat# 927-65001) diluted 1:1 with 0.1 M PBS for 2 hr. Then the nitrocellulose membrane was cut horizontally to subject the pieces with the expected cacophony bands and the expected loading control (β-actin) separately to antibody

labeling. This was followed by primary antibody incubation with 1:500 polyclonal rabbit anti-GFP antibody (Thermo Fisher Scientific, Cat# A11122) or 1:10,000 monoclonal mouse anti-actin (DSHB, JLA20) diluted in Intercept T20 Antibody Diluent (see above), first for 2 hr at room temperature and then for two nights at 4°C. Then antibody solution was removed and membrane washed 3 × 15 min with TBS-Tween20. This was followed by incubation with secondary antibodies diluted in Intercept T20 Antibody Diluent for 2 hr at room temperature in the dark with IRDye 680 donkey anti-rabbit (LI-COR, Cat# 926-68073) at 1:10,000 to detect GFP-tagged cacophony bands, or IRDye 800 donkey anti-mouse 1:10,000 to detect β-actin, respectively. This was followed by 2 × 15 min washes with TBS-Tween-20 and then 1 × 15 min with 0.1 M PBS to reduce background. Bands were detected with an LI-COR Odyssey Fc Imaging System with 30 s or 2 min exposure times.

## Western blot solutions

Lyse buffer: 25 ml 4× Tris–HCl/SDS pH 6.8, 20 ml glycerol, 4 g SDS, 1 mg bromophenol blue.

6× sample buffer (10 ml): 0.3 M Tris–HCl pH 6.8, 8% SDS, 50% glycerol, 0.25% bromophenol blue. 8% mercapto-ethanol was added fresh directly before use.

10× SDS running buffer: 30 g Tris Base (pH 8.3), 150 g glycine, 10 g SDS. Add ddH$_2$O to 1000 ml total volume.

Transfer buffer: 15.5 g Tris Base, 72 g glycine, 1000 ml methanol, 5000 ml ddH$_2$O.

## Dissection for electrophysiology and immunohistochemical labeling

Third instar larvae were dissected in HL3.1 saline (composition below). Larvae were fixed dorsal side up to a Sylgard (Sylgard 184, DowCorning) coated lid of a 35-mm Falcon dish with two insect minuten pins through the mouth hook and through the tail. After covering the larva with saline (composition below), the body wall was cut along the dorsal midline. To obtain a filet preparation, the body wall was spread laterally and fixed with two minuten pins on each side. Gut and trachea were removed as well as the ventral nerve cord. For electrophysiology from the larval NMJ (see Electrophysiology), the motor nerves were left long, for immunohistochemical labeling (see Immunohistochemistry), the motor nerves were cut very short.

Adult flies were dissected in normal saline (composition below – but calcium current recordings were performed with different external solution – composition and electrophysiology see below). Adult 2-day-old female flies were anesthetized briefly on ice in pre-chilled empty fly rearing vials. Then legs and wings were removed and the fly fixed dorsal side up in a Sylgard coated lid of a 35-mm Falcon dish with two insect minuten pins through the head and through the tip of the abdomen. After covering the fly with saline (composition below), the cuticle was cut along the dorsal midline. To expose the ventral nerve cord with the motoneurons, the thoracic cuticle with the cut wing depressor muscle (DLM) was spread laterally and pinned down with one minuten pin on either side. After removal of gut, esophagus, and salivary glands, the ventral nerve cord was exposed.

## Electrophysiology

All electrophysiological recordings were carried out at room temperature (~22°C). For TEVC recording microelectrodes were pulled with a Sutter Flaming Brown P97 microelectrode puller from borosilicate glass capillaries with filament, with inner diameter of 0.5 mm, and an outer diameter of 1 mm (Sutter BF100-50-10 or World Precision Instruments 1B100F-4). Patch pipettes were pulled with a Narishige PC10 electrode puller from borosilicate glass capillaries without filament, with an inner diameter of 1 mm, and an outer diameter of 1.5 mm (World Precision Instruments PG52151-4).

## TEVC of muscle 6 or 12 in the larval body wall

For TEVC recordings, dissected third instar larvae (see dissection) were placed on an upright Olympus BX51WI fixed stage microscope with a 20× water dipping lens (Zeiss LD A-Plan). The motor nerves of either segment A2 or A3 on the left or the right side was sucked into a glass microelectrode filled with saline with an individually broken tip. After placing the recording electrodes filled with 3 M potassium chloride (KCl) and the ground wire (chlorinated silver) into the recording solution (HL3.1 saline, composition see below), both electrodes were nulled against the ground electrode (bridge mode) using an Axoclamp 2B intracellular amplifier (Molecular Devices). Then the muscle (M6 or M12) cell was first impaled with the larger tip current passing glass microelectrode (~8–13 MΩ) directly followed

at a small distance by the smaller tip recording electrode (~20–30 MΩ). If both electrodes recorded a membrane potential of at least −50 mV without differing by more than 3 mV, RMP balance was adjusted and then TEVC established by pressing the respective knob on the amplifier. Gain was set between 8 and 25. Anti-alias filter was set to ½ the sampling rate to minimize noise. Then the desired command potential was set, mostly at −70 mV. Holding current needed to keep the membrane potential at the desired −70 mV had to be below 4 nA, otherwise recordings were discarded. Data were digitized at 50 kHz (Digidata 1550B, Molecular Devices) and recorded with pClamp 11.1.0.23 software package (Molecular Devices). Data were filtered offline with a lowpass Gaussian Filter with a −3 dB cutoff frequency of 360 Hz.

The suction electrode with the motor nerve was connected to a pulse generator (A-M Systems Model 2100) to deliver low voltage (up to 10 V) 0.1 ms duration stimulations at varying patterns. Stimulation voltage was adjusted slowly to be sure that both motor axons innervating the recorded muscles were stimulated. This was determined unequivocally by the transmission amplitude, which is distinctly smaller if just one motor axon is stimulated as compared to stimulation of both motor axons.

For recordings of mEPSCs, spontaneous occurrence of mEPSCs was recorded for 1 min without stimulation. For single evoked EPSCs, the motor nerve was stimulated at 0.1 Hz frequency for 50 s and the amplitude was recorded. Mean quantal content was determined by dividing the mean amplitude of several EPSCs of one recording by the mean amplitude of several mEPSC of the same recording. For short-term plasticity, PPs were recorded by stimulation of the motor nerve twice with varying interspike intervals (IPIs) of 10, 15, 20, 25, 30, 50, and 100 ms (applied in that order with intersweep intervals of 10 s). PPRs were calculated by dividing the amplitude of the second pulse by that of the first pulse. Mean PPR values are based on the PPRs of each sweep and then averaged and also by first averaging the amplitudes of all first and all second pulses separately, to then divide the mean of the second by the mean of the first amplitude. In contrast to work on pyramidal neurons (*Kim and Alger, 2001*) in our data PPRs not different depending on these two analysis schemes. Short-term facilitation or short-term depression were induced by 60 s stimulation at 1 Hz or by 1 s stimulation at 10 Hz or by 200 ms stimulation at 60 and 120 Hz stimulation frequency. For each animal, the amplitudes of all subsequent EPSCs in each train were plotted over time and fitted with a single exponential. For depression at 1 and 10 Hz, we used one train per animal, and 5–6 animals per genotype. Motoneuron burst firing as occurring during crawling (*Kadas et al., 2017*) was simulated by higher-frequency stimulation for 200 ms at 60 and 120 Hz.

PHP was induced either acutely by application of the bee wolf toxin Philanthotoxin 433 or by using animals carrying a null mutation in the GluRIIA glutamate receptor that is expressed in the muscle membrane. PhTx (10 μM) is applied in calcium-free HL3.1 saline (composition see below) after dissection but before mounting the preparation on the microscope stage. For the toxin to work properly, it must be applied after cutting the animal open along its dorsal midline but prior to spreading the larval body wall laterally (see dissection). The larva must be fixed with two minuten pins through the mouth hooks and the tail but without stretching the animal and without removing any inner organs yet. The preparation is bathed in the toxin solution for 10 min. After rinsing the preparation with HL3.1 saline, the body wall is spread laterally and dissected as described above (see dissection), and the specimen is mounted on the microscope stage ready for recording.

Synaptic transmission on muscle M6 in segments A2 and A3 was assessed in male animals carrying excisions of either *IS4A*, *I–IIA*, or *I–IIB* in cac on their only X-chromosome. Animals lacking exon *IS4B* are homozygous lethal. Thus, we used heterozygous females that expressed cac with *IS4B* excision on one X-chromosome over a cac^FLPStop construct which contains a tdTomato reporter followed by a stop codon within the open reading frame of all cac variants upon expression of a flipase transgene. Briefly, a reverse stop cassette followed by UAS-tdTomato flanked by canonical FRT sites of opposing orientation is inserted within an MiMIC residing in an intron of the *Drosophila* cac channel gene *cacophony*. Presence of canonical flipase turns the entire FlpStop-tdTomato cassette around and -re-inserts it in frame, which leads to pre-mature termination of cac transcription that is reported by tdTomato (*Fisher et al., 2017*). We expressed UAS-flipase under the control of the OK6 (Rapgap1)-GAL4 driver (*Sanyal, 2009*), that drives expression of UAS-transgenes also in larval crawling motoneurons.

However, in our hands, flp events happen in a more or less stochastic manner. Flp events in motoneurons on M6 are less reliable as compared to Flp events on M12. As it is necessary that the FlpStop

technique works in all motoneurons that innervate a target muscle to investigate the role of the IS4B exon for synaptic transmission, we used M12 (left and right) in A3 for this experiment rather than M6.

## Whole-cell voltage clamp recordings

To investigate calcium currents that remain upon excision of IS4B, we used the same FlpStop approach as for synaptic transmission at the larval NMJ. UAS-flipase was expressed under the control of a DLM (dorsal longitudinal muscle) motoneuron Split-GAL4 fly strain that targets mainly the 5 wing depressor motoneurons (MN1–5) on each side of the body (10 neurons total, *Hürkey et al., 2023*). Expression of UAS-flipase under the control of this Split-GAL4 driver reliably led to expression of tdTomato in these motoneurons. In addition, these flies are flight-less.

Adult MN patch clamp experiment was carried out as previously reported (*Ryglewski et al., 2012*). After dissection, the fly was positioned under a 40× water dipping lens (Zeiss, W Apochromat 40×/1.0 DIC VIS-IR ∞/0) on a fixed stage Zeiss AxioExaminer A1 upright fluorescence microscope. MN5 is situated on the dorsal surface of the ventral nerve cord in the mesothoracic neuromere. For whole-cell voltage clamp recordings from MN5, protease was applied focally by alternating positive and negative pressure applied through a broken patch pipette to clean the membrane of the MN5 soma to facilitate seal formation. After cleaning, the preparation was perfused constantly with fresh saline (composition below). Voltage-gated potassium channels were blocked by tetraethylammonium (TEA), 4-aminopyridine (4-AP), and cesium in the recording solution (composition below). Tetrodotoxin ($10^{-7}$ M) to block voltage-gated sodium channels was applied directly to the recording chamber while perfusion was halted for 3 min before approaching the cell with the patch pipette. Then perfusion resumed. Patch pipettes were filled with internal patch solution (composition below) and had resistances of ~3.5–4 MΩ with this combination of internal and external solution. Recordings were carried out with an Axopatch 200B patch clamp amplifier (Molecular Devices). Data were digitized at 50 kHz (Digidata 1440, Molecular Devices) and filtered at 5 kHz through a lowpass Bessel filter. Data were recorded with pClamp 10.7.0.3 software package (Molecular Devices). After giga seal formation, whole-cell configuration was achieved by a brief application of negative pressure to the patch pipette. We allowed 2 min for solution exchange with the cell interior and stabilization of the recording before adjusting whole-cell capacitance and series resistance compensations and setting prediction at around 80% and correction at around 45%. Holding potential was −70 mV between voltage clamp protocols. Voltage-gated calcium currents were elicited by voltage command steps to +20 mV in 10 mV increments from a holding potential of −90 mV. Leak was subtracted offline.

## Dissection and electrophysiology salines

Adult dissection saline [mM]: NaCl 128, KCl 2, $CaCl_2$ 1.8, $MgCl_2$ 4, HEPES 5, Sucrose ~35.5. Osmolality was adjusted to 300 mOsM/kg with sucrose. pH was adjusted to 7.24 with 1 N NaOH.

Larval dissection saline HL3.1 calcium free [mM]: NaCl 70, KCl 5, $MgCl_2$ 4, $NaHCO_3$ 10, trehalose 5, sucrose 115, HEPES 5. Osmolality was adjusted to 300 mOsM/kg with sucrose. pH was adjusted to 7.24 with 1 N NaOH.

TEVC recording saline HL3.1 [mM]: NaCl 70, KCl 5, $CaCl_2$ 0.5, $MgCl_2$ 4, $NaHCO_3$ 10, trehalose 5, sucrose 115, HEPES 5. Osmolality was adjusted to 300 mOsM/kg with sucrose. pH was adjusted to 7.24 with 1 N NaOH.

Whole-cell voltage clamp saline [mM]: NaCl 93.2, KCl 5, $MgCl_2$ 4, $CaCl_2$ 1.8, $BaCl_2$ 1.8, TEA Cl 30, 4-AP 2, HEPES 5, sucrose ~35.5. Osmolality was adjusted to 300 mOsM/kg with sucrose. pH was adjusted to 7.24 with 1 N NaOH.

Tetrodotoxin ($10^{-7}$ M) in normal saline was applied directly to the recording chamber.

Whole-cell voltage clamp intracellular solution [mM]: CsCl 144, $CaCl_2$ 0.5, EGTA 5, HEPES 10, TEA-Br 20, 4-AP 0.5, Mg-ATP 2. Osmolality was adjusted to 300 mOsm/kg with glucose. pH was adjusted to 7.24 with 1 N CsOH.

## mEOS4b-tagged cac channel counting

The calculation of cac channel splice form numbers in individual AZs of the NMJ are based on the photoconversion of the fluorescent tag mEOS4b fused to the N-terminus of the cac channel (cac::mEOS4b-N; *Ghelani et al., 2023*). Third instar larvae were used for a body wall dissection. This preparation was suitable to image individual AZs within single boutons of an NMJ. We focused on

muscle 6/7 on large type Ib boutons (*Jia et al., 1993*). Experiments were conducted at 25°C with HL3.1 solution (composition see above). We used a TIRF setup based on an inverted microscope (Nikon Eclipse Ti) equipped with a 60× 1.49 NA oil immersion objective (Nikon). Image series of up to 5000 frames were acquired at a frame rate of 20 Hz, using a sCMOS camera (Hamamatsu, Orca flash 4.0) controlled by the NIS-Element acquisition software (Nikon). Labeling of the NMJ with an anti-HRP antibody (diluted 1:1000) directly tagged with Alexa-488 for 5 min was used to have an initial landmark for NMJs within the preparation. During imaging of cac channels, we use a 1.5 magnification lens to reduce the effective pixel size to 71 × 71 nm. The used illumination protocol started with an initial continuous illumination with a 561-nm laser (80% of initial laser power of 100 mW) for 250 frames to reduce the autofluorescent background. Afterward, we triggered the photoconversion of mEOS4b by switching on the photoconversion with a 405-nm UV-laser (5% of initial laser power of 100 mW) in addition to the 561-nm laser. The UV excitation was sufficient to convert the majority of mEOS4b molecules into the red fluorescent state that was read out by the 561-nm laser. With the dual excitation of 405 and 561 nm, synapses were imaged until complete bleaching of the cac::mEOS4b-N population. The ability of the TIRF setup to adjust the laser beam orientation in the specimen was used to obtain an oblique illumination profile that limited the contribution of out of focus fluorescence. The number of channels was calculated as ratio of the maximal fluorescent intensity shortly after starting the photoconversion and the fluorescence of single mEOS4b molecules, which occurred as stochastic blinking events in the second half of the illumination sequence (see *Figure 4G*, *Ghelani et al., 2023*). Two-dimensional x–y movements of the NMJ was corrected by using the NanoJ-Core drift correction from the NanoJ-Plugin for ImageJ/Fiji (*Laine et al., 2019*). After background subtraction, regions of 5 × 5 pixels were used to read out the fluorescence intensity profile of individual AZs. At least 30 AZs per ROI and larva were analyzed. For plotting the data and statistical calculations, we used Igor Pro 8 and GraphPad Prism 10.2.3.

## Immunohistochemistry

For cacophony immunohistochemistry, L3 larvae were dissected as described above and the VNC along with the motor nerves were removed. The specimen was then fixed for 7 min in ice cold (−30°C) 100% ethanol. For this, the saline was replaced by ice cold ethanol and rinsed a few times to assure cooling down of the specimen. Then the preparation was kept in the freezer at −30°C for 7 min. Afterwards, the specimen was treated with PBS 0.1 M with 0.3% Triton X (PBS-Tx) at room temperature for 3 × 10 min, rocking.

cac[sfGFP] (*Gratz et al., 2019*) localization (*Figures 2 and 3*): Primary antibodies (α-brp nc-82, 1:400, DSHB; α-HRP rabbit, 1:500, Jackson Immunoresearch Cat# 323-005-021) were applied in PBS-Tx 0.3% overnight at 4°C, rocking. Next day, primary antibody solution was removed, the preparation rinsed a few times with PBS-Tx 0.3% and then washed 3 × 10 min with PBS-Tx 0.3% at room temperature, rocking. Two-hour application of α-GFP nanobody (Fluotag X4 α-GFP Alexa 647, Nanotag Biotechnologies, Germany, Cat# N0304-AF647) and secondary antibodies donkey α-mouse Alexa 555 (Jackson Immunoresearch, Cat# 715-165-151) and donkey α-rabbit Alexa 488 (Jackson Immunoresearch, Cat# 711-545-152), all at a concentration of 1:500 at room temperature.

Assessment of ΔI–IIA and ΔI–IIB channel expression: Primary antibodies (α-brp nc-82, 1:400, DSHB; α-mEOS rabbit, 1:500, Badrilla, Cat# A010-mEOS2) were applied in PBS-Tx 0.3% overnight at 4°C, rocking. Next day, primary antibody solution was removed, the preparation rinsed a few times with PBS-Tx 0.3% and then washed 3 × 10 min with PBS-Tx 0.3% at room temperature, rocking. Two-hour application of α-GFP nanobody (Fluotag X4 α-GFP Abberior Star Red, Nanotag Biotechnologies, Germany, Cat# N0304-ABRED) and secondary antibodies donkey α-mouse Alexa 555 (Jackson Immunoresearch, Cat# 715-165-151) and donkey α-rabbit Alexa 488 (Jackson Immunoresearch, Cat# 711-545-152), all at a concentration of 1:500 at room temperature.

For GluRIIA[GFP] immunostaining, L3 larvae were fixed for 5 min in Bouin's solution at room temperature. Then 3 × 10 min washed were performed with 0.3% PBS-Tx at room temperature. This was followed by primary antibody incubation with α-brp (nc-82, 1:400, DSHB) overnight. After thre more 10 min washes with PBS-Tx 0.3%, specimen were incubated with α-GFP nanobody (Fluotag X4 α-GFP Abberior Star Red, Nanotag Biotechnologies, Germany, Cat# N0304-ABRED) and secondary donkey α-mouse Alexa 555 antibody (Jackson Immunoresearch, Cat# 715-165-151) for 3 hr at room temperature, covered to protect from light.

All immunolabel: Preparations were covered to prevent bleaching of fluorophores. Then, antibodies were removed, the preparations were rinsed a few times, which was followed by 3 – 10 min PBS (no Tx). Then an ascending ethanol series with 10 min each with 50, 70, 90, and 100% ethanol was applied at room temperature, rocking. Preparations were then mounted in methylsalicylate, covered with a high precision cover slip that was sealed with clear nail polish. Preparations were kept in the dark until scanning with a Leica TSC SP8 upright confocal laser scanning microscope with a 40× oil NA 1.3 oil lens. Overviews were scanned without zoom at 1 μm z-step size. Magnifications were scanned with a 3.5 zoom with z-steps of 0.3 μm. All scans were done with a 3× line average.

## Assessment of fluorescence intensity

For fluorescence intensity measurement of sfGFP-tagged cac channels in larval AZs, immunohistochemical labeling was carried out as for localization of cac channels. The only difference was that each round contained animals of all genotypes that were assessed. Animals were treated in the identical dish at the same time to ensure identical treatment. Confocal images were acquired with identical settings (lasers, detectors, scanning speed).

Fluorescence intensity was compared between genotypes by employing a custom Python script (available on Zenodo) in which we used brp labeling (nc-82 antibody, DSHB) as a mask and asked for cac intensity within this mask. Intensity was then normalized to control. brp intensity was assessed identically.

GluRIIA fluorescence intensity was assessed identically, but as GluRIIA fields are larger than brp, we inflated the brp mask to encompass GluRIIA label. We then used the aforementioned python script to quantify intensity. For GluRIIA label we used flies that expressed a GluRIIA$^{GFP}$ transgene that was previously shown to exhibit native GluRIIA expression levels (*Rasse et al., 2005*).

## Co-localization of cac and brp scaffold protein

To determine the Pearson's correlation coefficient and the Manders co-occurrence coefficients of cac and brp stainings using the Costes' threshold calculation (*Manders et al., 1993*; *Costes et al., 2004*) a custom Python script was utilized (available on Zenodo).

In image stacks triple stained for cac, brp, and HRP that had been deconvolved with the software Huygens, HRP staining served as a mask in which the desired coefficients were calculated for cac and brp.

## Super resolution microscopy (STED) and deconvolution

STED images were acquired at a Leica Stellaris 8 STED system (*Leica microsystems*) equipped with a pulsed white light laser (WLL) for excitation ranging from 440 to 790 nm and a 775-nm pulsed laser for depletion. Samples were imaged with a 93× glycerol objective (*Leica,* HC APO 93×/1.30 GLYC motCORR). For excitation of the respective channels the WLL was set to 640 nm for FluoTag-X4 anti-GFP Atto647N (NanoTag Biotechnologies, Cat# N0304-At647N) or 580 nm for Abberior Star 580 conjugated FluoTag-X2 anti-mouse antibody (NanoTag Biotechnologies, Cat# N2702-Ab580). Emission were detected in between 650 and 710 nm for Atto647N and in between 590 and 640 nm for Abberior Star 580. STED was attained by using the 775 nm laser for both channels. Additionally a third confocal channel was acquired (HRP, not shown, labeled with rabbit α-HRP antibody, Jackson Immunoresearch Cat# 323-005-021) by using 488 nm light for excitation of Alexa488 (Jackson Immunoresearch, Cat# 711-545-152) and an emission band of 500–530 nm. Scanning properties were set to a format of 1024 × 1024 pixels, optical zoom factor to 5 ($x/y$ = 24.44 nm, $z$ = 191.69 nm) and scanning speed to 400 lines per second. Detectors were operated in in photon counting mode for both channels by three times line accumulation. For $_{gated}$STED, detector time gates were set to 0.5–6 ns for both channels.

Deconvolution of STED stacks was done with Huygens Essential (*Scientific Volume Imaging, The Netherlands,* http://svi.nl). Within the *Deconvolution wizard*, images were subjected to background correction. *Signal-to-noise ratio* was set to 20. *The Optimized iteration mode of the CMLE* was applied by using 40 Iterations.

## Behavioral experiments

Wandering L3 larvae were selected off vial walls for crawling experiments and starved in empty vials for 5 min. Afterwards, they were placed in the middle of a plasticine square on an agarose gel (1% in $H_2O$) using a brush. This square itself was placed on a tracking table with an acrylic glass plate. Usually, 10 larvae of the same genotype were recorded at a time. Occasionally, less than 10 larvae were available, in which case fewer larvae were recorded. Dishes were freshly poured each day before starting experiments. To prevent the larvae from escaping the boundary, an electrically charged wire was placed inside the square boundaries. Crawling pictures were obtained with a camera (Basler acA2040-90 µm) from below the glass plate at a frame rate of 4 Hz for 5 min via pylon viewer (Version 6). Since few larvae were able to dig into the gel through small holes, only larvae that stayed inside the arena for 5 min were included into analysis. Tracking data of larval crawling were analyzed via the free software FIMtrack (University of Münster, Germany). Parameters that were read out included (but were not limited to) number of stops, average, and maximum speed. The full list of data that were read out is available on Zenodo. For explanation of read-out parameters, consult the FIMtrack manual (University of Münster website).

## Statistics and figure generation

Statistical analysis was performed with GraphPad Prism 10.2.1 (395). Data were tested for normal distribution using the Shapiro–Wilk test. Normally distributed data were compared using either Student's *t*-test (for two groups) or one-way ANOVA (for more than two groups) with subsequent Sidak's post hoc test for multiple comparisons or Dunnett's test for each test group against control (planned comparisons). For non-normally distributed data either Mann–Whitney *U*-test (for two groups) or Kruskal–Wallis ANOVA (with more than two groups) with subsequent Dunn's post hoc test for multiple comparisons or for test groups against control (planned comparisons) were used.

Differences were considered significant if $*p \leq 0.05$, $**p \leq 0.01$, $***p \leq 0.001$, $****p \leq 0.0001$.

Diagrams were generated with GraphPad Prism 10.2.1 (395) or Microsoft Excel 365. Figures with immunohistochemical images were generated with LasX software (Leica Microsystems, Wetzlar, Germany), with Fiji (https://www.Fiji.sc), or with Amira (version 6.5, Thermo Scientific). Images and diagrams were imported into CorelDraw Graphics Suite 2022 (Corel Corporation) either as Tiff images or as enhanced meta files (.emf) for figure production. Final figures were exported as.tif files.

All data and script are available on Zenodo. Cacophony exon-out flies will be sent to the *Drosophila* Bloomington Stock Center upon publication.

## Acknowledgements

We thank Kerstin Birod for performing western blots, Kate O'Connor-Giles for providing cacophony[mEOS4b] flies, Tayfun Göncü for participating in gathering STED data, Susanne Hornig for confocal recordings of co-expressed cacophony exon-out variants and Jan Werner for GluRIIA immunohistochemistry. This work was supported by a DFG research grant to S Ryglewski (RY117/3-2) and to the Johannes Gutenberg Light Microscopy Core facility (Leica Stellaris STED, DFG INST 247/1004-1 FUGG).

## Additional information

### Funding

| Funder | Grant reference number | Author |
| --- | --- | --- |
| Deutsche Forschungsgemeinschaft | RY117/3-2 | Stefanie Ryglewski |
| Deutsche Forschungsgemeinschaft | INST 247/1004-1 FUGG | Martin Heine |

The funders had no role in study design, data collection and interpretation, or the decision to submit the work for publication.

## Author contributions
Christopher Bell, Daniel Gottschalk, Jashar Arian, Lea Deneke, Hanna Kern, Julia Strauß, Formal analysis, Investigation; Lukas Kilo, Oliver Kobler, Formal analysis, Investigation, Methodology; Christof Rickert, Formal analysis, Methodology; Martin Heine, Conceptualization, Formal analysis, Supervision, Investigation, Methodology, Writing – original draft; Carsten Duch, Conceptualization, Resources, Formal analysis, Validation, Investigation, Writing – original draft, Writing – review and editing; Stefanie Ryglewski, Conceptualization, Resources, Data curation, Software, Formal analysis, Supervision, Funding acquisition, Validation, Investigation, Visualization, Methodology, Writing – original draft, Project administration, Writing – review and editing

## Author ORCIDs
Oliver Kobler ⓘ https://orcid.org/0000-0001-7173-9919
Carsten Duch ⓘ https://orcid.org/0000-0002-6962-6023
Stefanie Ryglewski ⓘ https://orcid.org/0000-0002-3066-6298

Reviewer #2 (Public review): https://doi.org/10.7554/eLife.100394.3.sa1
Reviewer #3 (Public review): https://doi.org/10.7554/eLife.100394.3.sa2
Author response https://doi.org/10.7554/eLife.100394.3.sa3

## Additional files

### Supplementary files
MDAR checklist

### Data availability
All data have been deposited to the public repository Zenodo.

The following dataset was generated:

| Author(s) | Year | Dataset title | Dataset URL | Database and Identifier |
|---|---|---|---|---|
| Ryglewski S | 2024 | Specific presynaptic functions require distinct Drosophila Cav2 splice isoforms | https://doi.org/ 10.5281/zenodo. 11383963 | Zenodo, 10.5281/ zenodo.11383963 |

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
