## [Editor Report · eLife Assessment]

Cav2 voltage-gated calcium channels play key roles in regulating synaptic strength and plasticity. In contrast to mammals, invertebrates like *Drosophila* encode a single Cav2 channel, raising questions on how diversity in Cav2 is achieved from a single gene. Here, the authors present **solid** evidence that two alternatively spliced Cac isoforms enable **important** changes in Cav2 expression, localization, and function in synaptic transmission and plasticity at the *Drosophila* neuromuscular junction. How the isoforms affect synaptic calcium channel levels remains less clear. This study provides insights into the roles of voltage-gated calcium channel splice isoforms in synaptic transmission.

---

## [Referee Report · Reviewer #2 (Public review)]

This study by Bell et al. focuses on understanding the roles of two alternatively spliced exons in the single *Drosophila* Cav2 gene cac. The authors generate a series of cac alleles in which one or the other mutually exclusive exons are deleted to determine the functional consequences at the neuromuscular junction. They find alternative splicing at one exon encoding part of the voltage sensor impacts the activation voltage as well as localization to the active zone. In contrast, splicing at the second exon pair does not impact Cav2 channel localization, but it appears to determine the abundance of the channel at active zones. Together, the authors propose that alternative splicing at the Cac locus enables diversity in Cav2 function generated through isoform diversity generated at the single Cav2 alpha subunit gene encoded in *Drosophila*.

Overall this is an excellent, rigorously validated study that defines unanticipated functions for alternative splicing in Cav2 channels. The authors have generated an important toolkit of mutually exclusive Cac splice isoforms that will be of broad utility for the field, and show convincing evidence for distinct consequences of alternative splicing of this single Cav2 channel at synapses. Importantly, the authors use electrophysiology and quantitative live sptPALM imaging to determine the impacts of Cac alternative splicing on synaptic function. There remain some questions regarding the mechanisms underlying the changes in Cac localization to somatodendritic compartments. Nonetheless, this is a compelling investigation of alternative splicing in Cav2 channels that should be of interest to many researchers.

---

## [Referee Report · Reviewer #3 (Public review)]

Summary:

Bell and colleagues studied how different splice isoforms of voltage-gated CaV2 calcium channels affect channel expression, localization, function, synaptic transmission, and locomotor behavior at the larval *Drosophila* neuromuscular junction. They reveal that one mutually exclusive exon located in the fourth transmembrane domain encoding the voltage sensor is essential for calcium channel expression, function, active zone localization, and synaptic transmission. Furthermore, a second mutually exclusive exon residing in an intracellular loop containing the binding sites for Caβ and G-protein βγ subunits promotes the expression and synaptic localization of around ~50% of CaV2 channels, thereby contributing to ~50% of synaptic transmission. This isoform enhances release probability, as evident from increased short-term depression, is vital for homeostatic potentiation of neurotransmitter release induced by glutamate receptor impairment, and promotes locomotion. The roles of the two other tested isoforms remain less clear.

Strengths:

The study is based on solid data that was obtained with a diverse set of approaches. Moreover, it generated valuable transgenic flies that will facilitate future research on the role of calcium channel splice isoforms in neural function.

Weaknesses:

Comments on revisions:

The authors addressed most points. However, from my point of view, the new data (somatodendritic cac currents in adult motoneurons of IS4B mutants without the pre-pulse, and localization of IS4A channels in the larval brain) do not strongly support that the IS4B exon is required for cacophony localization. According to their definition of localization, IS4B is required for cacophony channels to enter motoneuron boutons and to localize to active zones. In case of a true localization defect (without degradation, as they claim), IS4A channels should mislocalize to the soma, axon, or dendrite. However, they do not find them in motoneurons of IS4B mutants. Furthermore, I find the interpretation of the voltage clamp data in flight motoneurons rather difficult. On the one hand, sustained HVA cac currents are strongly attenuated/absent in IS4B mutants. On the other hand, total cac currents (without the -50 mV pre-pulse, not shown in the original submission) are less affected in IS4B mutants. Based on these data, they conclude that IS4B is required for sustained HVA cac currents and that IS4A channel isoforms are expressed and functional. How does this relate to a localization defect at the NMJ? Finally, detecting IS4A channels in other cell types and regions is not a strong argument for a localization defect at the NMJ. I, therefore, suggest toning down the conclusions regarding a localization defect in IS4B mutants/a role for the IS4B exon in cac localization. It should be also discussed how a splice isoform in S4 may result in no detectable cac channels at the NMJ or regulate subcellular channel localization.

I have a few additional points, mainly related to the responses to my previous points:

(1) The authors state "active zones at the NMJ contain only cac isoforms with the IS4B exon. Nevertheless, the small representative EPSC remaining in IS4B mutants suggests that there is synchronous release in the absence of IS4B (Fig. 3B). Are the small EPSCs in dIS4B (Fig. 3B) indeed different from noise/indicative of evoked release? If yes, which cac channels may drive these EPSCs? IS4A channels?

(2) (Related to previous point 4) The authors argue that EPSC amplitudes are not statistically different between Canton S and IS4A mutants (Fig. 2F). However, the Canton S group appears undersampled, thus precluding conclusions based on statistics. Moreover, the effect size Canton S vs. dIS4A looks similar to the one of Canton S vs. dIS4A/dIS4B.

(3) (Related to previous point 11): Can they cite a paper relating calcium channel inactivation to EPSC half width/decay kinetics to support their speculation that "decreased EPSC half width could be caused by significantly faster channel inactivation kinetics" (p. 42, l.42). In addition, many papers have demonstrated that mini decay kinetics provide valuable insights into GluR subunit composition at the *Drosophila* NMJ (e.g., Schmid et al., 2008 https://doi.org/10.1038/nn.2122). Thus, the statement "Mini decay kinetic analysis because this depends strongly on the distance of the recording electrode to the actual site of transmission in these large muscle cells" is not valid and should be revised.

---

## [Author Response]

The following is the authors’ response to the original reviews.

**Public Reviews:**

**Reviewer #1 (Public Review):**
Summary:The manuscript by Bell et. al. describes an analysis of the effects of removing one of two mutually exclusive splice exons at two distinct sites in the *Drosophila* CaV2 calcium channel Cacophony (Cac). The authors perform imaging and electrophysiology, along with some behavioral analysis of larval locomotion, to determine whether these alternatively spliced variants have the potential to diversify Cac function in presynaptic output at larval neuromuscular junctions. The author provided valuable insights into how alternative splicing at two sites in the calcium channel alters its function.Strengths:The authors find that both of the second alternatively spliced exons (I-IIA and I-IIB) that are found in the intracellular loop between the 1st and second set of transmembrane domains can support Cac function. However, loss of the I-IIB isoform (predicted to alter potential beta subunit interactions) results in 50% fewer channels at active zones and a decrease in neurotransmitter release and the ability to support presynaptic homeostatic potentiation. Overall, the study provides new insights into Cac diversity at two alternatively spliced sites within the protein, adding to our understanding of how regulation of presynaptic calcium channel function can be regulated by splicing.Weaknesses:The authors find that one splice isoform (IS4B) in the first S4 voltage sensor is essential for the protein's function in promoting neurotransmitter release, while the other isoform (IS4A) is dispensable. The authors conclude that IS4B is required to localize Cac channels to active zones. However, I find it more likely that IS4B is required for channel stability and leads to the protein being degraded, rather than any effect on active zone localization. More analysis would be required to establish that as the mechanism for the unique requirement for IS4B.

(1) We thank the reviewer for this important point. In fact, all three reviewers raised the same question, and the reviewing editor pointed out that caution or additional experiments were required to distinguish between IS4 splicing being important for cac channel localization versus channel stability/degradation. We provide multiple sets of experiments as well as text and figure revisions to strengthen our claim that the IS4B exon is required for cacophony channels to enter motoneuron presynaptic boutons and localize to active zones.

a. If IS4B was indeed required for cac channel stability (and not for localization to active zones) IS4A channels should be instable wherever they are. This is not the case because we have recorded somatodendritic cacophony currents from IS4A expressing adult motoneurons that were devoid of cac channels with the IS4B exon. Therefore, IS4A cac channels are not instable but underlie somatodendritic voltage dependent calcium currents in these motoneurons. These new data are now shown in the revised figure 3C and referred to in the text on page 7, line 42 to page 8 line 9.

b. Similarly, if IS4B was required for channel stability, it should not be present anywhere in the nervous system. We tested this by immunohistochemistry for GFP tagged IS4A channels in the larval CNS. Although IS4A channels are sparsely expressed, which is consistent with low expression levels seen in the Western blots (Fig. 1E), there are always defined and reproducible patterns of IS4A label in the larval brain lobes as well as in the anterior part of the VNC. This again shows that the absence of IS4A from presynaptic active zones is not caused by channel instability, because the channel is expressed in other parts of the nervous system. These data are shown in the new supplementary figure 1 and referred to in the text on page 15, lines 3 to 8.

c. As suggested in a similar context by reviewers 1 and 2, we now show enlargements of the presence of IS4B channels in presynaptic active zones as well as enlargements of the absence of IS4A channels in presynaptic active zones in the revised figures 2A-C and 3A. In these images, no IS4A label is detectable in active zones or anywhere else throughout the axon terminals, thus indicating that IS4B is required for expressing cac channels in the axon terminal boutons and localizing it to active zones. Text and figure legends have been adjusted accordingly.

d. Related to this, reviewer 1 also recommended to quantify the IS4A and ISB4 channel intensity and co-localization with the active zone marker brp (recommendation for authors). After following the reviewers’ suggestion to adjust the background values in IS4A and IS4B immunolabels to identical (revised Figs. 2A-C), it becomes obvious that IS4A channel are not detectable above background in presynaptic terminals or active zones, thus intensity is close to zero. We still calculated the Pearsons co-localization coefficient for both IS4 variants with the active zone marker brp. For IS4B channels the Pearson’s correlation coefficient is control like, just above 0.6, whereas for IS4A channels we do not find colocalization with brp (Pearson’s below 0.25). These new analyses are now shown in the revised figure 2D and referred to on page 6, lines 33 to 38.

e. Consistent with our finding that IS4B is required for cac channel localization to presynaptic active zones, upon removal of IS4B we find no evoked synaptic transmission (Fig. 2 in initial submission, now Fig. 3B).

Together these data are in line with a unique requirement of IS4B at presynaptic active zones (not excluding additional functions of IS4B), whereas IS4A containing cac isoforms are not found in presynaptic active zones and mediate different functions.

**Reviewer #2 (Public Review):**
This study by Bell et al. focuses on understanding the roles of two alternatively spliced exons in the single *Drosophila* Cav2 gene cac. The authors generate a series of cac alleles in which one or the other mutually exclusive exons are deleted to determine the functional consequences at the neuromuscular junction. They find alternative splicing at one exon encoding part of the voltage sensor impacts the activation voltage as well as localization to the active zone. In contrast, splicing at the second exon pair does not impact Cav2 channel localization, but it appears to determine the abundance of the channel at active zones.Together, the authors propose that alternative splicing at the Cac locus enables diversity in Cav2 function generated through isoform diversity generated at the single Cav2 alpha subunit gene encoded in *Drosophila*.Overall this is an excellent, rigorously validated study that defines unanticipated functions for alternative splicing in Cav2 channels. The authors have generated an important toolkit of mutually exclusive Cac splice isoforms that will be of broad utility for the field, and show convincing evidence for distinct consequences of alternative splicing of this single Cav2 channel at synapses. Importantly, the authors use electrophysiology and quantitative live sptPALM imaging to determine the impacts of Cac alternative splicing on synaptic function. There are some outstanding questions regarding the mechanisms underlying the changes in Cac localization and function, and some additional suggestions are listed below for the authors to consider in strengthening this study. Nonetheless, this is a compelling investigation of alternative splicing in Cav2 channels that should be of interest to many researchers.

(2) We believe that the additional data on cac IS4A isoform localization and function as detailed above (response to public review 1) has strengthened the manuscript and answered some of the remaining questions the reviewer refers to. We are also grateful for the specific additional reviewer suggestions which we have addressed point-by-point and refer to below (section recommendations for authors).

**Reviewer #3 (Public Review):**
Summary:Bell and colleagues studied how different splice isoforms of voltage-gated CaV2 calcium channels affect channel expression, localization, function, synaptic transmission, and locomotor behavior at the larval *Drosophila* neuromuscular junction. They reveal that one mutually exclusive exon located in the fourth transmembrane domain encoding the voltage sensor is essential for calcium channel expression, function, active zone localization, and synaptic transmission. Furthermore, a second mutually exclusive exon residing in an intracellular loop containing the binding sites for Caβ and G-protein βγ subunits promotes the expression and synaptic localization of around ~50% of CaV2 channels, thereby contributing to ~50% of synaptic transmission. This isoform enhances release probability, as evident from increased short-term depression, is vital for homeostatic potentiation of neurotransmitter release induced by glutamate receptor impairment, and promotes locomotion. The roles of the two other tested isoforms remain less clear.Strengths:The study is based on solid data that was obtained with a diverse set of approaches. Moreover, it generated valuable transgenic flies that will facilitate future research on the role of calcium channel splice isoforms in neural function.Weaknesses:(1) Based on the data shown in Figures 2A-C, and 2H, it is difficult to judge the localization of the cac isoforms. Could they analyze cac localization with regard to Brp localization (similar to Figure 3; the term "co-localization" should be avoided for confocal data), as well as cac and Brp fluorescence intensity in the different genotypes for the experiments shown in Figure 2 and 3 (Brp intensity appears lower in the dI-IIA example shown in Figure 3G)? Furthermore, heterozygous dIS4B imaging data (Figure 2C) should be quantified and compared to heterozygous cacsfGFP/+.

According to the reviewer’s suggestion, we have quantified cac localization relative to brp localization by computing the Pearson’s correlation coefficient for controls and IS4A as well as IS4B animals. These new data are shown in the revised Fig. 2D and referred to on page 6, lines 33-38. Furthermore, we now confirm control-like Pearson’s correlation coefficients for all exon out variants except *ΔIS4B* and show Pearson’s correlation coefficients for all genotypes side-by-side in the revised Fig. 4D (legend has been adjusted accordingly). In addition, in response to the recommendations to authors, we now provide selective enlargements for the co-labeling of Brp and each exon out variant in the revised figures 2-4. We have also adjusted the background in Fig. 2C (*ΔIS4B*) to match that in Figs. 2A and B (control and *ΔIS4A*). This allows a fair comparison of cac intensities following excision of *IS4B* versus excision of *IS4A* and control (see also Fig 3). Together, this demonstrates the absence of IS4A label in presynaptic active zones much clearer. As suggested, we have also quantified brp puncta intensity on m6/7 across homozygous exon excision mutants and found no differences (this is now stated for IS4A/IS4B in the results text on page 6, lines 37/38 and for I-IIA/I-IIB on page 8, lines 42-44.). We did not quantify the intensity of cacophony puncta upon excision of *IS4B* because the label revealed no significant difference from background (which can be seen much better in the images now), but the brp intensities remained control-like even upon excision of *IS4B*.

(2) They conclude that I-II splicing is not required for cac localization (p. 13). However, cac channel number is reduced in dI-IIB. Could the channels be mis-localized (e.g., in the soma/axon)? What is their definition of localization? Could cac be also mis-localized in dIS4B? Furthermore, the Western Blots indicate a prominent decrease in cac levels in dIS4B/+ and dI-IIB (Figure 1D). How do the decreased protein levels seen in both genotypes fit to a "localization" defect? Could decreased cac expression levels explain the phenotypes alone?

We have now precisely defined what we mean by cac localization, namely the selective label of cac channels in presynaptic active zones that are defined as brp puncta, but no cac label elsewhere in the presynaptic bouton (page 6, lines 18 to 20). On the level of CLSM microscopy this corresponds to overlapping cac puncta and brp puncta, but no cac label elsewhere in the bouton. Based on the additional analysis and data sets outlined in our response 1 (see above) we conclude that excision of *IS4B* does not cause channel mislocalization because we find reproducible expression patterns elsewhere in the nervous system as well as somatodendritic cac current in *ΔIS4B* (for detail see above). Therefore, the isoforms containing the mutually exclusive *IS4A* exon are expressed and mediate other functions, but cannot substitute *IS4B* containing isoforms at the presynaptic AZ. In fact, our Western blots are in line with reduced cac expression if all isoforms that mediate evoked release are missing, again indicating that the presynapse specific cac isoforms cannot be replaced by other cac isoforms. This is also in line with the sparse expression of IS4A throughout the CNS as seen in the new supplementary figure 1 (for detail see above).

(3) Cac-IS4B is required for Cav2 expression, active zone localization, and synaptic transmission. Similarly, loss of cac-I-IIB reduces calcium channel expression and number. Hence, the major phenotype of the tested splice isoforms is the loss of/a reduction in Cav2 channel number. What is the physiological role of these isoforms? Is the idea that channel numbers can be regulated by splicing? Is there any data from other systems relating channel number regulation to splicing (vs. transcription or post-transcriptional regulation)?

Our data are not consistent with the idea that splicing regulates channel numbers. Rather, splicing can be used to generate channels with specific properties that match the demand at the site of expression. For the *IS4* exon pair we find differences in activation voltage between IS4A and IS4B channels (revised Fig. 3C), with IS4B being required for sustained HVA current. IS4A does not localize to presynaptic active zones at the NMJ and is only sparsely expressed elsewhere in the NS (new supplementary Fig. 1). By contrast, IS4B is abundantly expressed in many neuropils. Therefore, taking out *IS4B* takes out the more abundant IS4 isoform. This is consistent with different expression levels for IS4 isoforms that have different functions, but we do not find evidence for splicing regulating expression levels *per se*.

Similarly, the *I-II* mutually exclusive exon pair differs markedly in the presence or absence of G-protein βγ binding sites that play a role in acute channel regulation as well the conservation of the sequence for β-subunit binding (see page 5, lines 9-17). Channel number reduction in active zones occurs specifically if expression of the cac channels with the G_βγ_-binding site as well as the more conserved β-subunit binding is prohibited by excision of the *I-IIB* exon (see Fig. 5F). *Vice versa*, excision of *I-IIA* does not result in reduced channel numbers. This scenario is consistent with the hypothesis that conserved β-subunit binding affects channel number in the active zone (see page 17, lines 3 to 6 and lines 33-36), but we have no evidence that I-II splicing *per se* affects channel number.

(4) Although not supported by statistics, and as appreciated by the authors (p. 14), there is a slight increase in PSC amplitude in dIS4A mutants (Figure 2). Similarly, PSC amplitudes appear slightly larger (Figure 3J), and cac fluorescence intensity is slightly higher (Figure 3H) in dI-IIA mutants. Furthermore, cac intensity and PSC amplitude distributions appear larger in dI-IIA mutants (Figures 3H, J), suggesting a correlation between cac levels and release. Can they exclude that IS4A and/or I-IIA negatively regulate release? I suggest increasing the sample size for Canton S to assess whether dIS4A mutant PSCs differ from controls (Figure 2E). Experiments at lower extracellular calcium may help reveal potential increases in PSC amplitude in the two genotypes (but are not required). A potential increase in PSC amplitude in either isoform would be very interesting because it would suggest that cac splicing could negatively regulate release.

There are several possibilities to explain this, but as none of the effects is statistically significant, we prefer to not investigate this in further depth. However, given that we cannot find IS4A in presynaptic active zones (revised figures 2C and 3A plus the new enlargements 2Ci and 3Ai, revised text page 6, lines 22 to 24 and 29 to 31, and page 7, second paragraph, same as public response 1D) IS4A channels cannot have a direct negative effect on release probability. Nonetheless, given that IS4A containing cac isoforms mediate functions in other neuronal compartments (see revised Fig. 3C) it may regulate release indirectly by affecting e.g. action potential shape. Moreover, in response to the more detailed suggestions to authors we provide new data that give additional insight.

(5) They provide compelling evidence that IS4A is required for the amplitude of somatic sustained HVA calcium currents. However, the evidence for effects on biophysical properties and activation voltage (p. 13) is less convincing. Is the phenotype confined to the sustained phase, or are other aspects of the current also affected (Figure 2J)? Could they also show the quantification of further parameters, such as CaV2 peak current density, charge density, as well as inactivation kinetics for the two genotypes? I also suggest plotting peaknormalized HVA current density and conductance (G/Gmax) as a function of Vm. Could a decrease in current density due to decreased channel expression be the only phenotype? How would changes in the sustained phase translate into altered synaptic transmission in response to AP stimulation?

Most importantly, sustained HVA current is abolished upon excision of *IS4B* (not *IS4A*, we think the reviewer accidentally mixed up the genotype) and presynaptic active zones at the NMJ contain only cac isoforms with the *IS4B* exon. This indicates that the cac isoforms that mediate evoked release encode HVA channels. The somatodendritic currents shown in the revised figure 3C (previously 2J) that remain upon excision of *IS4B* are mediated by IS4A containing cac isoforms. Please note that these never localize to the presynaptic active zone, and thus do not contribute to evoked release. Therefore, the interpretation is that specifically sustained HVA current encoded by IS4B cac isoforms is required for synaptic transmission. Reduced cac current density due to decreased channel expression is not the cause for impaired evoked release upon IS4B excision, but instead, the cause is the absence of any cac channels in active zones. IS4B-containing cac isoforms encode sustained HVA current, and we speculate that this might be a well suited current to minimize cacophony channel inactivation in the presynaptic active zone. Given that HVA current shows fast voltage dependent activation and fast inactivation upon repolarization, it is useful at large intraburst firing frequencies as observed during crawling (Kadas et al., 2017) without excessive cac inactivation (see page 15, Kadas, lines 16 to 20).

However, we agree with the reviewer that a deeper electrophysiological analysis of splice isoform specific cac currents will be instructive. We have now added traces of control and *ΔIS4B* from a holding potential of -90 mv (revised Fig. 3C, bottom traces and revised text on page 7, line 43 to page 8, lines 1 to 10), and these are also consistent with IS4B mediating sustained HVA cac current. However, further analysis of activation and inactivation voltages and kinetics suffers form space clamp issues in recordings from the somata of such complex neurons (DLM motoneurons of the adult fly contain roughly 6000 µm of dendrites with over 4000 branches, Ryglewski et al., 2017, Neuron 93(3):632-645). Therefore, we will analyze the currents in a heterologous expression system and present these data to the scientific community as a separate study at a later time point.

(6) Why was the STED data analysis confined to the same optical section, and not to max. intensity z-projections? How many and which optical sections were considered for each active zone? What were the criteria for choosing the optical sections? Was synapse orientation considered for the nearest neighbor Cac - Brp cluster distance analysis? How do the nearest-neighbor distances compare between "planar" and "side-view" Brp puncta?

Maximum intensity z-projections would be imprecise because they can artificially suggest close proximity of label that is close by in x and y but far away in z. Therefore, the analysis was executed in xy-direction of various planes of entire 3D image stacks. We considered active zones of different orientations (Figs. 5C, D) to account for all planes. In fact, we searched the entire z-stacks until we found active zones of all orientations within the same boutons, as shown in figures 5C1-C6. The same active zone orientations were analyzed for all exon-out mutants with cac localization in active zones. The distance between cac and brp did not change if viewed from the side or any other orientation. We now explain this in more clarity in the results text on page 9, lines 23/24.

(7) Cac clusters localize to the Brp center (e.g., Liu et al., 2011). They conclude that Cav2 localization within Brp is not affected in the cac variants (p. 8). However, their analysis is not informative regarding a potential offset between the central cac cluster and the Brp "ring". Did they/could they analyze cac localization with regard to Brp ring center localization of planar synapses, as well as Brp-ring dimensions?

In the top views (planar) we did not find any clear offset in cac orientation to brp between genotypes. In such planar synapses (top views, Fig. 5D, left row) we did not find any difference in Brp ring dimensions. We did not quantify brp ring dimensions rigorously, because this study focusses on cac splice isoform-specific localization and function. Possible effects of different cac isoforms on brp-ring dimensions or other aspects of scaffold structure are not central to our study, in particular given that brp puncta are clearly present even if cac is absent from the synapse (Fig. 3A), indicating that cac is not instructive for the formation of the brp scaffold.

(8) Given the accelerated PSC decay/ decreased half width in dI-IIA (Fig. 5Q), I recommend reporting PSC charge in Figure 3, and PPR charge in Figures 5A-D. The charge-based PPRs of dI-IIA mutants likely resemble WT more closely than the amplitude-based PPR. In addition, miniature PSC decay kinetics should be reported, as they may contribute to altered decay kinetics. How could faster cac inactivation kinetics in response to single AP stimulation result in a decreased PSC half-width? Is there any evidence for an effect of calcium current inactivation on PSC kinetics? On a similar note, is there any evidence that AP waveform changes accelerate PSC kinetics? PSC decay kinetics are mainly determined by GluR decay kinetics/desensitization. The arguments supporting the role of cac splice isoforms in PSC kinetics outlined in the discussion section are not convincing and should be revised.

We agree that reporting charge in figure 3 is informative and do so in the revised text. Since the result (no significant difference in the PSCs between between CS, cac^GFP^, ^ΔI-IIA^, and transheterozygous *I-IIA/I-IIB*, but significantly smaller values in *ΔI-IIB*) remained unchanged no matter whether charge or amplitude were analyzed, we decided to leave the figure as is and report the additional analysis in the text (page 8, lines 40 to 42). This way, both types of analysis are reported. Please note that EPSC amplitude is slightly but not significantly increased upon excision of I-IIA (Fig. 4J), whereas EPSC half amplitude width is significantly smaller (Fig. 5Q, now revised Fig 6R). Together, a tendency of increased EPSC amplitudes and smaller half amplitude width result in statistically insignificant changes in EPSC in ∆I-IIA (now discussed on page 15, lines 37 to 40). We also understand the reviewer’s concern attributing altered EPSC kinetics to presynaptic cac channel properties. We have toned down our interpretation in the discussion and list possible alterations in presynaptic AP shape or cac channel kinetics as alternative explanations (not conclusions; see revised discussion on page 15, line 40 to page 16, line 2). Moreover, we have quantified postsynaptic GluRIIA abundance to test whether altered PSC kinetics are caused by altered GluRIIA expression. In our opinion, the latter is more instructive than mini decay kinetic analysis because this depends strongly on the distance of the recording electrode to the actual site of transmission in these large muscle cells. Although we find no difference in GluRIIA expression levels we now clearly state that we cannot exclude other changes in GluR receptor fields, which of course, could also explain altered PSC kinetics. We have updated the discussion on page 16, lines 2/3 accordingly.

(9) Paired-pulse ratios (PPRs): On how many sweeps are the PPRs based? In which sequence were the intervals applied? Are PPR values based on the average of the second over the first PSC amplitudes of all sweeps, or on the PPRs of each sweep and then averaged? The latter calculation may result in spurious facilitation, and thus to the large PPRs seen in dI-IIB mutants (Kim & Alger, 2001; doi: 10.1523/JNEUROSCI.21-2409608.2001).

We agree that the PP protocol and analyses had to be described more precisely in the methods and have done so on page 23, lines 31 to 37 in the methods. Mean PPR values are based on the PPRs of each sweep and then averaged. We are aware of the study of Kim and Alger 2001 and have re-analyzed the PP data in both ways outlined by the reviewer. We get identical results with either analyses method. Spurious facilitation is thus not an issue in our data. We now explain this in the methods section along with the PPR protocol. The large spread seen in dI-IIB is indeed caused by reduced calcium influx into active zones with fewer channels, as anticipated by the reviewer (see next point).

(10) Could the dI-IIB phenotype be simply explained by a decrease in channel number/ release probability? To test this, I propose investigating PPRs and short-term dynamics during train stimulation at lower extracellular Ca2+ concentration in WT. The Ca2+ concentration could be titrated such that the first PSC amplitude is similar between WT and dI-IIB mutants. This experiment would test if the increased PPR/depression variability is a secondary consequence of a decrease in Ca2+ influx, or specific to the splice isoform.

In fact, the interpretation that decreased PSC amplitude upon I-IIB excision is caused mainly by reduced channel number is precisely our interpretation (see discussion page 14, last paragraph to page 15, first paragraph in the original submission, now page 16, second paragraph paragraph). In addition, we are grateful for the reviewer’s suggestion to triturate the external calcium such that the first PSC amplitude in matches in ∆I-IIB and control. This experiment tests whether altered short term plasticity is solely a function of altered channel number or whether additional causes, such as altered channel properties, also play into this. We triturated the first pulse amplitude in *∆I-IIB* to match control and find that paired pulse ratio and the variance thereof are not different anymore. Therefore, the differences observed in identical external calcium can be fully explained by altered channel numbers. This additional dataset is shown in the revised figures 6D and E and referred to in the results section on page 10, lines 14 to 25 and the discussion on page16, lines 36 to 38.

(11) How were the depression kinetics analyzed? How many trains were used for each cell, and how do the tau values depend on the first PSC amplitude? Time constants in the range of a few (5-10) milliseconds are not informative for train stimulations with a frequency of 1 or 10 Hz (the unit is missing in Figure 5H). Also, the data shown in Figures 5E-K suggest slower time constants than 5-10 ms. Together, are the data indeed consistent with the idea that dIIIB does not only affect cac channel number, but also PPR/depression variability (p. 9)?

For each animal the amplitudes of all subsequent PSCs in each train were plotted over time and fitted with a single exponential. For depression at 1 and 10 Hz, we used one train per animal, and 5-6 animals per genotype (as reflected in the data points in Figs. 6I, M). This is now explained in more detail in the revised methods section (page 23, lines 39 to 41). The tau values are not affected by the amplitude of the first PSC. First, we carefully re-fitted new and previously presented depression data and find that the taus for depression at low stimulation frequencies (1 and 10Hz) are not affected by exon excisions at the I-II site. We thank the reviewer for detecting our error in units and tau values in the previous figure panels 5H and L (this has now been corrected in the revised figure panels 6I and M). Given that PSC amplitude upon I-IIB excision is significantly smaller than in controls and following I-IIA excision, we suspected that the time course of depression at low stimulation frequency is not significantly affected by the amount of calcium influx during the first PSC. To further test this, we followed the reviewer ’s suggestion and re-measured depression at 1 and 10 Hz for cac-GFP controls and for delta I-IIB in a higher external calcium concentration (1.8 mM), so that the first PSC was increased in amplitude in both genotypes (1.8 mM external calcium triturates the PSC amplitude in delta I-IIB to match that of controls measured in 0.5 mM external calcium, see revised Figs. 6H, L). Neither in control, nor in delta I-IIB did this affect the time course of synaptic depression (see revised Figs. 6I, M). This indicates that at low stimulation frequencies (1 and 10Hz) the time course of depression is not affected by mean quantal content. This is consistent with the paired pulse ratio at 100 ms interpulse interval shown in figures 6A-D. However, for synaptic depression at 1 Hz stimulation the variability of the data is higher for delta I-IIB (independent of external calcium concentration, see rev. Fig. 6I), which might also be due to reduced channel number in this genotype. Taken together, the data are in line with the idea that altered cac channel numbers in active zones are sufficient to explain all effects that we observe upon I-IIB excision on PPRs and synaptic depression at low stimulation frequencies. This is now clarified in the revised text on page 12, lines 3 to 7.

(12) The GFP-tagged I-IIA and mEOS4b-tagged I-IIB cac puncta shown in Figure 6N appear larger than the Brp puncta. Endogenously tagged cac puncta are typically smaller than Brp puncta (Gratz et al., 2019). Also, the I-IIA and I-IIB fluorescence sometimes appear to be partially non-overlapping. First, I suggest adding panels that show all three channels merged. Second, could they analyze the area and area overlap of I-IIA and I-IIB with regard to each other and to Brp, and compare it to cac-GFP? Any speculation as to how the different tags could affect localization? Finally, I recommend moving the dI-IIA and dI-IIB localization data shown in Figure 6N to an earlier figure (Figure 1 or Figure 3).

We now show panels with the two I-II cac isoforms merged in the revised figure 7H (previously 6N). We also tested merging all three labels as suggested, but found this not instructive for the reader. We thank the reviewer for pointing out that the Brp puncta appeared smaller than the cac puncta in some panels. We carefully went through the data and found that the Brp puncta are not systematically smaller than the cac puncta. Please note that punctum size can appear quite differently, depending on different staining qualities as well as different laser intensities and different point spread in different imaging channels. The purpose of this figure was not to analyze punctum size and labeling intensity, but instead, to demonstrate that I-IIA and I-IIB are both present in most active zones, but some active zones show only I-IIB labeling, as quantified in figure 7I. We did not follow the suggestion to conduct additional co-localization analyses and compare it with cac-GFP controls, because Pearson co-localization coefficients for cac-GFP and all exon-out variants analyzed, including delta I-IIA and delta I-IIB are presented in the revised figure 4D. Moreover, delta I-IIA and delta I-IIB show similar Manders 1 and 2 co-localization coefficients with Brp (see Figs. 4E, F). We do not want to speculate whether the different tags have any effect on localization precision. Artificial differences in localization precision can also be suggested by different antibodies, but we know from our STED analyses with identical tags and antibodies for all isoforms that I-IIA and I-IIB co-localize identically with Brp (see Figs. 5A-E). Finally, we prefer to not move the figure because we believe it is informative to show our finding that active zones usually contain both splice I-II variants together with the finding that only I-IIB is required for PHP.

**Recommendations for the authors:**

**Reviewing Editor Comments:**
We thank you for your submission. All three reviewers urge caution in interpreting the S4 splice variant playing a role specifically in Cac localization, as opposed to just leading to instability and degradation. There are other issues with the electrophysiological experiments, a need for improved imaging and analyses, and some areas of interpretation detailed in the reviews.

We agree that additional data was required to conclude that IS4 splicing plays a specific role in cac channel localization and is not just leading to channel instability and degradation. As outlined in detail in our response to reviewer 1, comment 1, we conducted several sets of experiments to support our interpretation. First, electrophysiological experiments show that upon removal of IS4B, which eliminates synaptic transmission at the larval NMJ and cac positive label in presynaptic active zones, somatodendritic cac current is reliably recorded (new data in revised figure 3C). This is not in line with a channel instability or degradation effect, but instead with IS4B containing isoforms being required and sufficient for evoked release from NMJ motor terminals, whereas IS4A isoforms are not sufficient for evoked release from axon terminals, but IS4A isoforms alone can mediate a distinct component of somatodendritic calcium current. Second, immunohostochemical analyses reveal that IS4A, which is not present in NMJ presynaptic active zones, is expressed sparsely, but in reproducible patterns in the larval brain lobes and in specific regions of the anterior VNC parts (new supplementary figure 1). Again, the absence of a IS4A-containing cac isoform from presynaptic active zones but their simultaneous presence in other parts of the nervous system is in accord with isoform specific localization, but not with general channel isoform instability. Third, enlargements of NMJ boutons with brp positive presynaptic active zones confirm the absence of IS4A and the presence of IS4B in active zones (these enlargements are now shown in the revised figures 2A-C, 3A, and 4A-C). Fourth, as suggested we have quantified the Pearson co-localization of IS4 isoforms with Brp in presynaptic active zones (revised Fig. 2D). This confirms quantitatively similar co-localization of IS4B and control with Brp, but no co-localization of IS4A with Brp. In fact, the labeling intensity of IS4A in presynaptic active zones is quantitatively not significantly different from background, no IS4A label is detected anywhere in the axon terminals at the NMJ, but we find IS4 label in the CNS. Together, these data strongly support our interpretation that the IS4 splice site plays a distinct role in cac channel localization. Figure legends as well as results and discussion section have been modified accordingly (the respective page and line numbers are listed in our-point-by-point responses).

In addition, we have carefully addressed all other public comments as well as all other recommendations for authors by providing multiple new data sets, new image analyses, and revising text. Addressing the insightful comments of all three reviewers and the reviewing editor has greatly helped to make the manuscript better.

**Reviewer #1 (Recommendations For The Authors):**
The conclusion that the IS4B exon controls Cac localization to active zones versus simply being required for channel abundance is not well supported. The authors need to either mention both possibilities or provide stronger support for the active zone localization model if they want to emphasize this point.

We agree and have included several additional data sets as outlined in our response to point 1 of reviewer 1 and to the reviewing editor (see above). These new data strongly support our interpretation that the IS4B exon controls Cac localization to active zones and is not simply required for channel abundance. The additions to the figures and accompanying text (including the respective figure panel, page, and line numbers) are listed in the point-bypoint responses to the reviewers’ public suggestions.

Figure 2C staining for Cac localization in the delta 4B line is difficult to compare to the others, as the background staining is so high (muscles are green for example). As such, it is hard to determine whether the arrows in C are just background.

We had over-emphasized the green label to show that there really is no cacophony label in active zones. However, we agree that this hampered image interpretation. Thus, we have adjusted brightness such that it matches the other genotypes (see new figure panel 2C, and figure 3A, bottom). Revising the figure as suggested by the reviewer shows much more clearly that IS4B puncta are detected exclusively in presynaptic active zones, whereas IS4A channels are not detectable in active zones or anywhere else in the axon terminal boutons. Quantification of IS4A label in brp positive active zones confirms that labeling intensity is not significantly above background (page 6, lines 29 to 31 and page 7, lines 19 to 21). Therefore, IS4A is not detectable in active zones at the NMJ.

It seems more likely that the removal of the 4B exon simply destabilizes the protein and causes it to be degraded (as suggested by the Western), rather than mislocalizing it away from active zones. It's hard to imagine how some residue changes in the S4 voltage sensor would control active zone localization to begin with. The authors should note that the alternative explanation is that the protein is just degraded when the 4B exon is removed.

Based on additional data and analyses, we disagree with the interpretation that removal of IS4B disrupts protein integrity and present multiple lines of evidence that support sparse expression of IS4A channels (ΔIS4B). As outlined in our response to reviewer 1 and to the reviewing editor, we show (1) in new immunohistochemical stainings (new supplementary figure 1) that upon removal of IS4B, sparse label is detectable in the VNC and the brain lobes (for detail see above). (2) In our new figure 3C, we show cacophony-mediated somatodendritic calcium currents recorded from adult flight motoneurons in a control situation and upon removal of IS4B that leaves only IS4A channels. This clearly demonstrates that IS4A underlies a substantial component of the HVA somatodendritic calcium current, although it is absence from axon terminals. This is in line with isoform specific functions at different locations, but not with IS4A instability/degradation. (3) We do not agree with the reviewer’s interpretation of the Western Blot data in figure 1E (formerly figure 1D). Together with our immunohistochemical data that show sparse cacophony IS4A expression, we think that the faint band upon removal of IS4B in a heterozygous background (that reduces labeled channels even further) reflects the sparseness of IS4A expression. This sparseness is not due to channel instability, but to IS4A functions that are less abundant than the ubiquitously expressed cac^IS4B^ channels at presynaptic active zones of fast chemical synapses (see page 15, lines 24 to 29).

If they really want to claim the 4B exon governs active zone localization, much higher quality imaging is required (with enlarged views of individual boutons and their AZs, rather than the low-quality full NMJ imaging provided). Similarly, higher resolution imaging of Cac localization at Muscle 12 (Figure 2H) boutons would be very useful, as the current images are blurry and hard to interpret. Figure 6N shows beautiful high-resolution Cac and Brp imaging in single boutons for the I-II exon manipulations - the authors should do the same for the 4B line. For all immuno in Figure 2, it is important to quantify Cac intensity as well. There is no quantification provided, just a sample image. The authors should provide quantification as they do for the delta I-II exons in Figure 3.

We did as suggested and added figure panels to figure 2A-C and to new figures 3A (formerly part of figure 2 and 4A-C formerly figure 3) showing magnified label at the NMJ AZs to better judge on cacophony expression after exon excision. These data are now referred to in the results section on page 6, lines 22 to 24, page 7, lines 18 to 21 and page 8, lines 17/18.

As suggested, we now also provide quantification of co-localization with brp puncta as Pearson’s correlation coefficient for control, IS4B, and IS4A in the new figure panel 2D (text on page 6, lines 34 to 38). This further underscores control-like active zone localization of IS4B but no significant active zone localization of IS4A. As suggested, we quantified now also the intensity of IS4B label in active zones, and it was not different from control (see revised figure 4H and text on page 8, lines 38/39). We did not quantify the intensity of IS4A label, because it was not over background (text, page 6, lines 30/31).

**Reviewer #2 (Recommendations For The Authors):**
(1a) Questions about the engineered Cac splice isoform alleles:The authors using CRISPR gene editing to selectively remove the entire alternatively spliced exons of interest. Do the authors know what happens to the cac transcript with the deleted exon? Is the deleted exon just skipped and spliced to the next exon? Or does the transcript instead undergo nonsense-mediated decay?

We do not believe that there is nonsense mediated mRNA decay, because for all exon excisions the respective mRNA and protein are made. Protein has been detected on the level of Western blotting and immunocytochemistry. Therefore, we are certain that the mRNA is viable for each exon excision (and we have confirmed this for low abundance cac protein isoforms by rt-PCR), but only subsets of cac isoforms can be made from mRNAs that are lacking specific exons. However, we can not make any statements as to whether the lack of specific protein isoforms exerts feedback on mRNA stability, the rate of transcription and translation, or other unknown effects.

(1b) While it is clear that the IS4 exons encode part of the voltage sensor in the first repeat, are there studies in *Drosophila* to support the putative Ca-beta and G-protein beta-gamma binding sites in the I-II loop? Or are these inferred from Mammalian studies?

To the best of our knowledge, there are no studies in *Drosophila* that unambiguously show Caβ and Gβγ binding sites in the I-II loop of cacophony. However, sequence analysis strongly suggests that I-IIB contains both, a Caβ as well as a Gβγ binding site (AID: α-interacting domain) because the binding motif QXXER is present. In mouse Cav2.1 and Ca_v_2.2 channels the sequence is QQIER, while in *Drosophila* cacophony I-IIB it is QQLER. In the alternative IIIA, this motif is not present, strongly suggesting that G_βγ_ subunits cannot interact at the AID. However, as already suggested by Smith et al. (1998), based on sequence analysis, Ca_β_ should still be able to bind, although possibly with a lower affinity. We agree that this information should be given to the reader and have revised the text accordingly on page 5, lines 9 to 17.

(1c) The authors assert that splicing of Cav2/cac in flies is a means to encode diversity, as mammals obviously have 4 Cav2 genes vs 1 in flies. However, as the authors likely know, mammalian Cav2 channels also have various splice isoforms encoded in each of the 4 Cav2 genes. The authors should discuss in more detail what is known about the splicing of individual mammalian Cav2 channels and whether there are any homologous properties in mammalian channels controlled by alternative splicing.

We agree and now provide a more comprehensive discussion of vertebrate Ca_v_2 splicing and its impact on channel function. In line to what we report in *Drosophila*, properties like G_βγ_ binding and activation voltage can also be affected by alternative splicing in vertebrate Ca_v_2 channel, through the exon patterns are quite different from *Drosophila*. We integrated this part on page 14, first paragraph in the revised discussion. The respective text is below for the reviewer’s convenience:

“However, alternative splicing increases functional diversity also in mammalian Ca_v_2 channels. Although the mutually exclusive splice site in the S4 segment of the first homologous repeat (IS4) is not present in vertebrate Cav channels, alternative splicing in the extracellular linker region between S3 and S4 is at a position to potentially change voltage sensor properties (Bezanilla 2002). Alternative splice sites in rat Ca_v_2.1 exon 24 (homologous repeat III) and in exon 31 (homologous repeat IV) within the S3-S4 loop modulate channel pharmacology, such as differences in the sensitivity of Ca_v_2.1 to Agatoxin. Alternative splicing is thus a potential cause for the different pharmacological profiles of P- and Q-channels (both Ca_v_2.1; Bourinet et al. 1999). Moreover, the intracellular loop connecting homologous repeats I and II is encoded by 3-5 exons and provides strong interaction with G_βγ_-subunits (Herlitze et al. 1996). In Ca_v_2.1 channels, binding to G_βγ_ subunits is potentially modulated by alternative splicing of exon 10 (Bourinet et al. 1999). Moreover, whole cell currents of splice forms α1A-a (no Valine at position 421) and α1A-b (with Valine) represent alternative variants for the I-II intracellular loop in rat Ca_v_2.1 and Ca_v_2.2 channels. While α1A-a exhibits fast inactivation and more negative activation, α1A-b has delayed inactivation and a positive shift in the IV-curve (Bourinet et al. 1999). This is phenotypically similar to what we find for the mutually exclusive exons at the IS4 site, in which IS4B mediates high voltage activated cacophony currents while IS4A channels activate at more negative potentials and show transient current (Fig. 3; see also Ryglewski et al. 2012). Furthermore, altered Ca_β_ interaction have been shown for splice isoforms in loop III (Bourinet et al. 1999), similar to what we suspect for the I-II site in cacophony. Finally, in mammalian VGCCs, the C-terminus presents a large splicing hub affecting channel function as well as coupling distance to other proteins. Taken together, Ca_v_2 channel diversity is greatly enhanced by alternative splicing also in vertebrates, but the specific two mutually exclusive exon pairs investigated here are not present in vertebrate *Cav2* genes.”

(1d) In Figure 1, it would be helpful to see the entire cac genomic locus with all introns/exons and the 4 specific exons targeted for deletion.

We agree and have changed figure 1 accordingly.

(2a) Cav2.IS4B deletion alleles:More work is necessary to explain the localization of Cac controlled by the IS4B exon. First, can the authors determine whether actual Cac channels are present at NMJ boutons? The authors seem to indicate that in the IS4B deletion mutants, some Cac (GFP) signal remains in a diffuse pattern across NMJ boutons. However, from the imaging of wild-type Cac-GFP (and previous studies), there is no Cac signal outside of active zones defined by the BRP signal. It would benefit the study to (a) take additional, higher resolution images of the remaining Cac signal at NMJs in IS4B deletion mutants, and (b) comment on whether the apparent remaining signal in these mutants is only observed in the absence of IS4Bcontaining Cac channels, or if the IS4A-positive channels are normally observed (but perhaps mis-localized?).

We have conducted additional analyses to show convincingly that IS4A channels (that remain upon IS4B deletion) are absent from presynaptic active zone. Please see also responses to reviewers 1 and 3. By adjusting the background values in of CLSM images to identical values in control, delta IS4A, and delta IS4B, as well as by providing selective enlargements as suggested, the figure panels 2C, Ci and 3A now show much clearer, that upon deletion of IS4B no cac label remains in active zones or anywhere else in the axon terminal boutons (see text on page 6, lines 22 to 24). This is further confirmed by quantification showing the in IS4B mutants cac labeling intensity in active zones is not above background (see text on page 6, lines 27 to 31). We never intended to indicate that there was cac signal outside of active zones defined by the brp signal, and we now carefully went through the text to not indicate this possibility unintentionally anywhere in the manuscript.

(2b) Do the authors know whether any presynaptic Ca2+ influx is contributed by IS4Apositive Cac channels at boutons, given the potential diffuse localization? There are various approaches for doing presynaptic Ca2+ imaging that could provide insight into this question.

We agree that this is an interesting question. However, based on the revisions made, we now show with more clarity that IS4A channels are absent from the presynaptic terminal at the NMJ. IS4A labeling intensities within active zones and anywhere else in the axon terminals are not different from background (see text on page 6, lines 27 to 31 and revised Figs. 2C, Ci, and 3A with new selective enlargements in response to comments of both other reviewers). This is in line with our finding that evoked synaptic transmission from NMJ axon terminals to muscle cells is mostly absent upon excision of IS4B (see Fig. 3B). The very small amplitude EPSC (below 5 % of the normal amplitude of evoked EPSCs) that can still be recorded in the absence of *IS4B* is similar to what is observed in *cac null* mutant junctions and is mediated by calcium influx through another voltage gated calcium channels, a Ca_v_1 homolog named Dmca1D, as we have previously published (Krick et al., 2021, PNAS 118(28):e2106621118. Gathering additional support for the absence of IS4A from presynaptic terminals by calcium imaging experiments would suffer significantly from the presence of additional types of VGCCs in presynaptic terminals (for sure Dmca1D Krick et al., 2021) and potentially also the (Ca_v_3 homolog DmαG or Dm-α1T)). Such experiments would require mosaic null mutants for *cac* and *DmαG* channels in a mosaic *IS4B* excision mutant, which, if feasible at all, would be very hard and time consuming to generate. In the light of the additional clarification that IS4A is not located in NMJ axon terminal boutons, as shown by additional labeling intensity analysis, revised figures with selective enlargement, and revised text, we feel confident to state that IS4A is not sufficient for evoked SV release.

(2c) Mechanistically, how are amino acid changes in one of the voltage sensing domains in Cac related to trafficking/stabilization/localization of Cac to AZs?

This is an exciting question that has occupied our discussions a lot. Some sorting mechanism must exist that recognizes the correct protein isoforms, just as sorting and transport mechanisms exist that transport other synaptic proteins to the synapse. We do not think that the few amino acid changes in the voltage sensor are directly involved in protein targeting. We rather believe that the cacophony variants that happen to contain this specific voltage sensor are selected for transport out to the synapse. There are possibilities to achieve this cell biological, but we have not further addressed potential mechanisms because we do not want enter the realms of speculation.

(3) How are auxiliary subunits impacted in the Cac isoform mutants?Recent work by Kate O'Connor-Giles has shown that both Stj and Ca-Beta subunits localize to active zones along with Cac at the *Drosophila* NMJ. Endogenously tagged Stj and CaBeta alleles are now available, so it would be of interest to determine if Stj and particular Cabeta levels or localization change in the various Cac isoform alleles. This would be particularly interesting given the putative binding site for Ca-beta encoded in the I-II linker.

We agree that the synthesis of the work of Kate O'Connor-Giles group and our study open up new avenues to explore exciting hypotheses about differential coupling of specific cacophony splice isoforms with distinct accessory proteins such as Caβ and α_2_δ subunits. However, this requires numerous full sets of additional experiments and is beyond the scope of this study.

(4a) Interpretation of short-term plasticity in the I-IIB exon deletion:The changes in short-term plasticity presented in Figure 5 are interpreted as an additional phenotype due to the loss of the I-IIB exon, but it seems this might be entirely explained simply due to the reduced Cac levels. Reduced Cac levels at active zones will obviously reduce Ca2+ influx and neurotransmitter release. This may be really the only phenotype/function of the I-IIB exon. Hence, to determine whether loss of the I-IIB exon encodes any functions in short-term plasticity, separate from reduced Cac levels, the authors should compare short-term plasticity in I-IIB loss alleles compared to wild type with starting EPSC amplitudes are equal (for example by reducing extracellular Ca2+ levels in wild type to achieve the same levels at in Cac I-IIB exon deleted alleles). Reduced release probability, simply by reduced Ca2+ influx (either by reduced Cac abundance or extracellular Ca2+) should result in more variability in transmission, so I am not sure there is any particular function of the I-IIB exon in maintaining transmission variability beyond controlling Cac abundance at active zones.

For two reasons we are particularly grateful for this comment. First, it shows us that we needed to explain much clearer that our interpretation is that changes in paired pulse ratios (PPRs) and in depression at low stimulation frequencies are a causal consequence of lower channel numbers upon I-IIB exon deletion, precisely as pointed out by the reviewer. We have carefully revised the text accordingly on page 10, lines 14-25, page 11, lines 3-7 and 22-28; page 16, lines 36-38. Second, the experiment suggested by the reviewer is superb to provide additional evidence that the cause of altered PPRs is in fact reduced channel number, but not altered channel properties. Accordingly, we have conducted additional TEVC recordings in elevated external calcium (1.8 mM) so that the single PSC amplitudes in *I-IIB* excision animals match those of controls in 0.5 mM extracellular calcium. This makes the amplitudes and the variance of PPR for all interpulse intervals tested control-like (see revised Figs. 6D, E). This strongly indicates that differences observed in PPRs as well as the variance thereof were caused by the amount of calcium influx during the first EPSC, and thus by different channel numbers in active zones.

(4b) Another point about the data in Figure 5: If "behaviorally relevant" motor neuron stimulation and recordings are the goal, the authors should also record under physiological Ca2+ conditions (1.8 mM), rather than the highly reduced Ca2+ levels (0.5 mM) they are using in their protocols.

Although we doubt that the effective extracellular calcium concentration that determines the electromotoric force for calcium to enter the ensheathed motoneuron terminals *in vivo* during crawling is known, we followed the reviewer’s suggestion partly and have repeated the high frequency stimulation trains for *ΔI-IIB* in 1.8 mM calcium. As for short-term plasticity this brings the charge conducted to values as observed in control and in *ΔI-IIA* in 0.5 mM calcium. Therefore, all difference observed in previous figure 5 (now revised figure 6) can be accounted to different channel numbers in presynaptic active zones. This is now explained on page 11, lines 19-28. For controls recordings at high frequency stimulation in higher external calcium (e.g. 2 mM) have previously been published and show significant synaptic depression (e.g. Krick et al., 2021, PNAS). Given that in the exon out variants we do not expect any differences except from those caused by different channel numbers, we did not repeat these experiments for control and *ΔI-IIA*.

(5a) Mechanism of Cac's role in PHP :As the authors likely know, mutations in Cac were previously reported to disrupt PHP expression (see Frank et al., 2006 Neuron). Inexplicably, this finding and publication were not cited anywhere in this manuscript (this paper should also be cited when introducing PhTx, as it was the first to characterize PhTx as a means of acutely inducing PHP). In the Frank et al. paper (and in several subsequent studies), PHP was shown to be blocked in mutations in Cac, namely the CacS allele. This allele, like the I-IIB excision allele, reduces baseline transmission presumably due to reduced Ca2+ influx through Cac. The authors should at a minimum discuss these previous findings and how they relate to what they find in Figure 6 regarding the block in PHP in the Cac I-IIB excision allele.

We thank the reviewer for pointing this out and apologize for this oversight. We agree that it is imperative to cite the 2006 paper by Frank et al. when introducing PhTx mediated PHP as well as when discussing cac the effects of cac mutants on PHP together with other published work. We have revised the text accordingly on page 12, lines 9-11 and 21-23 and on page 17, lines 29-33.

In terms of data presentation in Fig. 6, as is typical in the field, the authors should normalize their mEPSC/QC data as a percentage of baseline (+PhTx/-PhTx). This makes it easier to see the reduction in mEPSC values (the "homeostatic pressure" on the system) and then the homeostatic enhancement in QC. Similarly, in Fig. 6M, the authors should show both mEPSC and QC as a percentage of baseline (wild type or non-GluRIIA mutant background).

We agree and have changed figure presentation accordingly. Figure 7 (formerly figure 6) was updated as was the accompanying results text on page 12, lines 23-40.

(6) Cac I-IIA and I-IIB excision allele colocalization at AZs:These are very nice and important experiments shown in Figures 6N and O, which I suggest the authors consider analyzing in further detail. Most significantly:(6i) The authors nicely show that most AZs have a mix of both Cac IIA and IIB isoforms. Using simple intensity analysis, can the authors say anything about whether there is a consistent stoichiometric ratio of IIA vs IIB at single AZs? It is difficult to extract actual numbers of IIA vs IIB at individual AZs without having both isoforms labeled mEOS4b, but as a rough estimate can the authors say whether the immunofluorescence intensity of IIA:IIB is similar across each AZ? Or is there broad heterogeneity, with some AZs having low vs high ratios of each isoform (as the authors suggest across proximal to distal NMJ AZs)?

We agree and have conducted experiments and analyses to provide these data. We measured the cac puncta fluorescence intensities for heterozygous cac^sfGFP^/cac, cacIIIA^sfGFP^/cacI-IIB, and cacI-IIB^sfGFP^/cacI-IIA animals. We preferred this strategy, because intensity was always measured from cac puncta with the same GFP tag. Next, we normalized all values to the intensities obtained in active zones from heterozygous cac^sfGFP^/cac controls and then plotted the intensities of I-IIA versus I-IIB containing active zones side by side. Across junctions and animals, we find a consistent ratio 2:1 in the relative intensities of I-IIB and I-IIA, thus indicating on average roughly twice as many I-IIB as compared to I-IIA channels across active zones. This is consistent with the counts in our STED analysis (see Fig. 5F). These new data are shown in the new figure panel 7J and referred to on page 13, lines 10-16 in the revised text.

(6ii) Intensity analysis of Cac IIA vs IIB after PHP: Previous studies have shown Cac abundance increases at NMJ AZs after PHP. Can the authors determine whether both Cac IIA vs IIB isoforms increase after PHP or whether just one isoform is targeted for this enhancement?

We already show that PHP is not possible in the absence of I-IIB channels (see figure 7). However, we agree that it is an interesting question to test whether I-IIA channel are added in the presence of I-IIB channels during PHP, but we consider this a detail beyond the scope of this study.

Minor points:(1) Including line numbers in the manuscript would help to make reviewing easier.

We agree and now provide line numbers.

(2) Several typos (abstract "The By contrast", etc).

We carefully double checked for typos.

(3) Throughout the manuscript, the authors refer to Cac alleles and channels as "Cav2", which is unconventional in the field. Unless there is a compelling reason to deviate, I suggest the authors stick to referring to "Cac" (i.e. cacdIS4B, etc) rather than Cav2. The authors make clear in the introduction that Cac is the sole fly Cav2 channel, so there shouldn't be a need to constantly reinforce that cac=Cav2.

We agree and have changed all fly Ca_v_2 reference to cac.

(4) In some figures/text the authors use "PSC" to refer to "postsynaptic current", while in others (i.e. Figure 6) they switch to the more conventional terms of mEPSC or EPSC. I suggest the authors stick to a common convention (mEPSC and EPSC).

We have changed PSC to EPSC throughout.

**Reviewer #3 (Recommendations For The Authors):**
(1) The abstract could focus more on the results at the expense of the background.

We agree and have deleted the second introductory background sentence and added information on PPRs and depression during low frequency stimulation.

(2) What does "strict" active zone localization refer to? Could they please define the term strict?

Strict active zone localization means that cac puncta are detected in active zones but no cac label above background is found anywhere else throughout the presynaptic terminal, now defined on page 6, lines 27-29.

(3) Single boutons/zoomed versions of the confocal images shown in Figures 2A-C, 2H, and 3A-C would be very helpful.

We have provided these panels as suggested (see above and revised figures 2-4). Figure 3 is now figure 4.

(4) The authors cite Ghelani et al. (2023) for increased cac levels during homeostatic plasticity. I recommend citing earlier work making similar observations (Gratz et al., 2019; DOI: 10.1523/JNEUROSCI.3068-18.2019), and linking them to increased presynaptic calcium influx (Müller & Davis, 2012; DOI: 10.1016/j.cub.2012.04.018).

We agree and have added Gratz et al. 2019 and Davis and Müller 2012 to the results section on page 12, lines 17/18 and lines 21-23, in the discussion on page 17, lines 29-33.

(5) The data shown in Figure 3 does not directly support the conclusion of altered release probability in dI-IIB. I therefore suggest changing the legend's title.

We have reworded to “Excisions at the I-II exon do not affect active zone cacophony localization but can alter cacsfGFP label intensity in active zones and PSC amplitude” as this is reflecting the data shown in the figure panels more directly.

(6) It would be helpful to specify "adult flight muscle" in Figure 2J.

We agree that it is helpful to specify in the figure (now revised figure 3C) that the voltage clamp recordings of somatodendritic calcium current were conducted in adult flight motoneurons and have revised the headline of figure panel 3C and the legend accordingly. Please note, these are not muscle cells but central neurons.

(7) Do dIS4B/Cav2null MNs indeed show an inward or outward current at -90 to -70 mV/-40 and -50 mV, or is this an analysis artifact?

No, this is due to baseline fluctuations as typical for voltage clamp in central neurons with more than 6000 µm dendritic length and more than 4000 dendritic branches.

(8) Loss of several presynaptic proteins, including Brp (Kittel et al., 2006), and RBP (Liu et al., 2011), induce changes in GluR field size (without apparent changes in miniature amplitude). The statement regarding the Cav2 isoform and possible effects on GluR number (p. 8) should be revised accordingly.

We understand and have done two things. First, we measured the intensity of GluRIIA immunolabel in *ΔI-IIA*, *ΔI-IIB*, and controls and found no differences. Second, we reworded the statement. It now reads on page 9, lines 1-6: “It seems unlikely that presynaptic cac channel isoform type affects glutamate receptor types or numbers, because the amplitude of spontaneous miniature postsynaptic currents (mEPSCs, Fig. 4K) and the labeling intensity of postsynaptic GluRIIA receptors are not significantly different between controls, I-IIA, and I-IIB junctions (see suppl. Fig. 2, p = 0.48, ordinary one-way ANOVA, mean and SD intensity values are 61.0 ± 6.9 (control), 55.8 ± 8.5 (*∆I-IIA*), 61.1 ± 17.3 (*∆I-IIB*)). However, we cannot exclude altered GluRIIB numbers and have not quantified GluR receptor field sizes.”

(9) The statement relating miniature frequency to RRP size is unclear (p. 8). Is there any evidence for a correlation between miniature frequency to RRP size? Could the authors please clarify?

We agree that this statement requires caution. Although there is some published evidence for a correlation of RRP size and mini frequency (Neuron, 2009 61(3):412-24. doi: 10.1016/j.neuron.2008.12.029 and Journal of Neuroscience 44 (18) e1253232024; doi: 10.1523/JNEUROSCI.1253-23.2024), which we now refer to on page 9, it is not clear whether this is true for all synapses and how linear such a relationship may be. Therefore, we have revised the text on page 9, lines 6-9. It now reads: “Similarly, the frequency of miniature postsynaptic currents (mEPSCs) remains unaltered. Since mEPSCs frequency has been related to RRP size at some synapses (Pan et al., 2009; Ralowicz et al., 2024) this indicates unaltered RRP size upon I-IIB excision, but we have not directly measured RRP size.”

(10) Please define the "strict top view" of synapses (p. 8).

Top view is what this reviewer referred to as “planar view” in the public review points 6 and 7. In our responses to these public review points we now also define “strict top view”, see page 9, lines 17-19.

(11) Two papers are cited regarding a linear relationship between calcium channel number and release probability (p. 15). Many more papers could be cited to demonstrate a supralinear relationship (e.g., Dodge & Rahaminoff, 1967; Weyhersmüller et al., 2011 doi: 10.1523/JNEUROSCI.6698-10.2011). The data of the present study were collected at an extracellular calcium concentration of 0.5 mM, whereas Meideiros et al. (2023) used 1.5 mM. The relationship between calcium and release is supra-linear around 0.5 mM extracellular calcium (Weyhersmüller et al. 2011). This should be discussed/the statements be revised. Also, the reference to Meideiros et al. (2023) should be included in the reference list.

We have now updated the Medeiros reference (updated version of that paper appeared in eLife in 2024) in the text and reference list. We agree that the relationship of the calcium concentration and P_r_ can also be non-linear and refer to this on page 16, lines 26-32, but the point we want to make is to relate defined changes in calcium channel number (not calcium influx) as assessed by multiple methods (CLSM intensity measures and sptPALM channel counting) to release probability. We now also clearly state that we measured at 0.5 mM external calcium (page 16, lines 27/28) whereas Medeiros et al. 2024 measured at 1.5 mM calcium (page 16, lines 31/32).

(12) Figure 6: Quantal content does not have any units - please remove "n vesicles".

We have revised this figure in response to reviewer 2 (comment 5) and quantal content is now expressed as percent baseline, thus without units (see revised figure 7).

(13) Figure 6C should be auto-scaled from zero.

This has been fixed by revising that figure in response to reviewer 2 (comment 5)

(14) The data supporting the statement on impaired motor behavior and reduced vitality of adult IS4A should be either shown, or the statement should be removed (p. 13). Any hypotheses as to why IS4A is important for behavior and or viability?

As suggested, we have removed that statement.

(15) They do not provide any data supporting the statement that changes in PSC decay kinetics "counteract" the increase in PSC amplitude (p. 14). The sentence should be changed accordingly.

We agree and have down toned. It now reads on page 16, lines 7-9: “During repetitive firing, the median increase of PSC amplitude by ~10 % is potentially counteracted by the significant decrease in PSC half amplitude width by ~25 %...”.

(16) How do they explain the net locomotion speed increase in dI -IIA larvae? Although the overall charge transfer is not affected during the stimulus protocols used, could the accelerated PSC decay affect PSP summation (I would actually expect a decrease in summation/slower speed)? Independent of the voltage-clamp data, is muscle input resistance changed in dI-IIA mutants?

Muscle input resistance is not altered in I-II mutants. We refer to potential causes of the locomotion effects of I-IIA excision in the discussion. On page 16, lines 12 to 21 it reads: “there is no difference in charge transfer from the motoneuron axon terminal to the postsynaptic muscle cell between *∆I-IIA* and control. Surprisingly, crawling is significantly affected by the removal of I-IIA, in that the animals show a significantly increased mean crawling speed but no significant change in the number of stops. Given that the presynaptic function at the NMJ is not strongly altered upon I-IIA excision, and that I-IIA likely mediates also Ca_v_2 functions outside presynaptic AZs (see above) and in other neuron types than motoneurons, and that the muscle calcium current is mediated by Cav1>/i> and Cav3, the effects of I-IIA excision of increasing crawling speed is unlikely caused by altered pre- or postsynaptic function at the NMJ. We judge it more likely that excision of I-IIA has multiple effects on sensory and pre-motor processing, but identification of these functions is beyond the scope of this study.”